# Functional hierarchy of the human neocortex across the lifespan

Hoyt Patrick Taylor IV[1,2,3 ✉], Khoi Minh Huynh[1,2], Kim-Han Thung[1,2], Guoye Lin[1,2], Wenjiao Lyu[1,2], Weili Lin[1,2], Sahar Ahmad[1,2] & Pew-Thian Yap[1,2 ✉]

Large-scale gradients of functional connectivity between brain areas organize the human neocortex, linking brain topography to the texture of cognition[1,2]. In adults, three dominant axes—sensory–association, visual–somatosensory and modulation–representation—run, respectively, from primary sensory to transmodal association areas, from visual to body-centred systems and from control and attention networks to default mode and sensory areas[1–4]. These gradients provide a compact description of large-scale cortical hierarchies that underlie distinct modes of information processing. However, how these gradients and their multiscale biological and cognitive correlates evolve across the lifespan is unknown. Here we establish a continuous normative reference of functional organization from birth to 100 years of age, revealing complex, nonlinear developmental trajectories. Gradient architecture is anchored by primary sensory systems in infancy, differentiates along association and control axes during childhood and adolescence and gradually dedifferentiates during ageing. The importance of this functional architecture is corroborated by biology and behaviour: gradient metrics predict cognitive performance across development; structure–function coupling varies by axis and age; and distinct transcriptomic signatures are strongest early in life and weaken with age, consistent with a transient genetic scaffold for gradient architecture. Our lifespan gradients unify diverse research into developmental brain connectivity and provide a shared multimodal reference for future studies.

Understanding how brain network organization changes across the human lifespan is a central and long-standing goal in neuroscience. Previous work has shown that functional connectivity (FC) is organized along smoothly varying cortical 'gradients'[1]. These gradients stratify cortical locations according to orthogonal connectivity patterns, capturing smooth variation in fundamental motifs of network architecture. These motifs align with established progressions in information processing[2,5], supporting the interpretation of gradients as proxies for large-scale processing hierarchies.

This gradient-based framework exists in a healthy tension with the foundational principle of cortical arealization—the parcellation of the cortex into discrete areas defined by their unique cytoarchitecture and connectivity. Previous work has highlighted this apparent conflict, suggesting that the sharp boundaries between functional systems observed in individuals are at odds with the smooth, continuous nature of population-level gradients[6]. This debate is reminiscent of historical arguments in physics over light as a wave or a particle, where the resolution lies not in choosing one correct description, but in understanding the context in which each provides value. A key test of any framework's validity is therefore its explanatory utility: its ability to provide novel insights into brain function and behaviour. The initial discovery of the principal functional gradient, for example, provided a new explanation for the spatial distribution of functional networks[1], an insight that had not emerged from purely arealization-based studies. Therefore, a central goal of this study is to directly test the explanatory utility of the lifespan gradient framework by linking its organizational metrics to cognitive and behavioural outcomes, microstructural and morphological metrics and gene ontology data from infancy through to old age. By establishing a normative timeline of gradient development and demonstrating its behavioural relevance, we aim to provide compelling evidence for the value of gradients as a crucial organizing principle of the human brain.

In adults, FC is reliably organized along three dominant gradients that together provide a compact coordinate system for large-scale cortical architecture[1,2]. These axes are often interpreted as topographic hierarchies[2,7,8] that enumerate continuous transitions between processing motifs rather than strict causal or unidirectional pathways. The most well-studied connectivity gradient is the canonical sensory–association (SA) axis. Explaining the most variance in FC, the SA axis is anchored at one end by primary unimodal cortex and at the other end by high-order association cortex coinciding with the default mode network[1,2]. Similarly, repeated observations have shown that the gradient that explains the second-most variance in FC is an axis spanning between visual and somatosensory cortex (the VS axis), smoothly partitioning cortical locations according to their preferential involvement in modality-specific processing[1,2]. Finally, many studies have identified a tertiary gradient

[1]Department of Radiology, University of North Carolina, Chapel Hill, NC, USA. [2]Biomedical Research Imaging Center, University of North Carolina, Chapel Hill, NC, USA. [3]Department of Computer Science, University of North Carolina, Chapel Hill, NC, USA. ✉e-mail: hptaylor@live.unc.edu; ptyap@med.unc.edu

of FC, often described as a modulation–representation (MR) axis, anchored at opposite ends by cortical regions that are preferentially involved in modulation (frontoparietal control and attention networks) versus regions that are involved in representation (default, sensory and visual networks)[1,3,4].

Establishing a normative timeline for the development of these governing motifs is essential to gain a principled understanding of brain disorders. Many neurodevelopmental and psychiatric conditions present diverse alterations across sensory, cognitive and social domains that have proved difficult to unify under a single explanatory model[9]. A framework centred on information-processing hierarchies, as captured by functional gradients, could help to consolidate previous models[10]. Enumerating normative lifespan trajectories is therefore a crucial step for understanding how deviations from these trajectories might give rise to multifaceted symptoms across neurodevelopmental and neurodegenerative disorders.

Several studies have characterized how one or more of these axes change during specific developmental periods, providing a foundation for a multi-dimensional lifespan gradient atlas[11,12]. One study[13] reported altered gradient ordering during childhood, with the VS axis explaining the most variance in FC and an SA-like axis (with notable topographic differences) constituting the secondary gradient. Sliding-window analyses from 6 to 17 years suggested a convergence towards adult ordering during early adolescence, with more protracted development in high-order association cortex than in unimodal cortex[13]. In later adulthood, three-dimensional gradient embeddings of canonical resting-state networks become increasingly dispersed, consistent with decreasing within-network coherence during ageing[14]. However, to our knowledge, no work has so far offered a cohesive analysis of gradient architecture spanning from infancy to old age.

Concurrently, a vast body of non-gradient analyses provides crucial context. During infancy and early childhood, spatially dispersed, high-order resting-state networks are weakly expressed, with FC dominated by strong local connections in unimodal cortex[15,16]. During childhood, the emergence of complex cognition coincides with a transition towards an adult-like FC architecture. Graph-theoretic work highlights the importance of integration, segregation and modular architecture in this transition, supporting the emergence of high-order systems, including default and control networks[17–19]. Because transmodal networks are distributed, their strengthening is enabled by increases in long-range connection density and strength, mediating the transition from local infant to more global adult-like FC organization by late childhood and early adolescence[20].

Once adult-like FC organization is established, adolescence and early adulthood coincide with subtler refinements to connectivity architecture[21]. In particular, fine-tuning of integration and segregation within high-order default mode, frontoparietal control and attention networks increases during this period, yielding more pronounced information-processing hierarchies[22,23]. During ageing, degradation of the segregation–integration balance that scaffolds hierarchical FC architecture has been associated with cognitive decline[24,25]. Moreover, this period is characterized by a global dedifferentiation in FC that coincides with decreasing fluid intelligence and working memory.

Previous FC gradient studies provide useful insights into development during particular periods, but none encapsulate the full lifespan, and methods and reporting vary widely. We address these shortcomings by analysing functional magnetic resonance imaging (fMRI) data from healthy individuals, encompassing neonates to centenarians, in a unified framework. Across the lifespan, data are preprocessed identically, FC gradients are aligned to a common space capturing axes of FC organization and vertex-wise trajectories are modelled using generalized additive mixed models (GAMMs). Using a large sample and consistent methodology, we chart lifespan trajectories of validated axes of FC organization and characterize age-related changes in gradient topography, global features and the multi-dimensional embedding that the gradients collectively constitute. Finally, we evaluate explanatory utility by relating gradients to cognitive measures, morphological and microstructural metrics, gene-expression data and meta-analytic task activation.

## Lifespan gradient atlas

To characterize lifespan changes in intrinsic FC organization, we computed vertex-wise FC gradients for 3,556 individuals (3,972 time points; 16 days–100 years) and aligned individual-specific gradients to a common lifespan template (Fig. 1 and 'Functional gradients' in Methods). We focused on three reproducible axes—sensory-association (SA), visual–somatosensory (VS) and modulation–representation (MR)—and modelled vertex-wise trajectories across age using GAMMs ('Lifespan trajectories' in Methods). This yields a continuous normative atlas of cortical functional hierarchy across the lifespan.

## Topography changes peak in early life

A central objective of this study was to characterize how the cortical topography of the SA, VS and MR axes varies with age. To this end, we analysed our vertex-wise GAMMs of gradient values mapped to the cortical surface (Fig. 2a and 'Lifespan trajectories' in Methods). The distribution of values for each gradient, depicted in a density plot, reflects the corresponding hierarchical architecture and allows for the characterization of global changes in the gradient. Peaks in gradient-value density plots arise when many vertices share the same FC profile with respect to a gradient, whereas uniform, widespread density distributions indicate highly heterogeneous or stratified FC. The cortical topography of each gradient provides insight into the spatial realization of crucial patterns of variation in connectivity. Notably, gradient topography changes most markedly during the first four years of life, consistent with observations that infancy and early childhood are key periods for development towards a large-scale adult-like network architecture[15,16,26–28].

In adulthood, the SA axis describes the unimodal-to-transmodal functional hierarchy and exhibits stable topography from adolescence through ageing. By contrast, early life shows a markedly immature SA organization: the association pole is weakly differentiated and SA values exhibit a more local, unimodal-driven pattern, consistent with a 'proto-SA' architecture (Extended Data Fig. 1). Between two and ten years, SA topography reorganizes towards its mature form, with progressive focalization and stratification of the association pole converging on canonical default mode hubs (Extended Data Fig. 2 and Supplementary Fig. 6), indicating that childhood is a key window for establishing the SA hierarchy.

The MR gradient—spanning control and attention systems at its modulation pole, and default mode and sensory regions at its representation pole—shows the weakest resemblance to its mature layout at birth. Both poles sharpen substantially across childhood, yielding increasingly adult-like differentiation between modulation- and representation-oriented cortices by ten years (Extended Data Fig. 2 and Supplementary Fig. 6). Notably, visual cortex anchors the representation pole during the first year of life, whereas somatosensory cortex approaches the representation pole later, with continued refinement into early adulthood (Extended Data Fig. 3), consistent with a protracted maturation of MR organization.

In contrast to the high-order information-processing hierarchies corresponding to the SA and MR axes and the complex topographic development that they undergo, the VS axis mainly differentiates visual from somatosensory systems and is comparatively stable across the lifespan. Its global range is strongest in early life (peaking around early childhood) and subsequently decreases, indicating that modality-specific segregation is most prominent early in

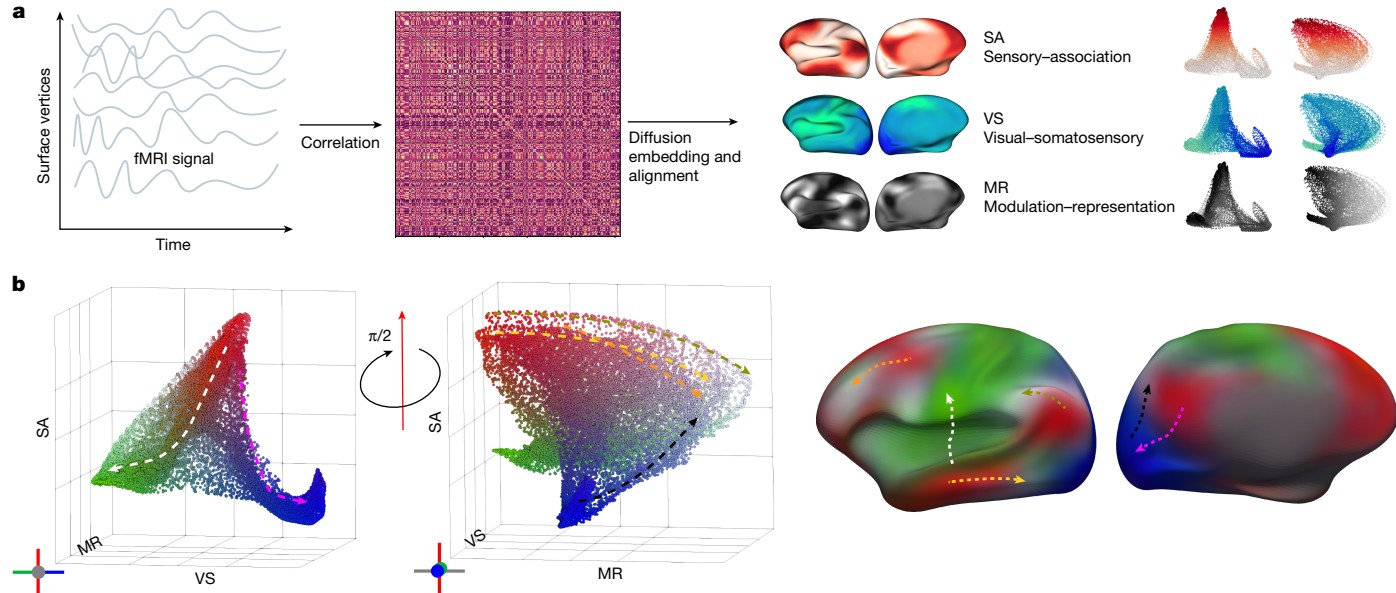

**Fig. 1 | Overview of gradient manifold computation and its interpretation. a**, For each individual, the fMRI signal is mapped to the cortical surface and an FC matrix is computed using the Pearson correlation coefficient. Individual FC gradients are obtained through diffusion embedding applied to the FC matrix, and are aligned to the template gradient axes of interest: SA, VS and MR. These axes, respectively, differentiate cortical locations by their implication in association (red) versus unimodal (white), their preferential recruitment in either visual (blue) or somatosensory (green) domains and their propensity to engage in top-down modulation (white) versus representation (black). **b**, The taxonomy of the adult gradient manifold in the embedding space (left) and the cortical surface (right) with a unified colour map that combines the three used in **a**. Several paths along the gradient manifold are shown with corresponding colour-coded paths along the cortical surface, demonstrating the cortical realization of hierarchies enumerated by the gradient manifold.

life and becomes a progressively less dominant feature of large-scale organization with maturation (Supplementary Fig. 6). Together, these developmental patterns reflect early anchoring by primary systems, followed by childhood differentiation along association and control axes and a later shift towards more multimodally integrated functional architecture.

## Global gradient signatures of maturation

We observe significant nonlinear developmental trajectories in global properties of all three gradient axes and the 3-dimensional embedding they constitute (Fig. 3a–e and 'Gradient measures' and 'Lifespan trajectories' in Methods). To quantify the global expression of each axis, we computed gradient range as the inter-vigintile spread (5th–95th percentile) and fitted GAMMs to the ranges of the SA, VS and MR gradients (Fig. 3a). SA and MR exhibit protracted expansion across infancy, childhood and adolescence, peaking in early adulthood (SA: 18.8 years, 95% confidence interval 15.0–20.8; MR: 19.0 years, 95% confidence interval 15.5–21.3), followed by relative stability in early adulthood and contraction thereafter that is more pronounced for SA than for MR. Both axes thus show an inverted U-shaped lifespan profile, consistent with previous studies of FC development across the lifespan[29]. By contrast, VS reaches its maximum range in childhood (5.1 years, 95% confidence interval 4.8–5.5) and then contracts gradually across the remainder of the lifespan (Fig. 3a), mirroring alternate depictions of VS contraction in the density tails and SA–VS embedding plots (Fig. 2a,b), and previous reports of decreasing VS dominance and increasing long-range FC organization across development[20,28]. Because gradient range reflects differentiation in FC profiles between vertices near each pole, SA and MR expansion indicate increasing dissimilarity between unimodal and transmodal systems and a growing contrast between executive–modulatory and representation-implicated systems. The protracted expansion of SA and MR is consistent with evidence for late maturation of heteromodal association and modulatory–executive systems[30–33].

To characterize the global differentiation of FC profiles in the embedding space, we computed gradient dispersion (mean Euclidean distance of vertices to the embedding centroid) and modelled its trajectory with a GAMM (Fig. 3b and 'Gradient measures' in Methods). Dispersion increases through infancy and childhood, peaks at 13.8 years (95% confidence interval 11.3–16.3), remains relatively stable into early adulthood and declines steadily thereafter, consistent with reduced functional segregation during ageing[34,35]. This trajectory reflects increasing cortex-wide differentiation during development (coincident with SA–MR expansion and VS contraction; Fig. 3a) followed by increasing global homogeneity in later adulthood as all three axes contract (Fig. 3b).

Finally, we assessed the fidelity of individual gradient topography to our template gradients using cosine similarity for each axis (Fig. 3c and 'Functional gradients' and 'Gradient measures' in Methods). Because cosine similarity is insensitive to overall scaling, it reflects vertex-wise correspondence of gradient topography to the template axes rather than changes in gradient magnitude. Template similarity increased from birth through adolescence, reaching maxima in the expected maturational order (VS: 9.8 years; SA: 13.5 years; MR: 15.8 years), consistent with consolidation of the canonical SA–VS–MR embedding geometry. Similarity then declined and its variance increased in later adulthood (most prominently for VS), indicating progressively less-canonical gradient topographies during ageing. To contextualize these gradient-based global measures with conventional graph metrics, we quantified each participant's mean FC degree (mean nodal strength; Fig. 3e) and related it to gradient dispersion (Fig. 3d). We found that mean FC degree was positively associated with dispersion, indicating that individuals whose embeddings are more expanded in the joint SA–VS–MR space also tend to exhibit greater global connectedness of the functional connectome. Across the lifespan, mean FC degree increased from infancy to a peak near ten years and then declined (Fig. 3e). This developmental profile did not mirror any single axis range, supporting the interpretation that gradient-derived measures capture organizational features beyond uniform shifts in overall FC strength.

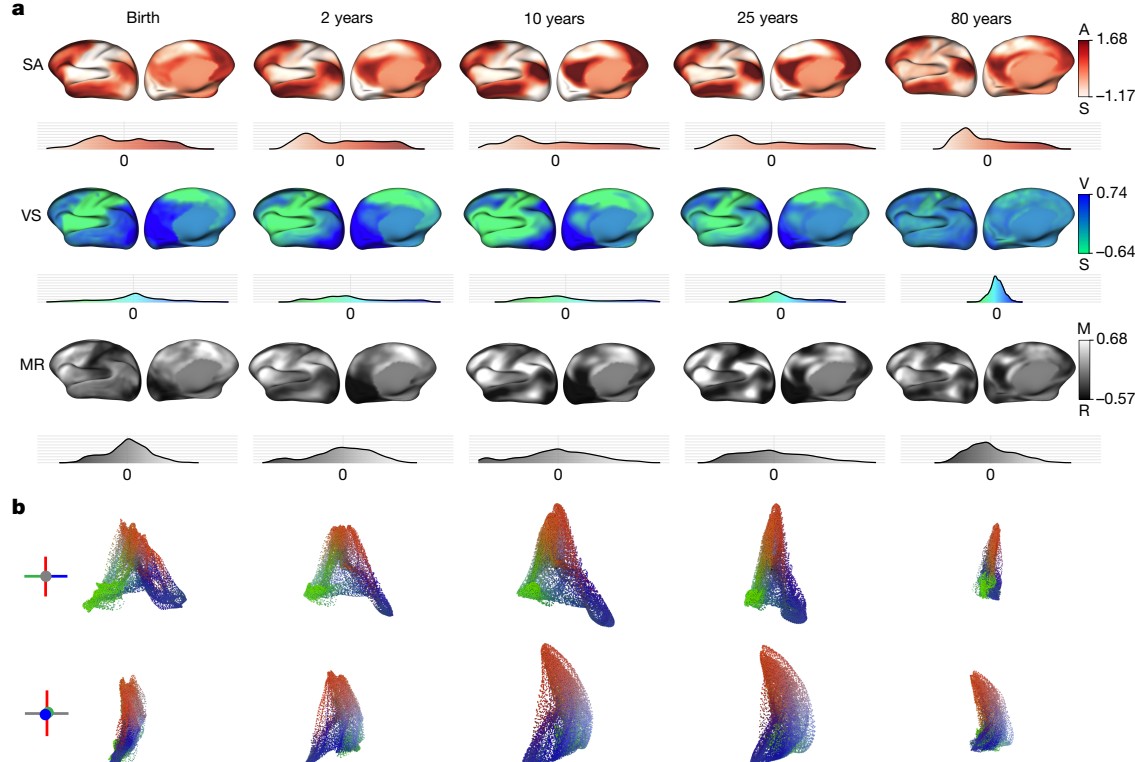

**Fig. 2 | Gradient organization across the human lifespan.** SA, VS and MR gradients of FC undergo differential development across the human lifespan. **a**, GAMM fits of FC gradient value are plotted on the cortical surface, with density plots showing the gradient value distribution at selected time points. **b**, Embedding plots given by gradient GAMM fits in the SA–VS plane (top) and the SA–MR plane (bottom); in the SA–MR view, MR polarity is oriented left to right from representation (R) to modulation (M). Time points were selected to reflect key developmental milestones across the lifespan. GAMMs were fit using $n = 3,972$ individual gradient sets.

## Networks reorganize along gradients

Canonical resting-state networks are stratified along primary FC gradients[1,36], and developmental change seems to unfold along these axes, with unimodal systems maturing earlier than association, limbic and attention networks[30,35,37–39]. To quantify lifespan reconfiguration of network organization in gradient space, we used the Schaefer 7-network parcellation[40] and computed, for each network, the mean gradient value (that is, the network centroid) on each axis. We then fit GAMMs to network centroids versus age (Fig. 4a–d and 'Gradient measures' and 'Lifespan trajectories' in Methods). Networks were ordered as expected, with default and control networks occupying the association and modulation extremes of the SA and MR axes, respectively, and visual and somatomotor networks occupying the poles of the VS axis. Notably, centroids followed distinct, axis-specific trajectories across the lifespan (Fig. 4a), indicating complex maturational timing relative to the principal axes of FC organization.

Along the SA axis, developmental expansion and later contraction (Fig. 3a) correspond to systematic reconfiguration of networks along the unimodal-to-transmodal hierarchy. Default and control networks move towards the association pole from birth to mid-adolescence (Fig. 4a, left), whereas salience–ventral attention, dorsal attention and visual networks shift gradually towards the sensory pole through early adulthood and become more central thereafter. The somatomotor network shows a distinctive early trajectory, moving from an extreme sensory position at birth towards a more central position by around three years, before converging with the attention and visual trajectories (Fig. 4c). This early-life somatomotor centroid shift is consistent with SA topographic reorganization, in which the unimodal pole is at first concentrated in somatomotor cortex and later becomes more balanced with the visual system (Fig. 2a).

Along the VS axis, heteromodal networks (default, control, limbic and dorsal attention) remain centred, whereas the visual and somatomotor networks occupy opposite poles throughout life, with salience–ventral attention positioned on the somatomotor side (Fig. 4c). Network separation along VS is greatest in infancy and early childhood (Fig. 4a, middle): the somatomotor centroid is most extreme at birth and drifts centrally across the lifespan, whereas the visual network remains near the visual pole throughout development. Pole networks (visual, somatomotor and salience–ventral attention) are maximally separated between birth and around ten years and contract rapidly thereafter, in tandem with global VS range contraction (Fig. 3a).

Network dynamics are most complex along the MR axis (Fig. 4c). Consistent with MR differentiating modulation versus representation, control and attention networks occupy the modulation pole, whereas default and unimodal systems lie towards the representation pole. Default and control become increasingly separated from birth to early adulthood, driven by a rapid early migration of default towards representation (birth to around five years) and a sustained migration of control towards modulation through early adulthood. During adulthood, the control centroid shifts centrally, coincident with MR contraction (Fig. 2b). Attention networks also diverge in timing: dorsal attention reaches its maximal modulation-pole position in early adolescence, whereas salience–ventral attention shows more protracted development, peaking in mid-adulthood. We also investigated the interaction of resting-state networks in our embedding space, revealing nonlinear lifespan trajectories (Extended Data Fig. 4 and 'Network interactions' in Supplementary Results).

## Structure–function coupling

To relate cortical microstructure to the principal FC gradients across the lifespan, we computed individual-specific structural gradients

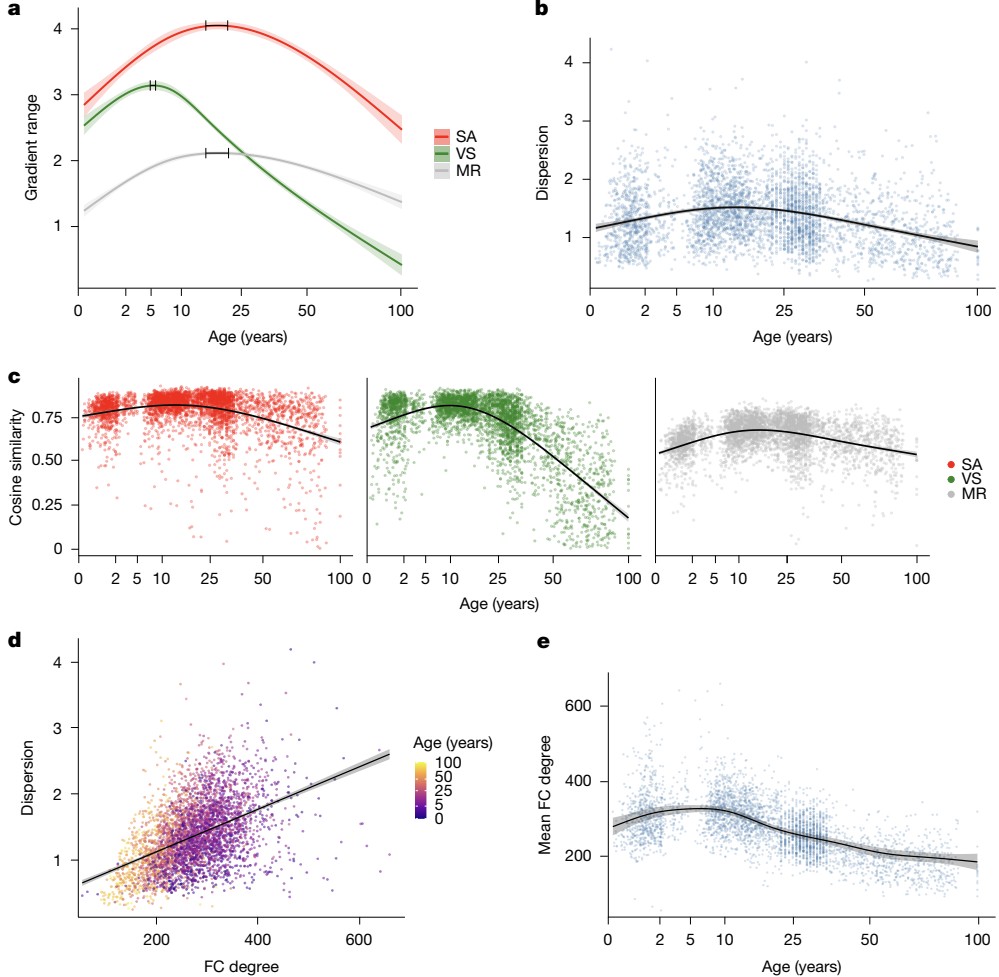

**Fig. 3 | Global gradient metrics denote changes in hierarchical FC architecture across the lifespan. a**, Inter-vigintile gradient range versus age with GAMM fits for the SA, VS and MR axes. **b**, Gradient dispersion versus age with GAMM fit. **c**, Cosine similarity between individual gradients and the template gradient set used for alignment. **d**, FC degree versus gradient dispersion; black line shows an ordinary least squares linear regression with 95% confidence interval for the mean. Association was quantified using a Pearson product–moment correlation (two-sided): $r = 0.455$ (95% confidence interval: 0.430–0.479), $P = 2.79 \times 10^{-202}$ (no multiple-comparisons adjustment). **e**, Mean FC degree across the lifespan with GAMM fit. Across all panels, $n = 3$, 972 gradient sets. Solid lines show population-level GAMM fits; shaded ribbons indicate 95% pointwise confidence intervals of the fitted mean.

from multivariate cortical-feature affinity matrices. We then aligned each individual's structural embedding to their corresponding template-aligned functional gradients (Fig. 5a–f and 'Structural gradients' and 'Structure–function coupling' in Methods). Structural and functional counterparts exhibited modest spatial correspondence, with mean spatial correlations of $\rho = 0.45$ (SA), $\rho = 0.41$ (VS) and $\rho = 0.38$ (MR), indicating that the dominant axes of functional organization are rooted only moderately in the underlying cortical microstructure. We quantified axis-specific structure–function coupling by computing cosine similarity between aligned structural and functional gradients and fitting GAMMs to obtain lifespan trajectories (Fig. 5b).

Despite alignment, structural gradients remained topographically distinct from functional gradients (Fig. 5a). Correspondence was strongest for SA, whereas VS- and MR-aligned structural gradients diverged more substantially from their functional counterparts, suggesting that microstructural organization provides a stronger underpinning for the unimodal–transmodal SA hierarchy than for the orthogonal VS and MR axes.

Structure–function coupling decreased nonlinearly with age (Fig. 5c). Coupling declined rapidly for SA and MR during infancy and early childhood, whereas VS coupling was comparatively stable early in life and declined more modestly thereafter. In parallel, structural-gradient range exhibited axis-specific lifespan trajectories (Fig. 5d), indicating

that the scale of microstructural differentiation evolves over development and ageing in a manner that is not isomorphic to functional differentiation.

Finally, we related the SA axis to individual microstructural metrics across age (Extended Data Fig. 5). Myelination showed the most persistent alignment with SA across the lifespan[41], whereas cortical thickness exhibited robust positive coupling, consistent with thicker association cortex[42]. Several metrics showed developmental sign changes, indicating that the microstructural features that track the SA hierarchy vary across developmental epochs. Together, these results show that microstructural organization provides a measurable but incomplete substrate for functional-gradient organization, with strongest coupling for SA and a general decoupling with age[43,44].

## Gradients predict lifespan cognition

We modelled domain-specific cognitive scores, functional-gradient metrics and their interactions using generalized additive models (GAMs) (Extended Data Fig. 6a,b and 'NIH Toolbox Cognition Battery' in Methods). Extended Data Fig. 6a summarizes, for each gradient–domain pair, the main effect at the pooled median age (30 years), quantified as the predicted cognitive difference between individuals at the 90th and 10th percentiles of the gradient. The most prominent

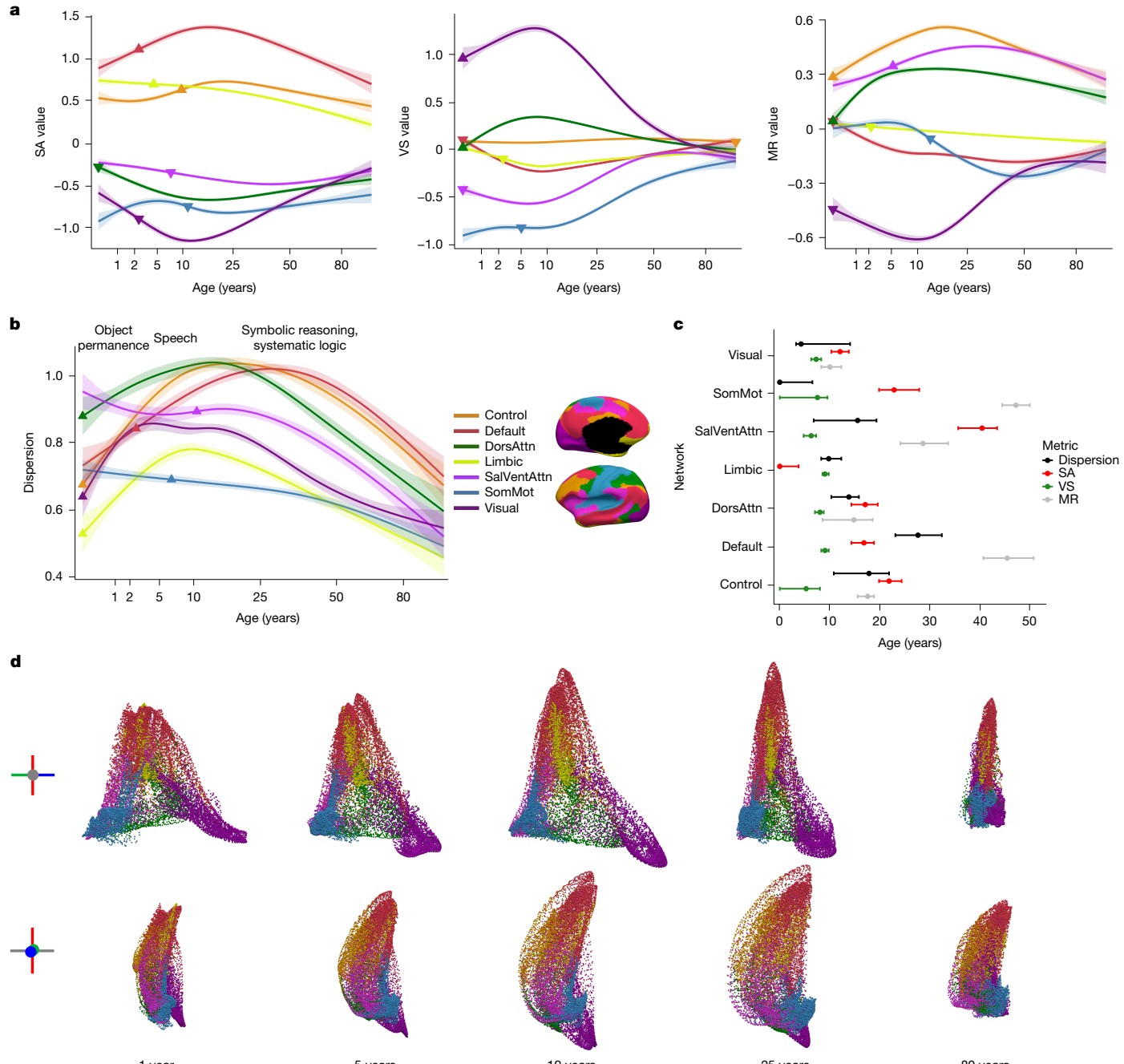

**Fig. 4 | Lifespan development of functional gradients with respect to canonical resting-state networks. a**, Population-level GAMM fits of mean gradient value versus age within each Schaefer 7-network parcel for the SA, VS and MR axes; maximum or minimum lifespan derivatives of mean network-wise gradient values are marked on the trajectories. **b**, GAMM fit of within-network dispersion versus age, computed as the mean Euclidean distance between each vertex's coordinate in the SA–VS–MR embedding space and the corresponding network centroid. DorsAttn, dorsal attention; SalVentAttn, salience and ventral attention; SomMot, somatomotor. **c**, Network maturation ages for gradient value and dispersion. For each network and metric, maturation age is defined as the age at which the fitted trajectory reaches its extremum (maximum or minimum, as appropriate); points denote the posterior median extremum age and horizontal error bars denote 95% highest-density intervals (HDIs) estimated from coefficient resampling. **d**, GAMM-fitted gradient embedding plots coloured by the Schaefer 7-network parcellation at selected ages, shown in the SA–VS (top) and SA–MR (bottom) planes; in the SA–MR view, MR polarity is oriented left to right from representation (M) to modulation (R). In **a**,**b**, $n = 3$, 972 gradient sets; shaded ribbons indicate approximate 95% confidence intervals of the fitted mean.

and widespread associations involve SA cosine similarity, which is positive across multiple domains (including composite cognition, attention and reading), with weaker but consistent support for SA range and global dispersion. By contrast, VS- and MR-linked metrics and gradient eigenvalues (variance captured by each embedding axis before alignment) show more selective effects at this age (Extended Data Fig. 6a).

Extended Data Fig. 6b shows how these associations shift with age. Two patterns emerge: (i) the cognitive benefit of SA fidelity decreases with age across several domains, consistent with dedifferentiation of the principal SA hierarchy; and (ii) the influence of pre-alignment eigenvalues and MR range increases with age in selected domains, suggesting a growing relevance of global embedding geometry and MR-axis dynamic range later in life (Extended Data Fig. 6b). These

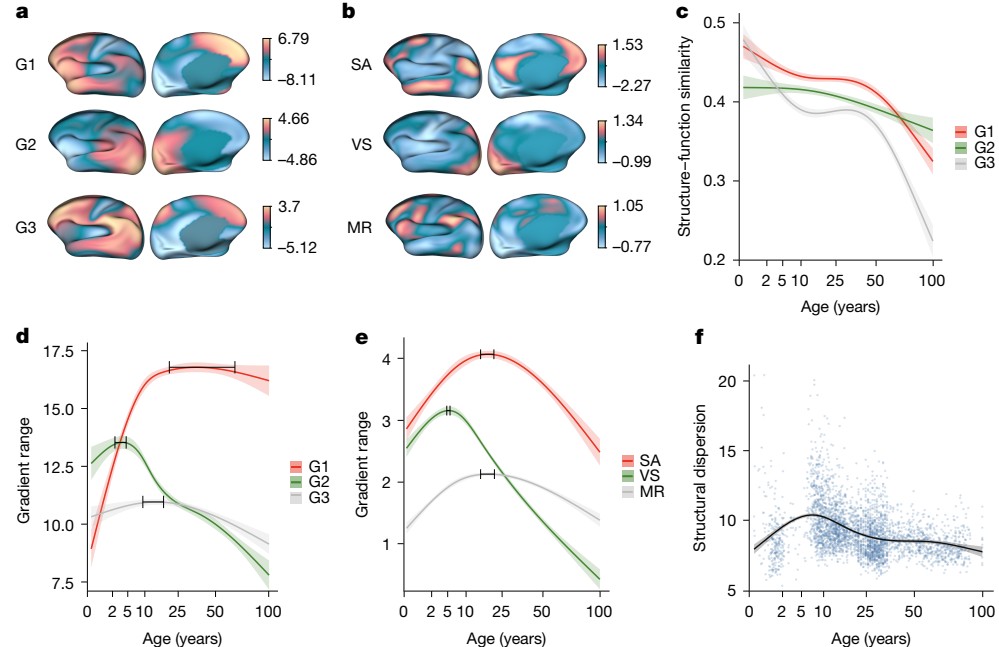

**Fig. 5 | Structural gradients and structure–function coupling across the lifespan. a**, Group-level structural gradients (G1–G3) derived from individual-specific morphometric similarity networks (MSNs) constructed from cortical thickness, myelination and microstructural indices. **b**, Functional-gradient template axes (SA, VS and MR). **c**, Structure–function coupling across the lifespan, quantified as cosine similarity between each individual's structural and functional gradients after Procrustes alignment (G1/SA, G2/VS, G3/MR; $n = 3,431$ individuals). **d**, Structural-gradient range across the lifespan (inter-vigintile range; $n = 3,431$ gradient sets); horizontal bars denote 95% HDIs for the age of the peak of each fitted trajectory. **e**, Functional-gradient range across the lifespan (inter-vigintile range; $n = 3,972$ gradient sets); horizontal bars denote 95% HDIs for the age of the peak of each fitted trajectory. **f**, Dispersion of structural gradients across the lifespan (mean Euclidean distance to the embedding centroid; $n = 3,431$ gradient sets). In **c**–**f**, solid lines show population-level GAMM fits of each metric versus age with a random intercept for cohort. Shaded ribbons indicate 95% pointwise confidence intervals of the fitted mean. HDIs in **d**,**e** were computed from 20,000 draws of the smooth-term coefficient posterior and summarize uncertainty in the age at which the fitted trajectory attains its maximum.

lifespan patterns reconcile cohort-specific findings: effects in the young-adult cohort of the Human Connectome Project (HCP-YA) (Extended Data Fig. 6c) align with the median-age slice in Extended Data Fig. 6a, whereas effects in the Baby Connectome Project (BCP) (Extended Data Fig. 6d) reflect earlier, domain-specific associations that have not yet generalized.

We next tested whether inter-individual variation in FC gradients predicts behaviour in two independent cohorts: infants and toddlers from BCP and young adults from HCP-YA. Identical metrics were evaluated in both datasets: dispersion, axis ranges, cosine similarity to SA, VS and MR, and the first three alignment-agnostic eigenvalues (Extended Data Fig. 6c,d and 'NIH Toolbox Cognition Battery' and 'Mullen scales of early learning' in Methods).

In BCP, age-adjusted random-intercept mixed-effects models revealed a small number of reliable associations after false discovery rate (FDR) correction (Supplementary Table 1 and Extended Data Fig. 6d), mainly involving MR cosine similarity and the third eigenvalue. Thus, by early childhood, both axis-specific topographic coherence (MR fidelity) and higher-order embedding structure explain a modest but significant fraction of variance in emerging skills.

In HCP-YA, a substantially larger set of gradient–cognition associations survived FDR correction (Supplementary Table 2 and Extended Data Fig. 6c). The dominant signal was the pervasive predictive power of the SA axis: SA cosine similarity showed positive associations across cognitive domains, with related SA metrics (dispersion and SA range) most strongly tracking composite measures of total and fluid cognition. Out-of-sample checks within HCP-YA were directionally consistent (see 'Out-of-sample checks in HCP-YA' in Supplementary Results).

Together, both cohorts support the behavioural relevance of functional-gradient architecture and suggest a developmental consolidation: modest, domain-specific associations in infancy mature into broader coupling in young adulthood, and SA fidelity in particular is robustly associated with cognitive measures. Interpreted through a hierarchical framework, dispersion and range measure differentiation along the gradients, whereas cosine similarity quantifies the topographical fidelity of the hierarchy itself. From this perspective, a well-organized and clearly expressed SA axis seems to be a key feature of mature brain function supporting high-level cognition.

## Meta-analytic gradient validation

We derived a canonical association–unimodal axis from Neurosynth[45] (Extended Data Fig. 7a–f and 'Neurosynth meta-analytic validation' in Methods) by parcellating 24 term maps (Extended Data Fig. 7b illustrates four example term maps) according to the Schaefer-400 parcellation, $z$-scoring each term map across parcels and contrasting the mean map of association-domain terms (for example, theory of mind, working memory, language and default mode) with the mean map of unimodal terms (visual, somatosensory, motor; attention, eye movements and pain excluded as intermediates). The resulting parcel-wise axis (Extended Data Fig. 7a) recapitulates the expected SA topography. Crucially, this meta-analytic axis strongly aligns with the SA axis (parcel-wise Spearman $\rho = 0.80$), indicating that the primary functional gradient recovered from resting-state data is well captured by an independent, task-derived unimodal–transmodal hierarchy. Extended Data Fig. 7c shows individual-level similarity between SA and the canonical axis versus age; as SA topography shifts during infancy, alignment decreases at first and then increases through development, stabilizing by early adulthood.

Median term–axis alignments show the predicted polarity for the SA axis (Extended Data Fig. 7d): association-related constructs (theory of mind, social cognition and self-referential or semantic processing)

load positively, whereas unimodal constructs (visual, somatosensory and motor) load negatively, showing that meta-analytic functional content is broadly distributed along the axis and anchored by unimodal and associative systems. Performing the same analysis for VS and MR further supports our functional interpretation. VS shows positive alignment with visual and attention terms and negative alignment with somatosensory, tactile and motor terms, consistent with a visual–somatosensory differentiation (Extended Data Fig. 7e). MR shows positive alignment with cognitive-control constructs (cognitive control, response inhibition, working memory, task switching, conflict and error monitoring) and negative alignment with representation-oriented terms (default mode, episodic memory, autobiographical memory and self-referential), consistent with a modulation-versus-representation axis (Extended Data Fig. 7f).

Beyond median ordering, the distribution of individual-level term–axis correlations in Extended Data Fig. 7d–f captures inter-individual heterogeneity in functional alignment. Variability is lowest for SA, suggesting that the SA hierarchy is not only robust at the group level but also consistently instantiated across individuals. By contrast, VS and especially MR exhibit wider distributions, consistent with greater sensitivity to individual differences and potentially age-related variation relative to mostly adult-derived meta-analytic task maps. Nevertheless, the axis-specific median term ordering remains coherent, reinforcing that VS and MR capture interpretable functional distinctions even as their expression is more heterogeneous than SA.

## Transcriptomic enrichment of gradients

To relate cortical gene expression to parcel-wise mean gradient values across development, we fitted partial least squares (PLS) models at selected ages (0.5, 2, 10, 25, 40 and 80 years) for each gradient axis ('Transcriptomic enrichment' in Methods). PLS models were fitted on adult human cortical gene expression (Allen Human Brain Atlas (AHBA)[46], abagen[47], Schaefer-400 parcellation) and oriented so that the correlation between the first PLS parcel score and the mean gradient value was non-negative at each age. The resulting component maps define smooth molecular axes across cortex for SA, VS and MR (Extended Data Figs. 8a, 9a and 10a; gene–gradient coupling compared to parcel-shuffle null models for each axis shown in Supplementary Fig. 13).

For the SA axis, the PLS map at 25 years recapitulates the SA hierarchy (Extended Data Fig. 8a). Gene ontology (GO) enrichment of genes ranked by PLS coefficient revealed a persistent synaptic signalling signature across ages, accompanied by vesicle-cycle themes (Extended Data Fig. 8c). The top GO term at 25 years, trans-synaptic signalling, produced a gene-set score map that closely matched both the SA PLS component and SA topography (Extended Data Fig. 8b). Gene–gradient coupling (permutation-tested; Benjamini–Hochberg-corrected across ages) peaked early in life and declined into adulthood (Extended Data Fig. 8d), indicating that a fixed adult-like molecular axis explains more SA topography during early development than later life.

Applying the same procedure to VS yielded a component emphasizing visual cortex, with lower scores in somatosensory and association territories (Extended Data Fig. 9a). GO enrichment highlighted transcriptional and RNA-metabolic regulation alongside neurite- and transport-related processes, with theme trajectories suggesting stronger vesicle-cycle enrichment in the early years and stronger transcription and RNA-metabolism enrichment in young to mid-adulthood (Extended Data Fig. 9c). The most significant VS gene set—regulation of mRNA metabolic process—generated a gene-set score map that closely resembled the VS PLS component (Extended Data Fig. 9b). VS gene–gradient coupling was highest in infancy, declined sharply through childhood, stabilized into mid-adulthood and declined again towards late life (Extended Data Fig. 9d).

For MR, the first PLS component contrasted unimodal regions (negative scores in visual and premotor cortex) with high-order control regions (positive scores in medial and lateral prefrontal and anterior temporal cortex), mirroring the MR axis (Extended Data Fig. 10a). The top GO term, detoxification of copper ion, produced a gene-set score map that closely matched the MR PLS topography (Extended Data Fig. 10b). At the theme level, ion transport and excitability emerged as the most consistent signal, with the strongest enrichment in late life (Extended Data Fig. 10c). MR gene–gradient coupling decreased monotonically across the lifespan (Extended Data Fig. 10d).

Together, these analyses indicate that all three gradients are associated with distinct but partially overlapping molecular programs: synaptic signalling and vesicle cycling for SA; transcriptional and RNA-metabolic regulation with a visual emphasis for VS; and ionic homeostasis and excitability for MR (Extended Data Figs. 8–10). In each case, gene–gradient coupling is strongest early in life and weaker in later adulthood, suggesting that adult-like spatiomolecular axes provide a scaffold for early differentiation of FC that becomes progressively less predictive of functional organization across the lifespan.

## Discussion

In this work, we charted the development of the functional connectome's fundamental organizing axes across the human lifespan. Our findings unify decades of developmental FC research under one comprehensive normative chart of gradient organization, recapitulating known developmental timing while revealing continuous, nonlinear trajectories from infancy into late old age. Integrating the vast and diverse body of fMRI literature into an easily interpretable and succinct framework is a central challenge in neuroscience; our study provides a substantial step towards achieving this goal, by establishing a shared coordinate system for cortical hierarchy. Although cortical morphology has been mapped in detail throughout the human lifespan[48], comparable normative charts for the global architecture of intrinsic FC from days after birth into late old age have been lacking until now. Notably, we substantiate the cognitive and multiscale organizational relevance of the principal FC gradients by relating them to cognitive measures, meta-analytic task activation maps, cortical microstructure and morphology and transcriptomic enrichment.

Our findings on cognition provide a crucial empirical anchor for these lifespan dynamics, directly addressing the question of the gradient framework's explanatory utility. In young adults, the SA axis emerged as a robust, domain-general correlate of cognitive performance: the topographical similarity of an individual's SA gradient to the canonical template was significantly associated with every cognitive domain assessed, from fluid and crystallized intelligence to processing speed and memory. This result provides strong evidence that a more faithful instantiation of the brain's primary hierarchical axis is broadly associated with higher cognitive performance. This relationship also seems to consolidate with age, as the strength of gradient–cognition associations increased from infancy to young adulthood. The early, more specific link we observed between MR metrics and emerging motor skills in infants further underscores that these organizational principles are functionally important from the earliest stages of postnatal life. The GAM analysis across the development, young-adult and ageing cohorts of the HCP (HCP-D, HCP-YA and HCP-A, respectively) clarifies how these relationships change with age: SA cosine similarity remains the dominant domain-general predictor around the pooled median age (Extended Data Fig. 6a), but its benefit attenuates across later adulthood (Extended Data Fig. 6b), whereas alignment-agnostic embedding metrics (eigenvalues) and MR range show selective increases with age. Together, these patterns reconcile the cohort findings and suggest that the same hierarchical scaffold supports cognition across the lifespan, whereas the gradient features that are most predictive of behaviour shift as functional organization dedifferentiates with ageing.

Complementing these behavioural associations, our Neurosynth validation ties the gradients to a canonical cognitive ontology. An

independently constructed association–unimodal axis closely recapitulated the SA axis (parcel-wise $\rho$ = 0.803), and term-wise alignments showed the predicted polarity for SA (association terms positive; unimodal terms negative). The VS axis aligned positively with visual and attention terms and negatively with somatosensory, tactile and motor terms, reinforcing its interpretation as a visual–somatosensory differentiation. Most importantly, the MR axis exhibited a strong positive alignment with cognitive-control constructs and negative alignment with representation-oriented constructs (Extended Data Fig. 7f), directly confirming our modulation–representation interpretation of this gradient axis. Because MR interpretation has been investigated to a lesser extent than SA has, this provides an independent functional anchor for our third axis.

Developmental trajectories were distinct for each of the three primary gradients we studied: the SA, VS and MR axes. We first sought to characterize topographic changes in these axes over the lifespan, observing that the largest changes occur by around four years, followed by continued childhood refinement (notably SA and MR) and more protracted tuning into adolescence and early adulthood. In keeping with previous observations that infant FC architecture is distinct from that of adults[20,27], we observe markedly altered SA and MR gradient topography during early life, consistent with weakly developed high-order information-processing hierarchies and limited long-range integration. Conversely, the VS axis is established earlier and its topography remains mainly stable while its global range steadily declines throughout adolescence and adulthood, indicating that the differentiation between primary sensory modalities becomes a progressively less dominant organizing motif with maturation.

The SA and MR gradients both enumerate information-processing hierarchies that are crucial to complex cognition, with high-order association and control regions positioned at their apexes, respectively. The shape of the GAMM-fitted embedding in the SA–MR plane (Fig. 2b, bottom) suggests that regions near the modulation pole of the MR axis are strongly stratified with respect to the SA axis, underscoring the distinct roles of these transmodal hierarchies. Despite the qualitative similarity in their global development, the decline in the range of the SA axis after mid-adulthood is more pronounced than that of the MR axis, suggesting that the relative prominence of MR-related organization increases during adulthood and ageing.

The VS axis undergoes distinct global development, compared with SA and MR: it expands rapidly in infancy and early childhood and contracts thereafter. Because the VS poles are spatially contiguous, decreasing VS range alongside increasing SA and MR range after late childhood is consistent with a transition from a locally organized, modality-dominated architecture to a more globally integrated FC organization[20]. Further discussions of methodological factors that might influence developmental gradient ordering, and of discrepancies with previous age-binned reports, are provided in Supplementary Discussion.

Our analysis of global gradient metrics yielded convergent evidence for these lifespan changes in hierarchical FC architecture. A monotonic increase in dispersion from infancy into early adolescence (Fig. 3b) indicates increasing global differentiation of FC profiles with respect to the SA, VS and MR axes, whereas declining dispersion thereafter reflects age-related dedifferentiation. These normative trajectories provide a compact reference for interpreting lifespan changes in segregation and integration (see Supplementary Discussion for an extended interpretation of dispersion alongside axis-specific range).

To anchor these developmental patterns in the underlying biology, we related adult cortical gene expression (AHBA, abagen) to age-varying functional gradients using a supervised PLS approach. Because AHBA expression is derived from adult post-mortem donors, these PLS components should be interpreted as adult-like spatiomolecular axes whose coupling to functional gradients changes with age, rather than as developmental trajectories of gene expression. Across axes, the first PLS component defined a smooth 'molecular axis' whose coupling to

functional gradients was generally strongest early in life and weaker in later adulthood, consistent with a transient genetic scaffold for early functional differentiation. This scaffold was axis-specific: SA was most consistently enriched for synaptic signalling and vesicle-related processes; VS implicated transcriptional, RNA-metabolic and transport-related programs with a visual emphasis; and MR implicated ionic homeostasis and excitability-related themes. Together, these results support the view that adult-like spatiomolecular axes provide a strong early-life scaffold for gradient architecture that becomes progressively less constraining as functional networks mature and diversify.

Our analysis of gradients of cortical morphology and microstructure further revealed distinct trajectories for structure–function alignment across the SA, VS and MR axes. Structure–function coupling decreased with respect to all three axes across the lifespan, with the most marked reduction seen for the transmodal SA and MR axes. By contrast, the decrease in similarity between the VS axis and its structural counterpart was less pronounced, potentially reflecting persistent microstructural differentiation between visual and somatosensory systems. One interpretation is that high structure–function similarity early in life partly reflects underdeveloped functional gradients that are initially tethered to coarse microstructural variation present at birth, whereas lifelong decreases in tethering suggest that mature functional hierarchies become increasingly driven by distributed patterns of functional coactivation that are not captured by corresponding microstructural gradients.

In summary, this study provides a comprehensive normative chart of functional-gradient development from infancy to old age. By demonstrating behavioural relevance and convergent validation across scales, we validate the importance of these organizing principles and provide a foundational reference for future studies to identify how deviations from normative trajectories might contribute to neurodevelopmental and neurodegenerative disorders.

### Limitations and future directions

It is important to note that there are many degrees of freedom in our methodology, and that the present work is fundamentally exploratory. The use of a template gradient set and alignment to that template was crucial for standardizing gradient architecture across the lifespan, but it also implies that some global organizational features persist from birth to old age. A global lifespan template might be biased towards periods of stable FC gradients; thus, our early-life results should be interpreted mainly as convergence to an adult-referenced coordinate system. Complementary approaches (for example, age-specific templates or embeddings) could help to characterize infant-native organizing principles more directly (see Supplementary Discussion).

Our lifespan sample is mainly cross-sectional; this captures population-level trends but cannot track within-individual change over time, and large-scale longitudinal studies will be essential to fully delineate these trajectories. Furthermore, our focus has been on population-level effects, and how these normative principles manifest alongside sharp functional boundaries in individuals remains a crucial area for future research. Finally, although we reveal links to cognition and multiscale biology, the mechanisms that drive these lifelong changes—from synaptic plasticity to metabolic constraints—remain important targets for future work.

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

# Methods

## Materials

The individuals in the study were selected from five imaging datasets: the BCP[49], the development, young-adult and ageing cohorts of the Human Connectome Project (HCP-D, HCP-YA and HCP-A, respectively)[50] and the Healthy Brain Network (HBN) dataset[51]. Individuals in the BCP ranged from 16 days to 6 years old, with 557 individuals scanned at 1,095 unique time points before gradient-based quality control. We initially used 652 individuals (age 5.6–21.9 years) from HCP-D, 1,206 individuals (age 22–37 years) from HCP-YA, 725 individuals (age 36–100 years) from HCP-A and 772 individuals from the HBN (age 5.6–21.9 years). Across all five cohorts, our dataset included 3,912 individuals.

## Ethics

All data were obtained from publicly available studies (BCP, HCP-D, HCP-YA, HCP-A and HBN). Each contributing study received ethics approval from its local institutional review board and obtained informed consent (and assent where appropriate). The present work analysed de-identified data and complied with all relevant data use agreements.

## Quality control

We excluded infant data with excessive motion, cutting the BCP dataset from 557 individuals with 1,095 time points to 343 individuals with 760 time points. For the entire lifespan dataset, we used visual and clustering-based gradient quality control, excluding problematic gradients from the main analysis. Using this procedure, we excluded one additional data point from the BCP for a total of 759 time points and 343 individuals (158 male individuals, 176 female individuals and 9 unreported). We excluded 2 sets of gradients from the HCP-D for a total of 650 individuals (301 male individuals and 349 female individuals). We excluded 2 individuals from the HBN dataset for a total of 770 individuals (458 male individuals, 275 female individuals and 37 unreported). Preprocessing-related failures (for example, unsuccessful surface reconstruction or unusable resting-state fMRI data) led to the exclusion of 88 HCP-YA participants. Visual inspection and *k*-means clustering of Procrustes-aligned gradients identified an additional 50 corrupted gradient sets, yielding a final HCP-YA sample of 1,068 participants (482 male and 586 female). We excluded no data from the HCP-A (725 individuals; 319 male and 406 female). These decisions were justified by examining individual gradients as well as their associated metrics (dispersion, range and similarity to template) in combination with *k*-means clustering applied to the set of all individual gradients. In total, our final analysis included 3,972 unique gradient sets from 3,556 individuals. Supplementary Fig. 3 shows the distribution of participants' ages by cohort after quality control. Cohort-wise median mean Power's frame-wise displacement (FD) after quality control is shown in Supplementary Table 3.

## fMRI preprocessing

We used an rs-fMRI preprocessing pipeline that is consistent with the HCP[52]. The preprocessing pipeline includes (1) head motion correction using FMRIB Software Library (FSL) mcflirt; (2) echo-planar imaging (EPI) distortion correction using FSL topup to generate distortion correction deformation fields using a pair of reverse phase-encoded field maps (that is, anterior–posterior and posterior–anterior or left–right and right–left); (3) rigid (six degrees of freedom) registration of a single-band reference (SBref) image to field maps; (4) rigid boundary-based registration (BBR) of distortion correcting the SBref image to the corresponding T1-weighted (T1w) images, with pre-alignment using a mutual information cost function; and (5) one-step sampling using combined deformation fields and translation matrices, producing motion- and distortion-corrected fMRI data in each individual's native space (T1w space).

We denoised the fMRI data before further analysis. First, we detrended the data by high-pass filtering with a cut-off frequency of 0.001 Hz to remove slow signal drift. Next, we used independent component analysis based automatic removal of motion artefacts (ICA-AROMA) denoising[53] to remove any residual motion artefacts. This involved performing a 150-component ICA on the fMRI data and classifying each component as either BOLD signal or artefact on the basis of high-frequency contents, correlation with realignment parameters from head motion correction, edge effects and cerebrospinal fluid fractions. Independent components classified as artefacts are (aggressively) removed by regression.

## Structural and diffusion preprocessing

Structural data underwent automated quality assessment[54,55], inhomogeneity correction and linear transformation of T2w images to the respective T1w images. We then delineated white matter, grey matter and the cerebrospinal fluid with automated segmentation (Supplementary Figs. 14 and 15). Usable diffusion data selected using a deep-learning-based semi-automated QC[56] were corrected for signal drop-out, pile-up and eddy-current and susceptibility-induced distortions[57]. Co-registration between diffusion MRI (dMRI) and structural MRI (sMRI) using multi-channel nonlinear registration was performed with the symmetric normalization method available in Advanced Normalization Tools (ANTs)[58]. Finally, manual visual inspections were done to ensure quality.

To establish a consistent coordinate system across the entire lifespan, we used in-house lifespan surface atlases extended from a previous study[59]. The reconstructed cortical surfaces of each individual were first registered to age-matched cortical surface atlases for vertex-wise correspondence among all the surfaces across the lifespan (Supplementary Fig. 14). Cortical surfaces were mapped onto a unit sphere and Spherical Demons[60] was used to perform registration between the spherical surfaces of the individuals and the age-matched surface atlases. The vertex-wise correspondence established as a result of spherical registration was then propagated to the white and pial surfaces using one-to-one correspondence between the spherical and standard representations of the white and pial surfaces. Cortical surfaces used in the analysis comprised 20,484 vertices including the medial wall and 18,644 vertices excluding the medial wall.

Structural and diffusion indices were computed in the same manner for all cohorts. Specifically, cortical thickness is defined as the Euclidean distance between two correspondent vertices on the white and pial surfaces and myelination is quantified using the ratio between T1w and T2w images[41]. Spherical mean spectrum imaging (SMSI)[61,62] was used to calculate intracellular volume fraction, extracellular volume fraction, intrasoma volume fraction, microscopic fractional anisotropy, microscopic mean diffusivity, microscopic anisotropy index and orientation coherence index. The diffusion tensor model was used to calculate fractional anisotropy and mean diffusivity.

## Functional gradients

FC matrices were computed from the vertex-wise fMRI time series for each acquisition from each individual using the pairwise Pearson correlation coefficient. For individuals with repeated scans (BCP participants), acquisitions were grouped by time point, and we averaged FC matrices within each time point. We then subjected each mean FC matrix to row-wise thresholding (top 10% of connections retained), and computed the normalized angle matrix, in keeping with previous gradient literature[1,14,63,64]. Mean FC degree was computed for each individual as the average across vertices of the sum FC weights in the individual's thresholded mean FC matrix. We computed ten FC gradients for each individual using the diffusion embedding implementation in BrainSpace[64], with the normalized angle matrix as the input affinity matrix. To align individual-specific gradient axes consistently across the lifespan, we computed a set of 'template gradients' through weighted principal

component analysis (WPCA) of all individuals' gradients. We began by applying a controlled nonlinear transformation to each individual's age, taking the square root of age to capture rapid changes in early life. The transformed ages were partitioned into ten equally spaced bins on the transformed scale and then mapped back to the original age units, yielding narrower bins (and thus finer sampling) for younger individuals. Each bin was assigned an equal total weight, and that weight was uniformly distributed among the individuals within the bin. After normalizing these weights to sum to 1 across all individuals, we constructed a single matrix of dimension $(N_{ind} \times N_{grad}) \times N_{vert}$ by stacking each individual's set of gradient maps (that is, $N_{grad}$ per individual). We then standardized each vertex column (z-scoring across all individual–gradient observations) before performing WPCA. In WPCA, each row (representing one individual–gradient combination) was assigned its individual-specific weight, ensuring that less densely sampled or younger age bins had a comparable overall influence relative to bins with many individuals. We extracted the top principal axes of variation (the first ten), and used the first three for subsequent analyses. This weighting strategy mitigates sampling imbalances and preserves important developmental transitions, ensuring that the template gradients capture large-scale connectivity variation equitably across the entire lifespan. For details of the validation analysis of our WPCA strategy for reducing age-related sampling bias in our gradient template, see 'Bootstrap validation of WPCA gradient template' in Supplementary Methods (Supplementary Fig. 12). We aligned the gradient sets of all individuals to this template embedding space using one-shot Procrustes alignment[64] without scaling or mean centring, after which gradients of the same order were correspondent across all individuals.

### Structural gradients and coupling

We also computed cortical gradients based on microstructural features to assess structure–function coupling across the lifespan by applying diffusion embedding to morphometric similarity networks (MSNs)[65]. Our structural features included 11 measures: cortical thickness, myelination, intracellular volume fraction, extracellular volume fraction, intrasoma volume fraction, fractional anisotropy, mean diffusivity, microscopic fractional anisotropy, microscopic mean diffusivity, microscopic anisotropy index and orientation coherence index. For each individual, we constructed feature vectors containing these features at each cortical surface vertex, and computed a structural affinity matrix using pairwise Pearson correlation between vectors of z-scored structural features. After setting negative correlations to zero, we then used this affinity matrix as input to the diffusion map embedding algorithm to obtain structural gradients. To compare structural and functional gradients across the lifespan, we aligned each individual's structural gradients to that individual's functional gradients, which were previously aligned to the functional template gradients comprising the SA, VS and MR axes. All subsequent comparisons between structural and functional gradients were based on these aligned structural gradients.

We further assessed structure–function coupling within canonical resting-state networks defined by the Schaefer 7-network parcellation. For each network and axis, we quantified coupling as the cosine similarity between the individual's aligned functional-gradient values and the corresponding aligned structural-gradient values across vertices belonging to that network, and modelled age effects using GAMMs (Supplementary Fig. 10a). To characterize network-level shifts in the structural gradients themselves, we computed the within-network centroid (mean) of the aligned structural-gradient values for each axis and fitted GAMMs to these trajectories versus age (Supplementary Fig. 10b).

To examine how cortical microstructure relates to the principal functional hierarchy, we quantified individual-level coupling between the SA axis and multiple microstructural indices. For each individual, vertex-wise microstructural maps were correlated with that individual's SA axis values using Pearson correlation across cortical vertices,

excluding the medial wall. This yielded one SA–metric coupling score per individual and metric. Developmental trajectories of SA–metric coupling were then modelled as a function of age using GAMMs with a random intercept for cohort, as described below.

### Gradient measures

To fully characterize how gradient architecture evolves throughout the human lifespan, we use and define a number of quantitative measures of their properties. The Euclidean distance between vertices in a diffusion embedding approximates the geodesic distance in the underlying graph from which it was computed. Thus, the Euclidean distance between vertices in our gradient embedding space approximates connectivity differentiation between those vertices. To measure the global degree of FC differentiation in an individual's FC diffusion embedding, we define gradient dispersion as the mean Euclidean distance between the embedding coordinate of all vertices and the centroid of the embedding. For a diffusion embedding with embedding coordinates $\xi_i = (x_i, y_i, z_i)$ for each vertex $i$, where $x$, $y$ and $z$ refer to the SA, VS and MR gradient values, respectively, we define dispersion among $N_{vert}$ vertices as $\sum_{i=1}^{N_{vert}} \| \xi_i - \bar{\xi} \|_2 / N_{vert}$. Within-network dispersion was computed analogously for constituent vertices of each of the seven resting-state networks (RSNs)[40].

To characterize the degree of FC differentiation along each gradient axis individually, we compute gradient range. We compute gradient range for each individual and each gradient using the inter-vigintile range (5th percentile to 95th percentile). By using robust statistical measures, we reduce the influence of noise in FC gradients on these global estimates. Gradient range quantifies the degree to which FC is differentiated along a particular gradient axis. Notably, gradient range is highly correlated with gradient variance (median absolute deviation of gradient value) across individuals ($\rho = 0.91, 0.86, 0.89$ for the SA, VS and MR axes, respectively), indicating consistent range in the diffusion embedding axes before scaling by eigenvalue.

The topographic alignment between individual gradients and our template gradient was of high interest, allowing us to assess both the quality of the Procrustes alignment procedure and the lifespan changes in gradient topography. To assess this similarity, we compute the cosine similarity between individual gradients and the corresponding group-level template gradient.

We also analysed between-network interactions in our gradient embedding space to probe the lifespan trajectories of inter-network coupling. To this end, we devised a principled measure of network–network embedding distance to quantify the degree of FC similarity between different RSNs. Specifically, between-network embedding distances shown in Extended Data Fig. 4 and described in 'Network interactions' in Supplementary Results were computed as the mean Euclidean distance between all possible vertex pairs from two networks normalized by gradient dispersion (between-network embedding centroid distances are shown in Supplementary Fig. 9). Network pairs that are highly segregated in the embedding space correspond to large between-network distances, whereas pairs that are integrated or share similar connectivity profiles have low between-network distances.

### Lifespan trajectories

Brain development throughout the lifespan is nonlinear. Furthermore, portions of our dataset use a longitudinal staggered cohort study design. To robustly estimate lifespan trajectories of cortical gradients and gradient-derived metrics, we used GAMMs[66]. This framework is well-suited for neurodevelopmental data because it can flexibly capture nonlinear relationships and incorporate hierarchical random effects, providing stable and biologically informed curve fitting across the wide age range used in this study.

We performed GAMM fitting and lifespan harmonization of gradient values at the vertex level to unify our analysis and streamline downstream computation and modelling of gradient-derived metrics. For

each cortical surface vertex, we modelled the gradient value as a smooth function of age, implemented using penalized cubic regression splines and restricted maximum likelihood (REML) for smoothing parameter estimation. Biological changes are more rapid early in life; to allow for this, the age variable was transformed by raising age in years to a fractional power, $\alpha = 0.5$, acting as a monotonic re-parameterization to increase temporal resolution during early development while maintaining model stability. Individual ID and study cohort were included as random effects to account for repeated measures within individuals in the BCP data and to control for between-cohort differences in data acquisition or processing.

To determine the optimal spline complexity of our GAMMs, we evaluated models with a range of $k$-values. A relatively low $k$ ($k = 4$) minimized overfitting and provided stable fits with the majority of cortical surface vertices. We confirmed fit quality using adjusted $R^2$ values and visual inspection of residuals. After fitting initial GAMMs to gradient value with this low complexity, we extracted and removed the random intercepts associated with individual ID and study cohort effects. This procedure harmonizes the mean level in gradient value across cohorts, producing mean-adjusted data in which site- or cohort-related biases are reduced and biologically meaningful variation is preserved.

To address any non-uniformity in the age and cohort distributions of our dataset, we applied a density-based weighting strategy before fitting our final GAMMs to the vertex-wise gradient values. We divided the full transformed age range into equal-sized bins and assigned weights to each data point inversely proportional to the bin-specific sample size, ensuring that each age bin contributes equally to our regression and mitigating temporal inhomogeneity. We further scaled these weights by the reciprocal of the cohort sample size to avoid undue influence of large cohorts.

To ensure that our fitted trajectories for gradient values reflect not only the mean trend but also stable variance structure, we controlled for heteroscedasticity using a two-step procedure. First, after mean harmonization using the cohort random effect intercepts of the low-$k$ GAMM, we computed vertex-wise residuals and their squares. A second GAMM with low basis complexity and cohort random effects was fitted to these squared residuals. The prediction of this variance model estimated how residuals change with age, identifying potential age-dependent heteroscedasticity. For each cohort, scaling the uncorrected residuals to match the predicted variance produced variance-adjusted data, preserving biologically meaningful variability across age while controlling for spurious cohort effects[67]. Finally, we refitted a GAMM with higher basis complexity (for example, $k = 10$) to the variance-adjusted data using weights equal to the inverse of the predicted variance at each point, reducing the effects of heteroscedasticity on the final model fits.

After harmonization, individual gradient values were mean-corrected for cohort differences and variance-adjusted to address heteroscedasticity. Supplementary Figs. 4 and 5 show diagnostic plots for exemplar vertices, illustrating gradient values and residual distributions before and after the harmonization procedure. Supplementary Fig. 4 exemplifies one of the most significant fits, and Supplementary Fig. 5 one of the least significant fits. Supplementary Fig. 7 shows surface maps of effective degrees of freedom values for the final weighted fits for all three gradients. After gradient values were effectively harmonized, analysis of subsequent gradient-derived metrics required no further harmonization. For metrics including gradient dispersion, gradient range and several network-parcellation-based gradient metrics, GAMMs with no random effects with basis complexities determined by optimal adjusted $R^2$ and Akaike information criterion (AIC) values were fitted with age treated as a smooth term. Supplementary Fig. 8 shows the vertex-wise population sample variance of the aligned SA, VS and MR gradients across the full dataset.

To account for the potential confound introduced by arousal state in the BCP data, we divided participants into two cohorts for our GAMM analysis. Specifically, all children aged three years old or younger were classified as the 'sleep cohort' because they were scanned during natural sleep. Participants older than three years formed the 'wake cohort', given that they were scanned either while they were asleep or during a passive movie-watching paradigm[49]. By stratifying the sample in this manner, we aimed to minimize any systematic bias arising from differences in arousal state across these age ranges.

To investigate the extent to which single cohorts could influence our vertex-wise gradient GAMM fits, we performed a leave-one-cohort-out analysis (see 'Leave-one-cohort-out (LOCO) stability analysis' in Supplementary Methods and Supplementary Fig. 11). This analysis indicated that no single cohort disproportionately influenced the developmental trajectories, and that the harmonized GAMM fits generalize well to held-out data.

To evaluate the presence of nonlinear age effects at each vertex, we compared a linear model to a GAM for each vertex using the harmonized gradient values. First, for a given vertex $v$, we fitted a linear model of the form $v = \beta_0 + \beta_1\sqrt{\text{age}} + \varepsilon$, reflecting the hypothesized monotonic but potentially non-uniform relationship between age and the response. Next, we fitted a GAM of the form $v = \beta_0 + s(\sqrt{\text{age}}; k = 6) + \varepsilon$, where $s(\cdot)$ represents an isotropic smooth function with spline basis dimension $k = 6$. The GAM was estimated using REML in the mgcv package. Because the linear and GAM formulations are nested (the linear model is a special case of the smoother with effectively one degree of freedom), we performed a partial $F$-test comparing the two models: LM: $v = \beta_0 + \beta_1\sqrt{\text{age}} + \varepsilon$ versus GAM: $v = \beta_0 + s(\sqrt{\text{age}}; k = 6) + \varepsilon$. This yields a $P$ value ($P_{\text{nonlinear}}$) indicating whether the smoother explains significantly more variance than the linear term alone. We repeated this procedure at each vertex, thus obtaining one $P_{\text{nonlinear}}$ per vertex. We then applied an FDR correction across all vertices' $P$ values to control for multiple comparisons. Finally, we defined the proportion of significantly nonlinear vertices as the fraction of vertices for which the FDR-corrected $P_{\text{nonlinear}}$ was below 0.05. This proportion summarizes the extent to which the age relationship exhibits significant departures from linearity across the cortical surface. We found significantly nonlinear relationships between harmonized gradient value and age for 98% of vertices for the SA axis, 99% of vertices for the VS axis and 97% of vertices for the MR axis.

We investigated whether males and females exhibited systematically different lifespan trajectories in their global gradient metrics using a two-step GAMM approach. First, to establish a population-level curve, we fitted a GAMM to each metric as a function of age (square-root-transformed to account for rapid early-life changes) while ignoring sex. Specifically, metric = $s(\sqrt{\text{age}}$, k = 4, bs = "cs"), yielding an overall (sex-agnostic) trajectory. We then computed residuals of each individual's metric from this population-level fit. Second, to capture sex-specific deviations, we fitted a new GAMM to these residuals with a smooth interaction term by sex, residual = $s(\sqrt{\text{age}}$, k = 5, bs = "cs", by = sex), allowing separate smooth functions for males and females. This effectively modelled whether either sex exhibited systematic departures from the population mean trajectory as a function of age. We subsequently obtained sex-specific trajectories by adding each sex's fitted residual curve back to the population mean fit. We examined the significance of the male- and female-specific smooth terms through their $P$ values in the GAMM summary. If these terms were significant, it would indicate that one or both sexes diverged from the overall trajectory in a manner that could not be attributed to random variability. Conversely, non-significant terms would suggest insufficient evidence for systematically distinct lifespan trajectories across sexes (see 'Sex-by-age deviations in global gradient metrics' in Supplementary Results and Supplementary Fig. 1).

We also investigated whether total brain volume could be a significant confound in lifespan gradient trajectories. To do this, we repeated vertex-wise GAMM fitting for gradient values with total brain volume as a covariate. We then examined the coefficients for the brain volume covariate to assess its effect on the lifespan trajectory of gradient values.

Supplementary Fig. 2 shows these coefficients as surface maps, revealing consistently small coefficients across vertices and gradients. On the basis of this, we conclude that total brain volume is not a significant confound for gradient value across age.

## NIH Toolbox Cognition Battery

For the HCP-YA cohort, we investigated whether FC gradient metrics predict performance on the NIH Toolbox Cognition Battery (https://nihtoolbox.org/domain/cognition/). We analysed nine unadjusted cognitive scores: CogTotalComp_Unadj, CogFluidComp_Unadj, CogCrystalComp_Unadj, ListSort_Unadj, Flanker_Unadj, CardSort_Unadj, ProcSpeed_Unadj, ReadEng_Unadj and PicSeq_Unadj. The FC gradient metrics matched those used elsewhere in the manuscript and included global dispersion, axis-specific ranges (grange_SA, grange_VS and grange_MR), cosine similarities to the canonical gradients (cossim_SA, cossim_VS and cossim_MR), and pre-alignment eigenvalues (eval1, eval2 and eval3).

After removing two individuals with missing cognition measures, the final HCP-YA sample comprised 1,066 participants aged 22–37 years. Age and all gradient metrics were standardized (z-scored). NIH Toolbox scores were normed (mean = 100, s.d. = 15). Because each participant contributed a single scan, we fitted an ordinary least squares model for each pairing of cognitive score $y$ and gradient metric $G$:

$$y_i = \beta_0 + \beta_{\text{age}} z(\text{Age}_i) + \beta_{\text{grad}} z(G_i) + \varepsilon_i.$$

Models were estimated with R::lm. For each model we extracted the standardized slope $\beta_{\text{grad}}$, standard error, $t$-value, and raw $P$ value, and controlled the FDR across the $9 \times 10 = 90$ tests using Benjamini–Hochberg at significance level 0.05.

To test whether these relationships generalize across the lifespan, we fitted a separate GAM for each pairing of NIH Toolbox score and gradient metric, pooling HCP-D, HCP-YA and HCP-A and z-scoring cognitive outcomes and gradient metrics. We modelled a cognitive outcome $y$ and a gradient metric $G$ as

$$y_i = \beta_0 + s(\text{Age}_i) + s(z(G_i)) + \text{ti}(\text{Age}_i, z(G_i)) + \gamma_{\text{cohort}(i)} + \varepsilon_i,$$

where $s(\cdot)$ are cubic regression splines and $\text{ti}(\cdot,\cdot)$ is a tensor-product interaction term. We used mgcv::bam with fast REML, $k_{\text{age}} = 8$ and $k_{\text{grad}} = 5$. Cohort (HCP-D, HCP-YA and HCP-A) was included as a fixed effect.

From each fitted model we summarized two quantities based on the age-specific gradient contrast:

$$\Delta(a) = E[y|z(G) = q_{0.90}, \text{Age} = a] - E[y|z(G) = q_{0.10}, \text{Age} = a].$$

1. The main effect at the pooled median age

$$\Delta_{\text{main}} = \Delta(a = \text{median})$$

reflects the predicted difference in cognition (in s.d. units) between individuals at the 90th versus the 10th percentile of the gradient metric, evaluated at the pooled median age. $\Delta_{\text{main}} > 0$ indicates that higher values of the gradient metric are associated with better cognitive performance at the median age. For example, $\Delta_{\text{main}} = 0.20$ means that, at this age, individuals at the 90th percentile of the gradient metric score about 0.20 s.d. higher on cognition than do those at the 10th percentile.

2. The age modulation of the gradient effect

$$\Delta_{\text{age}} = \Delta(a = q_{0.90}^{\text{age}}) - \Delta(a = q_{0.10}^{\text{age}})$$

reflects the change in the high–low gradient contrast from the 10th to the 90th percentile of age. $\Delta_{\text{age}} > 0$ indicates that the gradient's association with cognition strengthens with age; $\Delta_{\text{age}} < 0$ indicates that it weakens; values near zero indicate an age-invariant association. Note that $\Delta_{\text{age}}$ is a change in effect size, not a slope of cognition with respect to age.

For each (cognition, gradient) model, quantiles $q$ were computed from the samples used to fit that model; cohort-specific predictions at these quantiles were then averaged using weights proportional to cohort sample sizes. In Extended Data Fig. 6a, asterisks mark FDR-significant gradient main smooths $s(z(G))$; asterisks in Extended Data Fig. 6b mark FDR-significant age × gradient interactions $\text{ti}(\text{Age}, z(G))$.

## Mullen scales of early learning

For the BCP cohort, we tested whether the same FC gradient metrics explained variance in the Mullen scales of early learning. We analysed five raw sub-scores (Gross_raw, Fine_raw, Vis_raw, Rec_raw and Express_raw), as well as a composite score, Mullen_sum, computed as the sum of the four non-motor domain scores. The same gradient metrics used in the HCP-YA analysis were examined: global dispersion, grange_*, cossim_* and eval*.

After quality control, the final dataset included 453 data points from 239 infants aged 0.23–4.87 years. Age, gradient metrics and Mullen scores were all standardized (z-scored). Because some infants contributed multiple time points, we used a linear mixed-effects model with a random intercept for each individual to model the relationship between Mullen score $y$ and gradient metric $G$:

$$y_{ij} = \beta_0 + \beta_{\text{age}} z(\text{Age}_{ij}) + \beta_{\text{grad}} z(G_{ij}) + b_{\text{ind}(j)} + \varepsilon_{ij}, \tag{1}$$

where $b_{\text{ind}(j)} \sim N(0, \sigma_b^2)$. Models were fitted using lme4::lmer, and lmerTest was used to obtain Satterthwaite degrees of freedom and $P$ values. For each model, we extracted $\beta_{\text{grad}}$, its standard error, $t$-statistic and raw $P$ value. To correct for multiple comparisons across the $6 \times 10 = 60$ tests, we applied the Benjamini–Hochberg procedure to control the FDR at significance level 0.05. Full results are reported in Supplementary Table 1.

## Neurosynth meta-analytic validation

Twenty-four Neurosynth[45] term maps were parcellated according to the Schaefer-400 atlas and stacked into a parcel × term matrix $X \in \mathbb{R}^{400 \times 24}$. For each term $t$, parcel-wise values were z-scored across parcels to yield $Z$. A canonical association–unimodal axis $A$ was defined as

$$A_p = z\left( \frac{1}{|T_{\text{assoc}}|} \sum_{t \in T_{\text{assoc}}} Z_{p,t} - \frac{1}{|T_{\text{unimodal}}|} \sum_{t \in T_{\text{unimodal}}} Z_{p,t} \right), \tag{2}$$

with association terms including theory of mind, working memory, language, default mode and so on, and unimodal terms including visual, somatosensory, tactile and motor; attention, eye movements and pain were excluded as intermediates.

For individual-level validation, we computed similarity as the parcel-wise Spearman correlation between each individual's sign-aligned SA gradient and $A$. Similarity was then modelled as a smooth function of age using GAMM, and the fitted trajectory is shown with a 95% standard-error band (Extended Data Fig. 7c).

To quantify term–axis relationships, we computed Spearman correlations between each standardized term map $Z_{\cdot,t}$ and each individual's parcellated gradient axis map (SA, VS and MR) and summarized these individual-level term–axis correlations as box-and-whisker plots (Extended Data Fig. 7d–f). Boxes denote the interquartile range, with the median indicated, and whiskers denote the central percentile range. Terms are ordered by median alignment. All statistics were performed at the parcel level.

## Transcriptomic enrichment

We used the AHBA[46] microarray data processed with abagen[47] and parcellated according to the Schaefer-400 atlas (7-network, 2 mm, MNI152). Following abagen's standard workflow unless noted otherwise,

probes were reannotated and reduced by differential stability; samples were scaled-robust-sigmoid (SRS) normalized within donor and subsequently $z$-scored across samples; tissue samples were assigned to Schaefer parcels in MNI space and aggregated per donor; donors were then averaged to yield a parcels × gene expressions matrix $X$. Genes that were entirely missing within the retained parcels were dropped, and the remaining missing values were median-imputed per gene (no imputation was applied to the phenotypes).

For each target age, we formed a parcel-wise phenotype vector $\mathbf{y}$ comprising the mean gradient value at that age. Before modelling, both $X$ (each gene column) and $\mathbf{y}$ were standardized across parcels (mean = 0, s.d. = 1). We then fitted PLS regression with one component (PLS1; scikit-learn), obtaining the parcel scores $\mathbf{t} = \mathbf{Xw}$ and gene weights $\mathbf{w}$. To ensure interpretability, the component was oriented so that corr($\mathbf{t}$, $\mathbf{y}$) ≥ 0 at every age (that is, positive scores indicate parcels whose multigene expression profile aligns with a higher mean gradient). Model performance was summarized by the in-sample correlation corr($\mathbf{t}$, $\mathbf{y}$) and by permutation tests in which parcel labels of $\mathbf{y}$ were randomly shuffled $n$ times ($n$ = 5,000) to generate a null for corr($\mathbf{t}$, $\mathbf{y}$). Permutation $P$ values were adjusted across age using Benjamini–Hochberg FDR. To visualize these null distributions, for each gradient axis and age we refitted the PLS1 model after shuffling parcel labels 2,000 times and plotted the resulting absolute correlations as box plots, with the empirical (unshuffled) correlation overlaid as a point (Supplementary Fig. 13).

To interpret the molecular axis, we computed per-gene signed coefficients (standardized regression coefficients) and ranked genes from most positive to most negative. For each age we submitted the ranked list to GOrilla[68] (GO: Biological Process, ranked-list mode). Because GO terms are redundant, we summarized enrichment into a small set of themes using prespecified keyword rules (for example, synaptic signalling; vesicle cycle; ion transport and excitability; ECM, adhesion and migration; immune and microglia; cell cycle and proliferation; mitochondria and energy). For each age and theme we reported the median $-\log_{10}(q_{\mathrm{FDR}})$ across member terms. Heat maps were used to display these values (cells at 0 indicate no significant terms under the chosen cut-off).

For map-level validation, we projected selected enriched gene sets back onto cortex. Specifically, for a GO term with gene set $G$ we computed a parcel score $s_i = \frac{1}{\| \mathbf{w}_G \|_2} \sum_{g \in G} w_g \, z(x_{ig})$, where $z(\cdot)$ denotes parcel-wise $z$-scoring and $w_g$ is the PLS1 coefficient for gene $g$ (unweighted means were also checked for robustness).

## Visualization

Figure 1 illustrates our analysis framework and aims to establish visualization conventions to facilitate fast and simple interpretations of our lifespan gradients. First, we associate a colour map with each gradient axis. For the SA, VS and MR axes, these gradient-specific colour maps span from white to red, green to blue and black to grey, respectively. To visualize the topography of all three gradients on the cortical surface simultaneously, we devised a colour-mapping scheme that combines the gradient-specific colour maps such that vertices at the association pole of the SA axis are red, vertices at the modulation pole of the MR axis are grey and the visual and somatosensory poles of the VS axis are blue and green, respectively. In Fig. 1b, we display three-dimensional embeddings of all three gradients with our three-dimensional colour map, and establish notation to indicate embedding axis orientation.

Figure 2a uses the aforementioned gradient-specific colour maps to chart topographical progression of each gradient in combination with density plots for gradient value. Note that we use a constant range with respect to time for the colour maps of surface-based gradient plots. Density plots are computed using kernel density estimation (KDE) applied to the vertex-wise GAMM prediction of gradient value at selected time points. Before density estimation, we normalize gradient values across vertices and all time points to range between

0 and 1. To showcase temporal changes in both the shape and the scale of density distribution, we use constant horizontal and vertical axis ranges across time.

In Extended Data Fig. 4 (bottom), we display mean between-network embedding distance in matrix form, in which the mean distance between each network pair is encoded by the colour of that cell.

## Reporting summary

Further information on research design is available in the Nature Portfolio Reporting Summary linked to this article.

## Data availability

The Lifespan HCP fMRI data are publicly available through the National Institute of Mental Health data archive (NDA, https://nda.nih.gov). All of the data are deposited under the Connectome Coordination Facility repository with the following collection IDs: BCP, 2848; HCP-D, 2846; HCP-YA, 2825; HCP-A, 2847). The HBN MRI data are available from https://healthybrainnetwork.org/. In addition, meta-analytic decoding resources were obtained from the Neurosynth database (v.7; abstract-based term annotations) downloaded via the NiMARE Python package (https://nimare.readthedocs.io/en/0.0.1/auto_examples/01_datasets/download_neurosynth.html) and converted to a NiMARE dataset. AHBA normalized microarray datasets (six donor brains) were downloaded from the Allen Institute portal and processed with abagen (https://abagen.readthedocs.io/en/stable/user_guide/download.html) to generate parcellated regional expression matrices. Source data are provided with this paper.

## Code availability

Custom code used to generate the results is available from the corresponding authors upon reasonable request. Gradient computation and cortical surface visualization were done using the BrainSpace toolbox (https://brainspace.readthedocs.io/). Volume-to-surface mapping of fMRI time series and computation of functional correlation matrices were done using Connectome Workbench (https://www.humanconnectome.org/software/get-connectome-workbench). Manipulation of FC matrices and computation of gradient-based metrics were done in Python using standard libraries including Numpy (https://numpy.org/), Scipy (https://scipy.org/) and scikit-learn (https://scikit-learn.org/stable/). WPCA computation of template gradients was done using the wpca Python library (https://github.com/jakevdp/wpca). Neurosynth data were downloaded and manipulated using NiMARE (https://nimare.readthedocs.io/en/0.0.1/auto_examples/01_datasets/download_neurosynth.html). AHBA normalized microarray datasets (six donor brains) were downloaded from the Allen Institute portal and processed with abagen (https://abagen.readthedocs.io/en/stable/user_guide/download.html). Manipulation of neuroimaging-specific file types was done using NiBabel (https://nipy.org/nibabel/). GAMM fitting was done with the MGCV R package (https://cran.r-project.org/web/packages/mgcv/index.html), and visualization of lifespan curves and data manipulation in R were done using the Tidyverse R libraries (https://tidyverse.org/).

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

**Acknowledgements** This work was supported in part by the US National Institutes of Health (NIH) through grants R01 MH125479, R01 EB008374, R01 EB035160, R01 NS134849 and R01 MH133836 (P.-T.Y.).

**Author contributions** H.P.T.: methodology, formal analysis, software, investigation, conceptualization, validation, visualization, writing (original draft) and writing (review and editing). K.M.H.: data curation, methodology, software and writing (review and editing). K.-H.T.: data curation, methodology, software and writing (review and editing). G.L.: methodology and software. W. Lyu: data curation. W. Lin: resources. S.A.: resources, methodology, software and writing (review and editing). P.-T.Y.: conceptualization, methodology, investigation, validation, supervision, funding acquisition and writing (review and editing).

**Competing interests** The authors declare no competing interests.

**Additional information**
**Correspondence and requests for materials** should be addressed to Hoyt Patrick Taylor or Pew-Thian Yap.

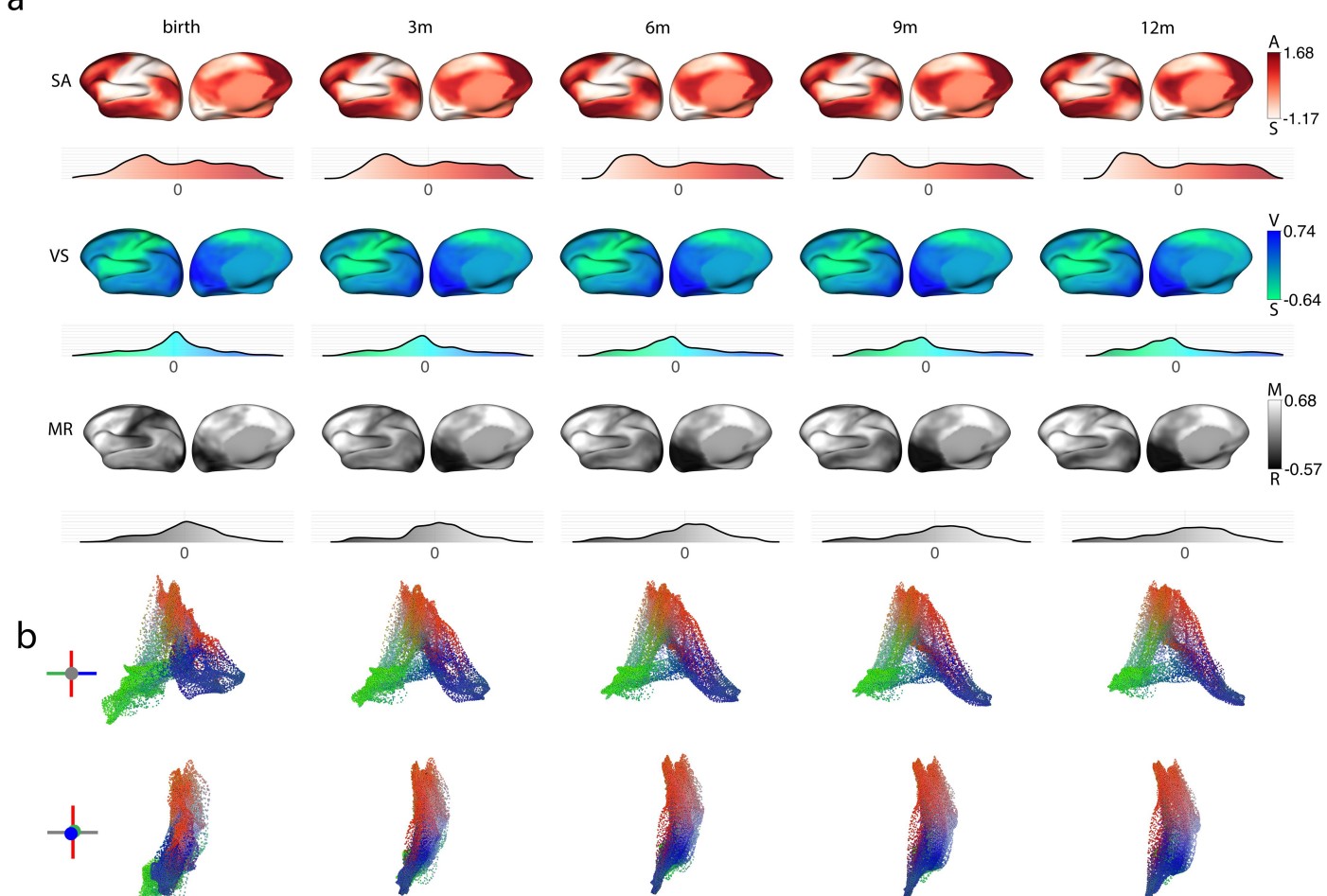

**Extended Data Fig. 1 | GAMM fitted gradients during the first year of life.**
**a**, GAMM fits of FC gradient value are plotted on the cortical surface with density plots displaying gradient value distribution at selected time points.

**b**, Embedding plots given by gradient GAMM fits in SA–VS plane (top) and SA–MR plane (bottom); in the SA–MR view, MR polarity is oriented left-to-right from representation (R) to modulation (M).

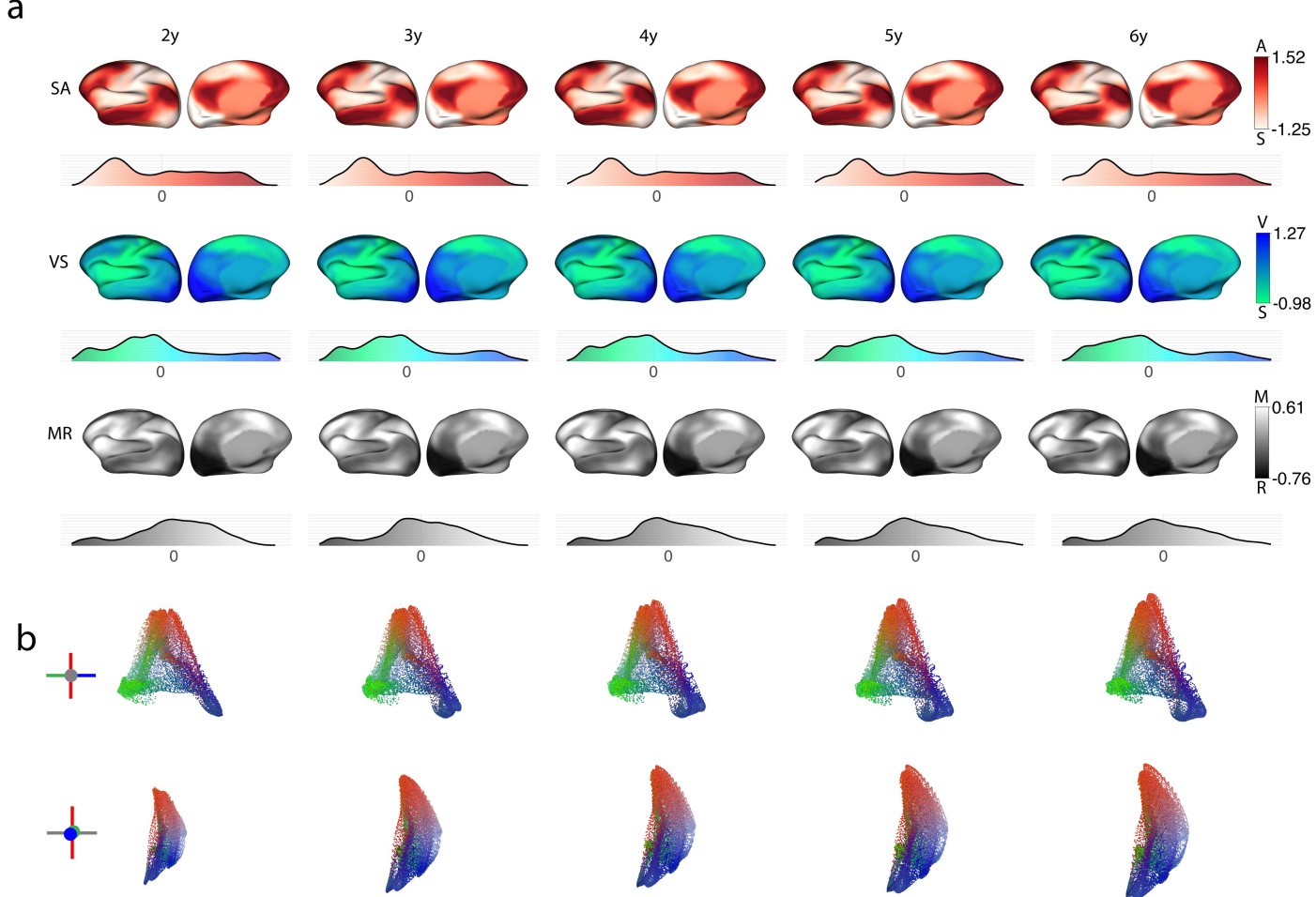

**Extended Data Fig. 2 | GAMM fitted gradients between 2 and 6 years.** **a**, GAMM fits of FC gradient value are plotted on the cortical surface with density plots displaying gradient value distribution at selected time points. **b**, Embedding plots given by gradient GAMM fits in SA–VS plane (top) and SA–MR plane (bottom); in the SA–MR view, MR polarity is oriented left-to-right from representation (R) to modulation (M).

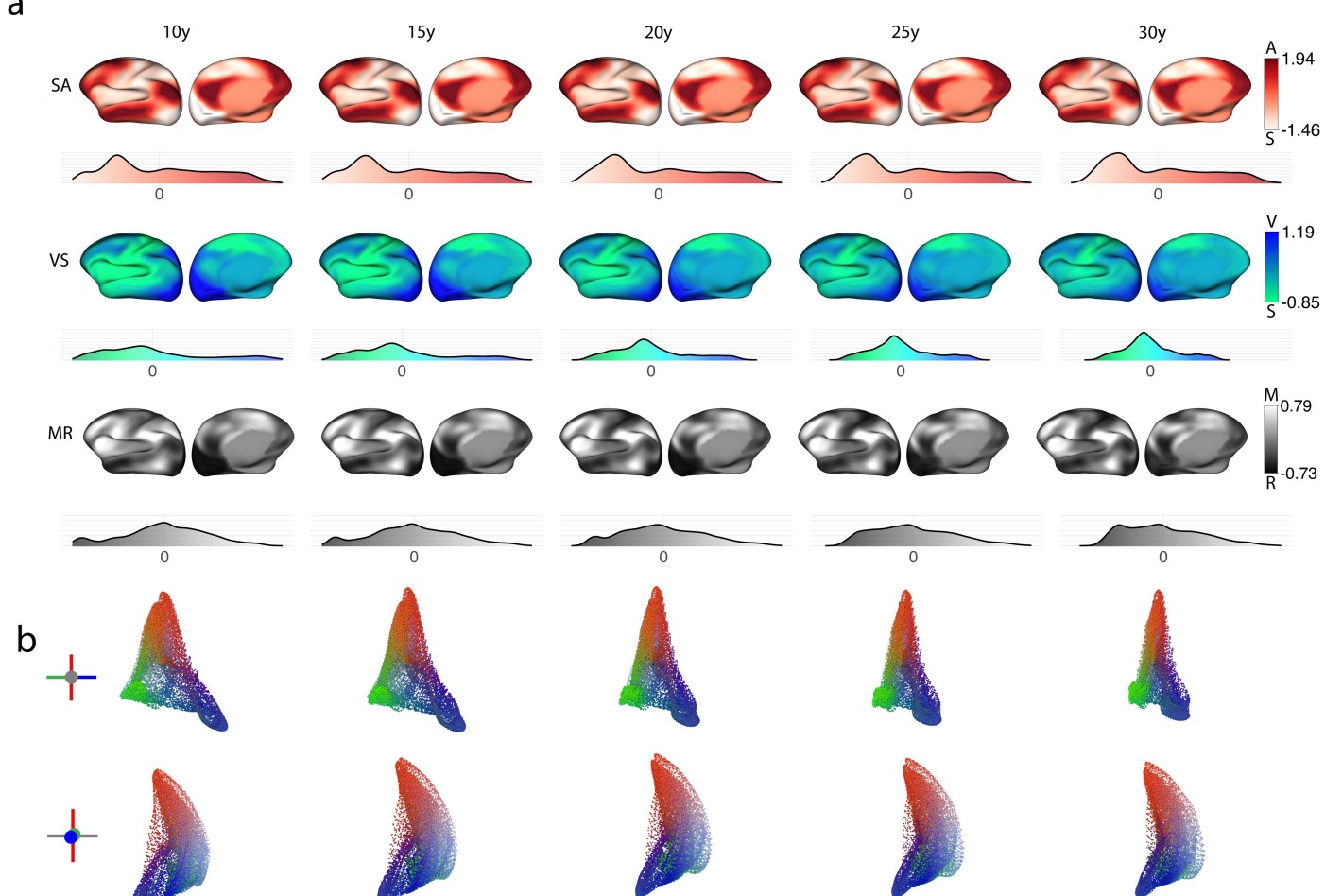

**Extended Data Fig. 3 | GAMM fitted gradients between 10 and 30 years.**
**a**, GAMM fits of FC gradient value are plotted on the cortical surface with density plots displaying gradient value distribution at selected time points.

**b**, Embedding plots given by gradient GAMM fits in SA–VS plane (top) and SA–MR plane (bottom); in the SA–MR view, MR polarity is oriented left-to-right from representation (R) to modulation (M).

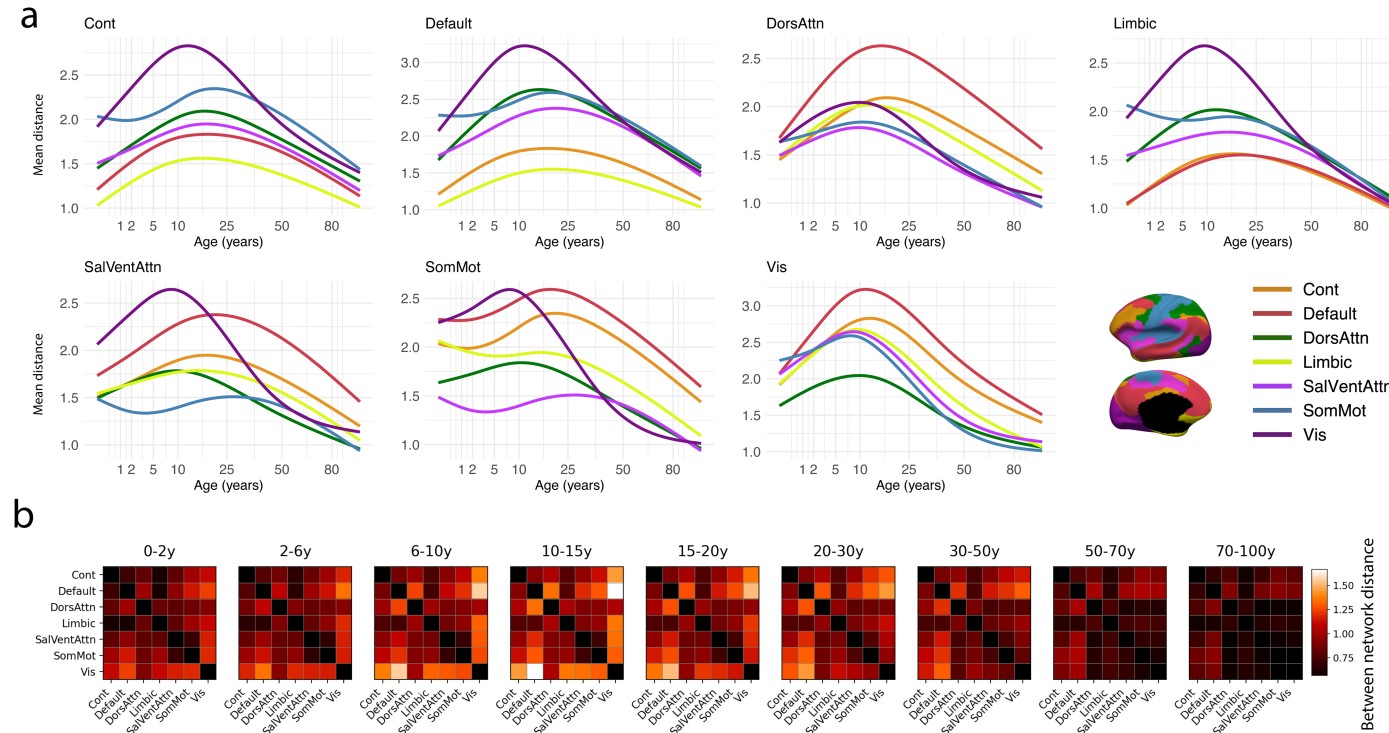

**Extended Data Fig. 4 | GAMM fits of mean gradient embedding distance between resting-state networks.** Larger distances correspond to higher levels of functional differentiation and segregation. Small distances indicate functional similarity and integration. **a**, Each plot displays the lifespan GAMM fit of the mean Euclidean distance in the SA–VS-MR gradient embedding space between vertices in a network and vertices in each other network. **b**, Matrix plots of average network–network mean embedding distance in selected temporal windows, with lighter values denoting larger distance.

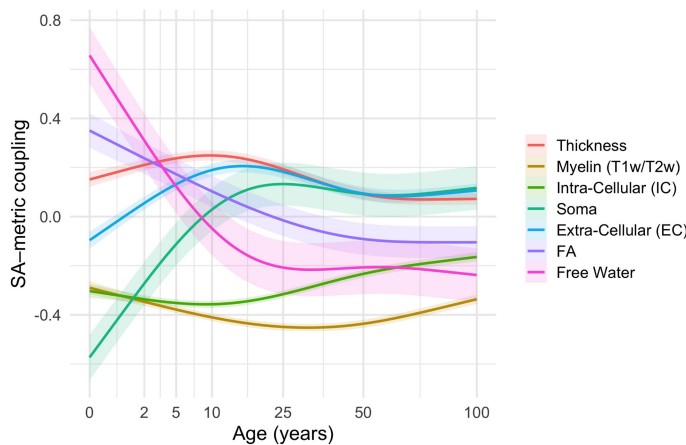

**Extended Data Fig. 5 | Lifespan trajectories of microstructural coupling to the SA axis.** Individual correlations between the SA axis and seven cortical microstructural indices (thickness, myelin [T1w/T2w], intracellular volume fraction (IC), soma, extracellular volume fraction (EC), fractional anisotropy (FA) and free water (FW)) were computed at each age and modelled with GAMMs including a cohort random effect. Curves show polarity-aligned mean fits with shaded 95% CIs.

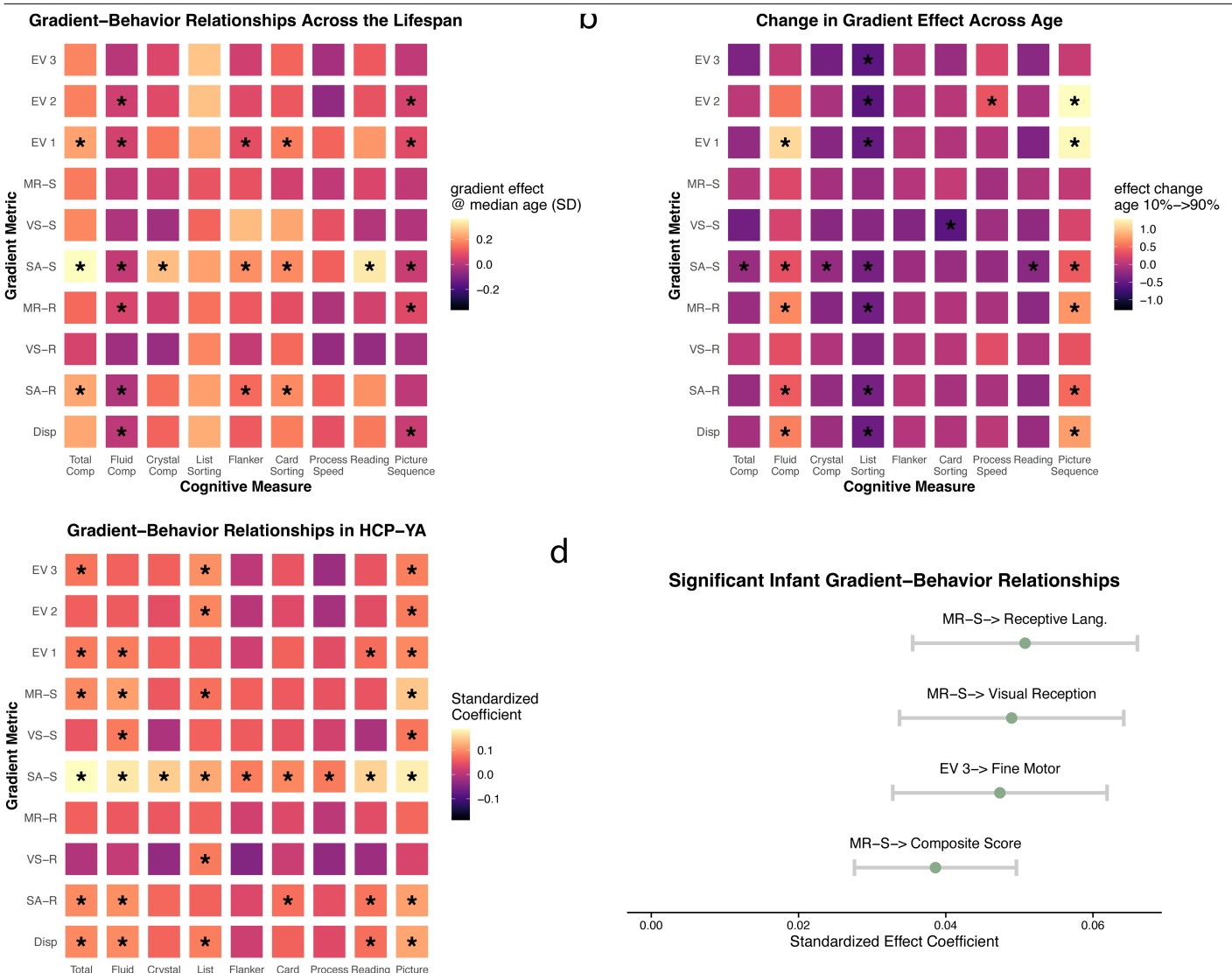

**Extended Data Fig. 6 | Functional-gradient metrics predict cognitive performance across the lifespan.** Gradient metrics (rows): EV1–EV3, alignment-agnostic gradient eigenvalue for the first three gradients; suffix -S indicates cosine similarity to the adult template, -R indicates gradient range, and Disp indicates gradient dispersion. **a**, Pooled main effect (combined HCP-D/YA/A, $n = 2,126$). Tiles show the predicted difference in cognition (z units) between the 90th vs 10th percentile of each gradient metric, evaluated at the pooled median age (30 years). Effects were estimated with tensor GAMs (mgcv::bam, fREML): cognition ~ $s(\text{age}) + s(z_{\text{grad}}) + \text{ti}(\text{age}, z_{\text{grad}}) + \text{cohort}$ (predictors and outcomes z-scored; cohort as a fixed effect). Asterisks denote the two-sided approximate $F$-test for the gradient smooth $s(z_{\text{grad}})$ after Benjamini–Hochberg FDR correction across all cognition × gradient tests ($q_{\text{FDR}} < 0.05$). **b**, Age modulation of gradient effects (combined HCP-D/YA/A) from the same GAMs.

Tiles show the change in the high–low gradient effect from the 10th to 90th percentile of age (14 and 65 years in the pooled sample), i.e., $\Delta_{65y} - \Delta_{14y}$. Asterisks mark the two-sided approximate $F$-test for the interaction $\text{ti}(\text{age}, z_{\text{grad}})$ after FDR correction across all interactions ($q_{\text{FDR}} < 0.05$). Both panels aggregate predictions across cohorts with sample-size weights. **c**, In young adults (HCP-YA, $n = 1,066$), tiles show standardized regression coefficients ($\beta$) for gradient metrics (rows) predicting nine NIH Toolbox domains (columns), adjusted for age, from standardized ordinary least squares models: cognition ~ age + gradient. Inference used two-sided $t$-tests for $\beta$ (df $= n - 3$) with Benjamini–Hochberg FDR correction across all cognition × gradient tests; asterisks indicate $q_{\text{FDR}} < 0.05$. **d**, In infants (BCP, $n = 239$ individuals, 453 samples), standardized coefficients are shown for the four Mullen-gradient associations surviving FDR correction; bars indicate ±1 s.e.

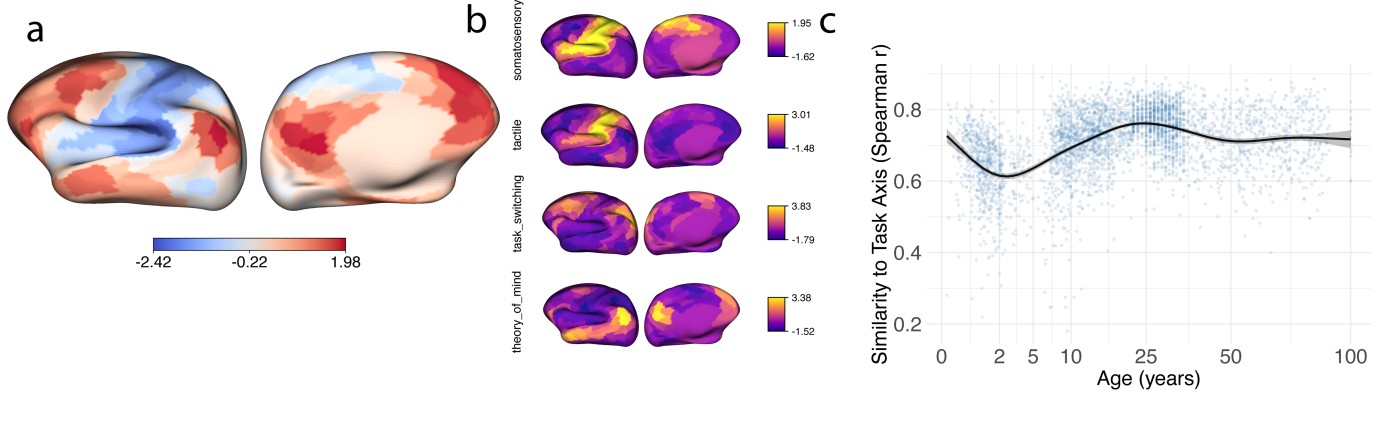

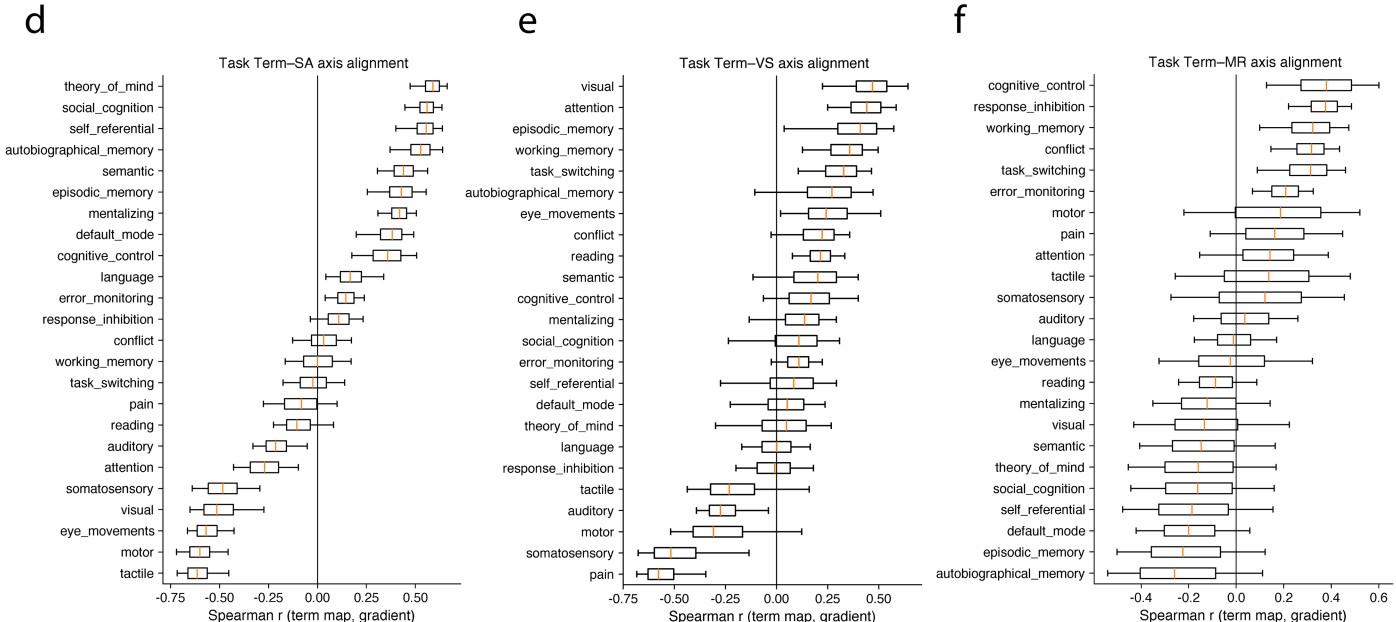

**Extended Data Fig. 7 | Neurosynth meta-analytic transmodal–unimodal axis aligns with the SA axis and consolidates across development.** **a**, Canonical axis *A* derived from 24 Neurosynth term maps parcellated with the Schaefer-400 atlas: each term map was *z*-scored across parcels and $A = z(\langle Z_{assoc}\rangle - \langle Z_{uni}\rangle)$. Warm colours indicate transmodal cortex; cool colours indicate unimodal (sensorimotor/visual) cortex. **b**, Example term *z*-maps illustrating unimodal (somatosensory, tactile) and transmodal (task switching, theory of mind) topographies. **c**, Individual-level similarity to *A* versus age (Spearman $\rho$ between *A* and each individual's SA gradient). Points denote individuals; the curve shows a GAMM fit with a 95% s.e. band. **d**–**f**, Lifespan distributions of term–axis alignment for SA (**d**), VS (**e**) and MR (**f**): for each term and axis, alignment was computed as the parcel-wise Spearman correlation between the standardized term map and each individual's axis map. Box plots show the median and interquartile range; whiskers indicate the 5th–95th percentiles; terms are ordered by median alignment. In **c**–**f**, $n = 3,972$.

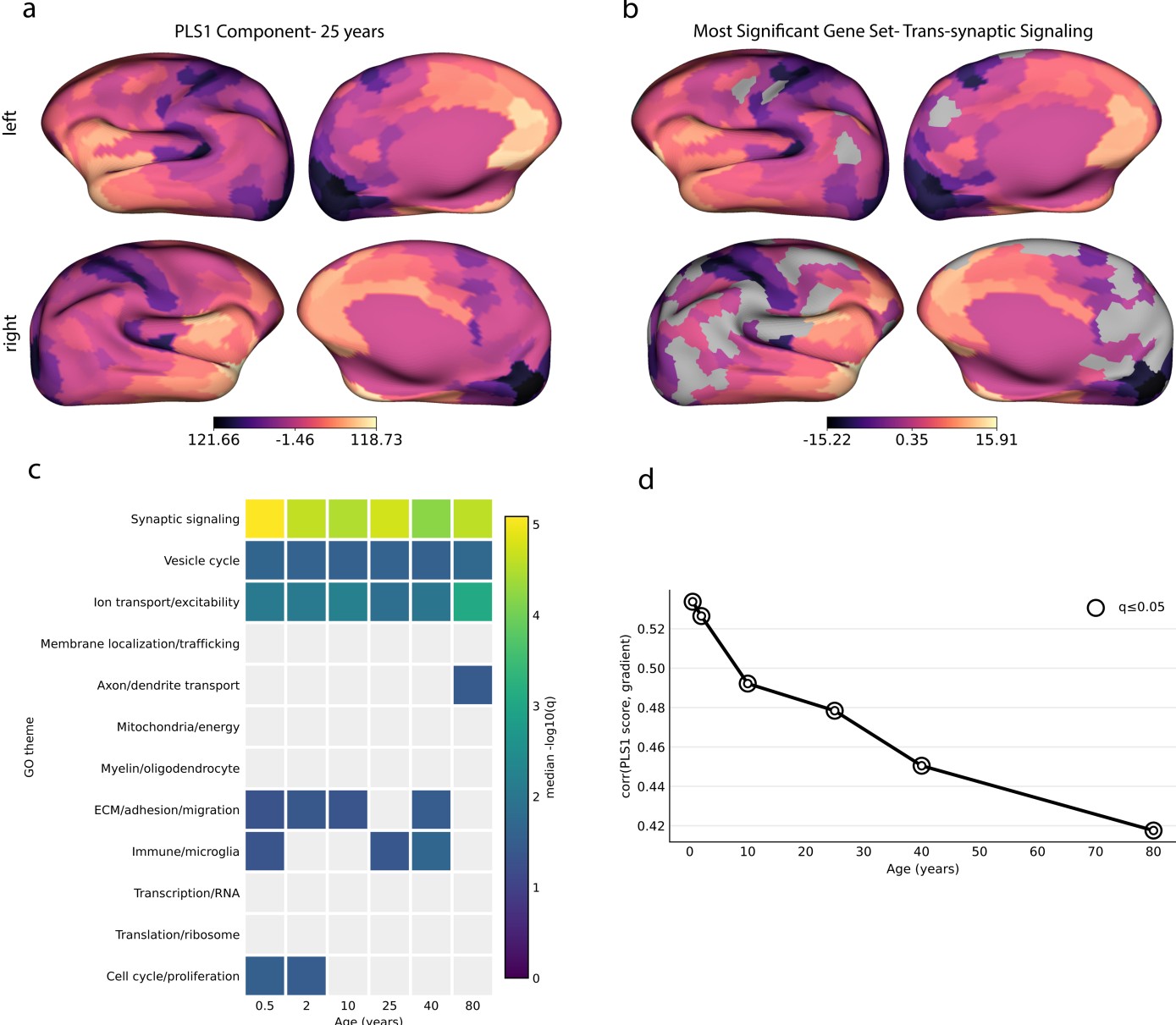

a

**PLS1 Component- 25 years**

left

right

121.66    -1.46    118.73

b

**Most Significant Gene Set- Trans-synaptic Signaling**

left

right

-15.22    0.35    15.91

c

d

**Extended Data Fig. 8 | Molecular axis predicting cortical mean SA gradient and its biological interpretation across age. a**, Parcel-wise score map at 25 years based on the first component of a partial least squares (PLS) model, relating adult human cortical gene expression (AHBA, parcellated) to the mean SA axis. The component is sign-oriented so that corr(PLS1 score, mean gradient) ≥ 0: warm colours mark parcels whose multigene expression profile predicts higher mean gradient; cool colours mark parcels aligned with lower mean gradient. **b**, Spatial projection of the top enriched GO term, trans-synaptic signalling. A parcel-wise gene-set score was computed (z-scored expression, coefficient-weighted by PLS1) and mapped to cortex. Co-localization of this map with the PLS1 map (and with the phenotype) links the molecular axis in **a** to synaptic

biology. **c**, GO themes across age. For each age, genes were ranked by signed PLS1 coefficients and analysed with GOrilla. Cells show the median −log₁₀ (FDR $q$) of terms within each theme, summarizing which processes are most consistently associated with higher mean gradient at different developmental stages (e.g., strong synaptic/vesicle signatures with age-specific modulation of ion transport and occasional extracellular matrix/immune/cell-cycle signals). **d**, Gene–gradient coupling as a function of age: the Pearson correlation between the PLS1 parcel scores and the mean gradient, with points marked when permutation $q < 0.05$. This curve indicates when adult-like molecular patterns best predict the spatial distribution of the gradient (stronger earlier, declining into later adulthood in this dataset).

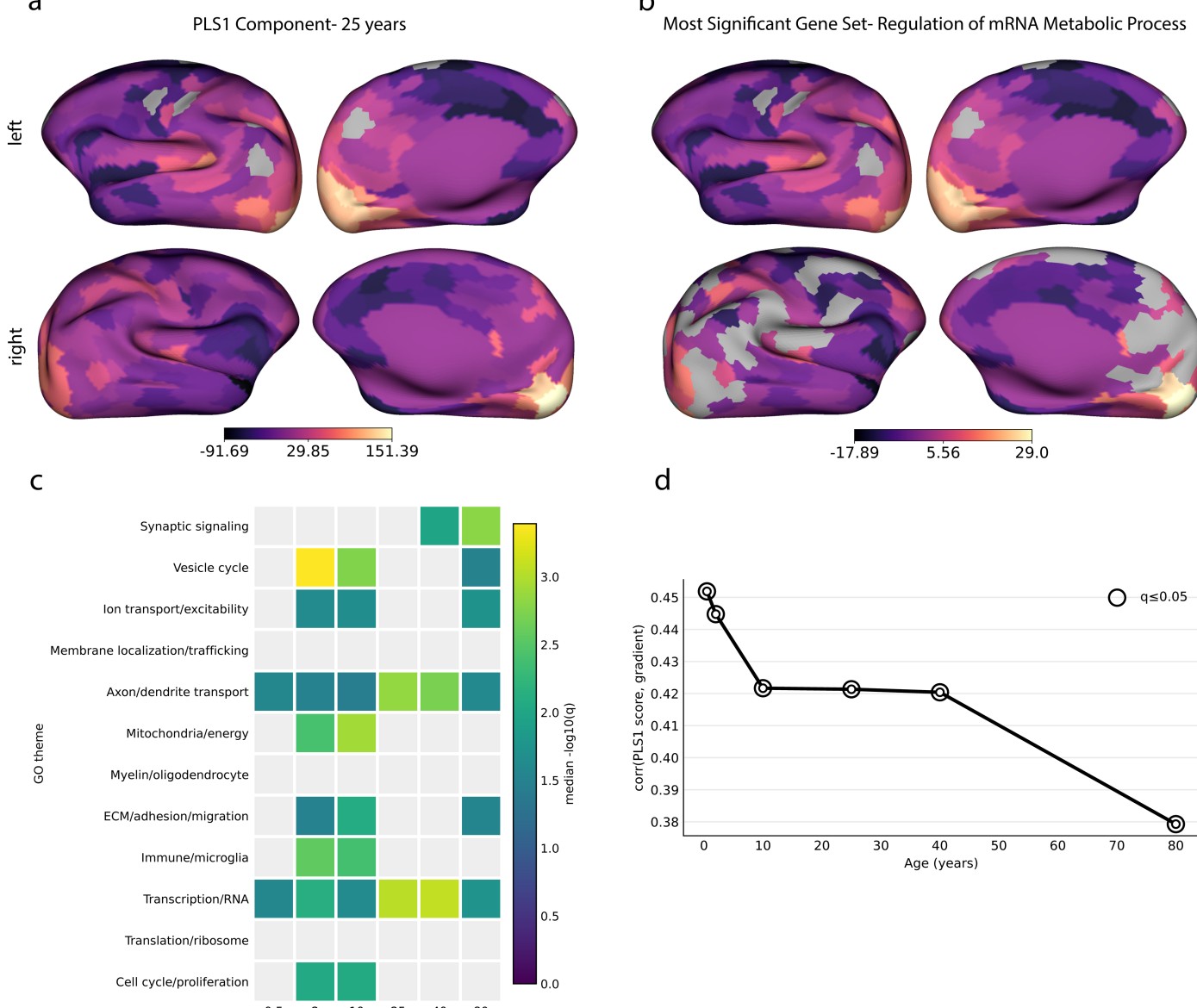

**a**

PLS1 Component- 25 years

left

right

-91.69   29.85   151.39

**b**

Most Significant Gene Set- Regulation of mRNA Metabolic Process

-17.89   5.56   29.0

**c**

GO theme

Synaptic signaling
Vesicle cycle
Ion transport/excitability
Membrane localization/trafficking
Axon/dendrite transport
Mitochondria/energy
Myelin/oligodendrocyte
ECM/adhesion/migration
Immune/microglia
Transcription/RNA
Translation/ribosome
Cell cycle/proliferation

0.5   2   10   25   40   80
Age (years)

median -log10(q)

**d**

q≤0.05

Age (years)

corr(PLS1 score, gradient)

**Extended Data Fig. 9 | Molecular axis predicting cortical mean VS gradient and its biological interpretation across age. a**, Parcel-wise score map at 25 years based on the first component of a partial least squares (PLS) model, relating adult human cortical gene expression (AHBA, parcellated) to the mean VS axis. The component is sign-oriented so that corr(PLS1 score, mean gradient) ≥ 0: warm colours mark parcels whose multigene expression profile predicts higher mean gradient; cool colours mark parcels aligned with lower mean gradient. **b**, Spatial projection of the top enriched GO term, regulation of mRNA metabolic processes. A parcel-wise gene-set score was computed (z-scored expression, coefficient-weighted by PLS1) and mapped to cortex. Co-localization of this map with the PLS1 map (and with the phenotype) links the molecular axis in **a** to mRNA metabolic processes. **c**, GO themes across age. For each age, genes were ranked by signed PLS1 coefficients and analysed with GOrilla. Cells show the median $-\log_{10}$(FDR $q$) of terms within each theme, summarizing which processes are most consistently associated with higher mean gradient at different developmental stages. **d**, Gene–gradient coupling as a function of age: the Pearson correlation between the PLS1 parcel scores and the mean gradient, with points marked when permutation $q < 0.05$. This curve indicates when adult-like molecular patterns best predict the spatial distribution of the gradient (stronger earlier, declining into later adulthood in this dataset).

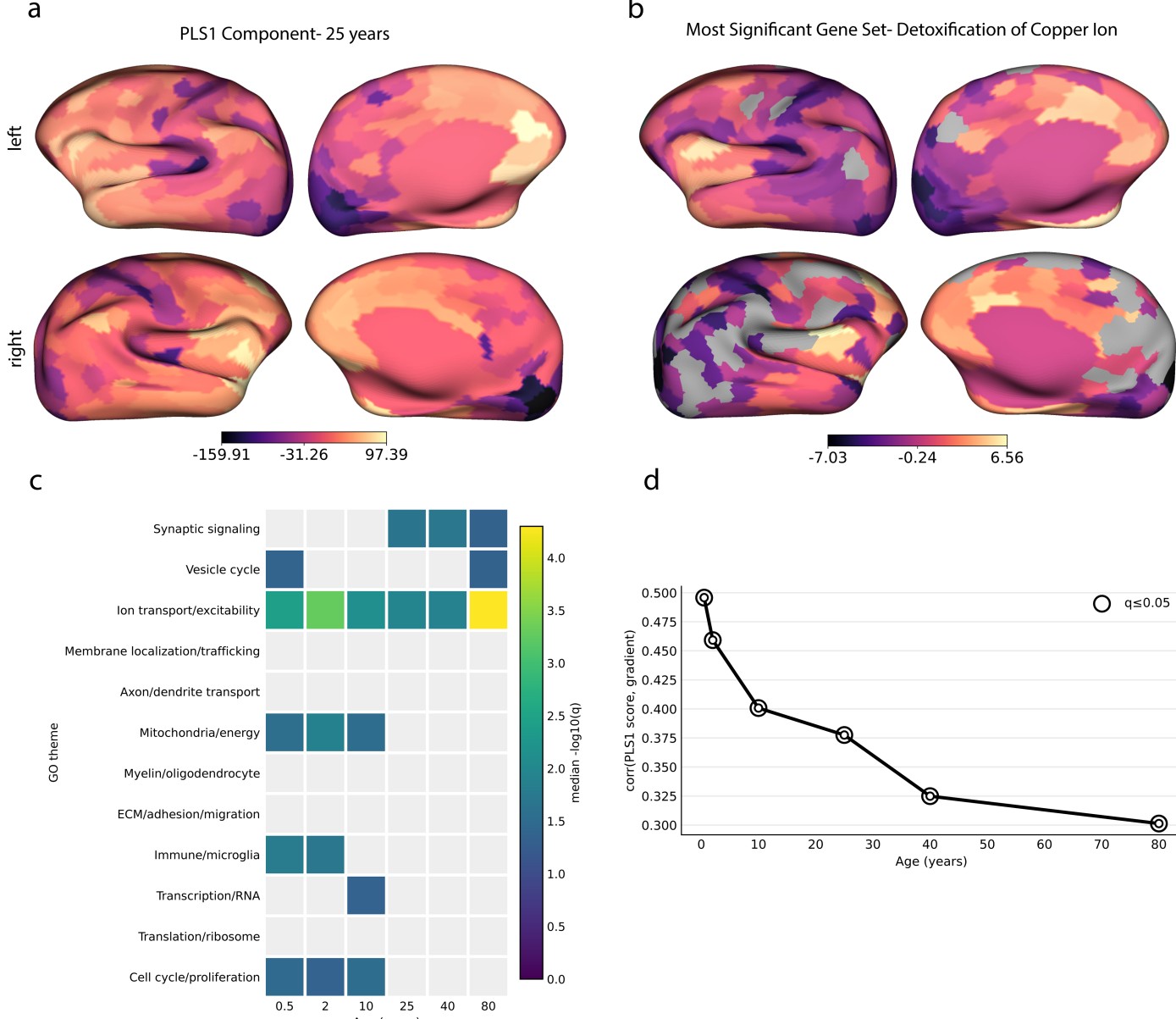

**a**
PLS1 Component- 25 years

left

right

-159.91   -31.26   97.39

**b**
Most Significant Gene Set- Detoxification of Copper Ion

left

right

-7.03   -0.24   6.56

**c**

GO theme

Synaptic signaling
Vesicle cycle
Ion transport/excitability
Membrane localization/trafficking
Axon/dendrite transport
Mitochondria/energy
Myelin/oligodendrocyte
ECM/adhesion/migration
Immune/microglia
Transcription/RNA
Translation/ribosome
Cell cycle/proliferation

0.5   2   10   25   40   80
Age (years)

median -log10(q)

**d**

corr(PLS1 score, gradient)

Age (years)

○ q≤0.05

**Extended Data Fig. 10 | Molecular axis predicting cortical mean MR gradient and its biological interpretation across age. a**, Parcel-wise score map at 25 years based on the first component of a partial least squares (PLS) model, relating adult human cortical gene expression (AHBA, parcellated) to the mean MR axis. The component is sign-oriented so that corr(PLS1 score, mean gradient) ≥ 0: warm colours mark parcels whose multigene expression profile predicts higher mean gradient; cool colours mark parcels aligned with lower mean gradient. **b**, Spatial projection of the top enriched GO term, detoxification of copper ion. A parcel-wise gene-set score was computed (z-scored expression, coefficient-weighted by PLS1) and mapped to cortex.

Co-localization of this map with the PLS1 map (and with the phenotype) links the molecular axis in **a** to ion detoxification. **c**, GO themes across age. For each age, genes were ranked by signed PLS1 coefficients and analysed with GOrilla. Cells show the median −log10(FDR $q$) of terms within each theme, summarizing which processes are most consistently associated with higher mean gradient at different developmental stages. **d**, Gene–gradient coupling as a function of age: the Pearson correlation between the PLS1 parcel scores and the mean gradient, with points marked when permutation $q < 0.05$. This curve indicates when adult-like molecular patterns best predict the spatial distribution of the gradient (stronger earlier, declining into later adulthood in this dataset).

# Reporting Summary

## Statistics

For all statistical analyses, confirm that the following items are present in the figure legend, table legend, main text, or Methods section.

| n/a | Confirmed | |
|---|---|---|
| ☐ | ☒ | The exact sample size (*n*) for each experimental group/condition, given as a discrete number and unit of measurement |
| ☐ | ☒ | A statement on whether measurements were taken from distinct samples or whether the same sample was measured repeatedly |
| ☐ | ☒ | The statistical test(s) used AND whether they are one- or two-sided<br>*Only common tests should be described solely by name; describe more complex techniques in the Methods section.* |
| ☐ | ☒ | A description of all covariates tested |
| ☐ | ☒ | A description of any assumptions or corrections, such as tests of normality and adjustment for multiple comparisons |
| ☐ | ☒ | A full description of the statistical parameters including central tendency (e.g. means) or other basic estimates (e.g. regression coefficient) AND variation (e.g. standard deviation) or associated estimates of uncertainty (e.g. confidence intervals) |
| ☐ | ☒ | For null hypothesis testing, the test statistic (e.g. *F*, *t*, *r*) with confidence intervals, effect sizes, degrees of freedom and *P* value noted<br>*Give P values as exact values whenever suitable.* |
| ☒ | ☐ | For Bayesian analysis, information on the choice of priors and Markov chain Monte Carlo settings |
| ☐ | ☒ | For hierarchical and complex designs, identification of the appropriate level for tests and full reporting of outcomes |
| ☐ | ☒ | Estimates of effect sizes (e.g. Cohen's *d*, Pearson's *r*), indicating how they were calculated |

*Our web collection on statistics for biologists contains articles on many of the points above.*

## Software and code

Policy information about availability of computer code

| Data collection | No software was used |
|---|---|
| Data analysis | Python libraries: Numpy 1.26.4, Scipy 1.13.1, Scikit-Learn 1.4.0, Nibabel 5.2.0, Brainspace 0.1.10, WPCA 0.1, Abagen 0.1.3, NiMare 0.5.5. fMRI + dMRI preprocessing: FSL 6.0, Connectome Workbench. SMSI microstructure fitting: Matlab 2022b R libraries: MGCV 1.8.39, dplyr 1.1.3, tidyr 1.3.0, ggplot2 3.4.4, readr 2.1.4, tibble 3.2.1, purrr 1.0.2. In-house surface atlas from Ahmad et al. 2022. Gene Ontology: GOrilla |

For manuscripts utilizing custom algorithms or software that are central to the research but not yet described in published literature, software must be made available to editors and reviewers. We strongly encourage code deposition in a community repository (e.g. GitHub). See the Nature Portfolio guidelines for submitting code & software for further information.

## Data

Policy information about availability of data

All manuscripts must include a data availability statement. This statement should provide the following information, where applicable:

- Accession codes, unique identifiers, or web links for publicly available datasets
- A description of any restrictions on data availability
- For clinical datasets or third party data, please ensure that the statement adheres to our policy

The Lifespan HCP fMRI data is publicly available via the National Institute of Mental Health data archive (NDA, http://nda.nih.gov). All the data are deposited under the Connectome Coordination Facility repository with the following Collection IDs (BCP: #2848; HCP-D: #2846; HCP-YA: #2825; HCP-A: #2847). The HBN MRI data

# Research involving human participants, their data, or biological material

Policy information about studies with <u>human participants or human data</u>. See also policy information about <u>sex, gender (identity/presentation), and sexual orientation</u> and <u>race, ethnicity and racism</u>.

| | |
|---|---|
| Reporting on sex and gender | Subjects in the Baby Connectome Project (BCP) ranged from 16 days to 6 years old, with 343 subjects 158 males, 176 females, 9 unreported). We used 650 subjects (301 males, 349 females) from the HCP-D, 770 subjects from the HBN (458 males, 275 females, 37 unreported), 1068 subjects (482 males, 586 females) from the HCP-YA, and 725 subjects (319 males, 406 females) from the HCP-A. Across all four cohorts, our dataset included 3,556 subjects. |
| Reporting on race, ethnicity, or other socially relevant groupings | We did not use race, ethnicity, or any other socially relevant groupings. |
| Population characteristics | BCP subjects age ranged from 16 days to 6 years old. Subjects from the HCP-D ranged from 5.58-21.92 years old. Subjects in the HBN ranged from 5.57-21.9 years. Subjects from the HCP-YA ranged from 22-37 years old. Subjects from the HCP-A ranged from 36-100 years old. The median age across all participants was 26 years with a standard deviation of 21.33 years. |
| Recruitment | For BCP data, participants were recruited from existing registries at UNC and UMN based on state-wide birth records as well as from broader community resources (e.g., community centers and targeted day-care centers) to ensure the sample approximates the racial/ethnic and socio-economic diversity of the US census. |
| Ethics oversight | For BCP data, the study protocols were approved by the Institutional Review Board of the School of Medicine of the University of North Carolina at Chapel Hill (UNC-CH), NC, USA. |

Note that full information on the approval of the study protocol must also be provided in the manuscript.

# Field-specific reporting

Please select the one below that is the best fit for your research. If you are not sure, read the appropriate sections before making your selection.

☒ Life sciences ☐ Behavioural & social sciences ☐ Ecological, evolutionary & environmental sciences

For a reference copy of the document with all sections, see nature.com/documents/nr-reporting-summary-flat.pdf

# Life sciences study design

All studies must disclose on these points even when the disclosure is negative.

| | |
|---|---|
| Sample size | Sample size was chosen based on available data across our five imaging datasets. Our final sample size of 3972 is large and ages of participants are evenly distributed across the human lifespan. |
| Data exclusions | Datapoints for which cortical surface meshes with vertex correspondence across the lifespan could not be constructed were excluded from subsequent analyses. Subjects for which fMRI motion parameters exceeded mean Power's FD (absolute sum of motion parameters) exceeding 0.5mm—a common threshold in fMRI studies, were excluded. Finally, we used visual inspection and a clustering procedure to exclude subjects with non-biological or corrupted FC gradients. |
| Replication | We evaluated reproducibility of key lifespan-derived quantities (trajectory shape, peak age, and gradient template definition) using multiple stability analyses. Peak-age estimates from GAMM fits of gradient metrics were reproducible across 20,000 coefficient-draw simulations (parametric bootstrap), vertex-wise trajectories were robust in 4 leave-one-cohort-out refits with good held-out generalization, and the WPCA template used for alignment was stable across 500 age-balanced bootstrap replicates. |
| Randomization | Our study does not involve the division of our dataset into multiple groups. Instead, we are tracking how functional connectivity organization changes with respect to age. Thus, no randomization was required. |
| Blinding | This study is observational and analyzes previously acquired datasets spanning the lifespan. Participants were not assigned to experimental conditions or intervention groups; therefore, randomization was not applicable and was not performed. |

# Reporting for specific materials, systems and methods

We require information from authors about some types of materials, experimental systems and methods used in many studies. Here, indicate whether each material, system or method listed is relevant to your study. If you are not sure if a list item applies to your research, read the appropriate section before selecting a response.

## Materials & experimental systems

| n/a | Involved in the study |
|-----|----------------------|
| ☒ ☐ | Antibodies |
| ☒ ☐ | Eukaryotic cell lines |
| ☒ ☐ | Palaeontology and archaeology |
| ☒ ☐ | Animals and other organisms |
| ☒ ☐ | Clinical data |
| ☒ ☐ | Dual use research of concern |
| ☒ ☐ | Plants |

## Methods

| n/a | Involved in the study |
|-----|----------------------|
| ☒ ☐ | ChIP-seq |
| ☒ ☐ | Flow cytometry |
| ☐ ☒ | MRI-based neuroimaging |

## Plants

| | |
|---|---|
| Seed stocks | N/A |
| Novel plant genotypes | N/A |
| Authentication | N/A |

## Magnetic resonance imaging

### Experimental design

| | |
|---|---|
| Design type | resting-state |
| Design specifications | For each subject/timepoint (some subjects in the BCP were scanned at multiple ages in staggered-cohort longitudinal study design), 4 resting-state sessions were carried out. For the BCP, each session contained 478 TRs, with TR = 800ms . For the HCP-D and HCP-A, each session contained 420 TRs, with TR = 800ms . For the HCP-YA, each session contained 1200 TRs, with TR = 720ms.  For the HBN, each session contained 375 TRs, with TR = 800ms. |
| Behavioral performance measures | Mullen Scales of Early Learning were used for BCP. NIH toolbox cognition battery was used for HCPYA, HCPD, and HCPA. |

### Acquisition

| | |
|---|---|
| Imaging type(s) | functional, structural, and diffusion. |
| Field strength | 3T |
| Sequence & imaging parameters | BCP sMRI data: MPRAGE, SPACE, FOV: 256mm x 256mm, slice thickness: 0.8mm.<br>BCP rfMRI: 2D multiband gradient-recalled echo-planar imaging (flip angle 52 degrees), FOV: 208mm x 208mm slice thickness: 2mm.<br>BCP dMRI: Single-shot EPI, FOV 210mm x 210 mm, slice thickness: 1.5mm, 6 shells, 144 gradient directions<br><br>HCPD + HCPA sMRI: MPRAGE, SPACE, FOV: 256mm x 240mm x 160mm, slice thickness: 0.8mm.<br>HCPD + HCPA fMRI: 2D multiband gradient-recalled echo-planar imaging (flip angle 52 degrees), slice thickness: 2mm, with 72 oblique-axial slices.<br>HCPD + HCPA dMRI: slice thickness: 1.5mm, 2 shells, 180 gradient directions<br><br>HCPYA sMRI:MPRAGE, SPACE, FOV 224mm x 224mm, slice thickness: 0.7mm.<br>HCPYA fMRI: Gradient-echo EPI, FOV: 208mm x 180mm, slice thickness: 2mm<br>HCPYA dMRI: Spin-echo EPI, FOV 210mm x 180mm, slice thickness 1.25mm, 3 shells, 270 gradient directions<br><br>HBN sMRI: MPRAGE, SPACE, FOV 224mm x 224mm, slice thickness: 0.8mm or 1mm depending on scanning sites.<br>HBN fMRI: Gradient-echo EPI, FOV: 208mm x 208mm, slice thickness: 2.4mm<br>HBN dMRI: Spin-echo EPI, FOV 208mm x 208mm, slice thickness 1.8mm, 2 shells, 128 gradient directions |
| Area of acquisition | whole-brain scans were acquired. |
| Diffusion MRI | ☒ Used   ☐ Not used |

| Parameters | BCP: 144 directions, 6 shells with b = 500, 1000, 1500, 2000, 2500, 3000, AP and PA encoding directions<br>HCPD + HCPA: 180 directions, 2 shells with b = 1500, 3000, AP and PA encoding directions<br>HCPYA: 270 directions, 3 shells with b = 1000, 2000, 3000, LR and RL encoding directions<br>HBN: 128 directions, 2 shells with b = 1000, 2000, PA encoding direction with 1 AP B0 |

## Preprocessing

| Preprocessing software | Preprocessing involved an in-house pipeline consistent with the HCP minimal preprocessing pipeline suited for infant neuroimaging data. Softwares used included FSL and Freesurfer. |
| Normalization | Rigid transform |
| Normalization template | In-house brain template |
| Noise and artifact removal | Bias field correction for sMRI data. ICA-AROMA was used for rsfMRI nuisance regression. |
| Volume censoring | Not applicable. |

## Statistical modeling & inference

| Model type and settings | Statistical analyses used generalized additive models/mixed models (GAM/GAMM) in R (mgcv; bam with fast REML for large models) with penalized regression spline smooths; smoothing parameters were estimated by (fast) REML. Lifespan models used a monotonic age reparameterization (square root of age). Vertex-wise lifespan models included random intercepts for cohort and subject ID (to account for between-cohort differences and repeated measures in BCP), sample-density weights to balance age bins (and cohort size), and a two-step heteroscedasticity correction (GAMM on squared residuals + inverse variance weighting in the final fit; basis dimension tuned with low-k harmonization and higher-k final fits). Additional models included ordinary least squares regression for cross-sectional HCP-YA cognition, linear mixed-effects models for repeated-measures BCP cognition, and partial least squares regression (PLS1; scikit-learn) with permutation testing for transcriptomic analyses. |
| Effect(s) tested | Primary effects tested were age-dependent changes in gradient values/metrics (smooth age terms) and departures from linear age effects (per-vertex LM vs GAM partial F-tests with FDR correction across vertices). Sex differences were assessed via sex-specific deviations from the population trajectory using sex-by-age smooths. Potential confounding by total brain volume was evaluated by refitting vertex-wise models including total brain volume as a covariate. Cognition analyses tested associations between gradient metrics and cognitive scores while adjusting for age ($\beta$_grad in OLS/LMM; smooth main effect of z(gradient) in GAM), and tested age modulation of these associations via an age×gradient interaction (tensor interaction ti(Age, z(gradient)) in GAM). Multiple comparisons were controlled using Benjamini–Hochberg FDR within each analysis family (e.g., across vertices or across cognition×metric tests). |

| Specify type of analysis: | ☐ Whole brain | ☐ ROI-based | ☒ Both |

| Anatomical location(s) | We used cortical surface meshes and previously studied network and whole brain parcellations from Yeo et al. 2011 and Schaefer et al. 2018 , respectively. |

| Statistic type for inference<br><br>(See Eklund et al. 2016) | We used frequentist inference. Lifespan trajectories were modeled with GAMMs/GAMs (mgcv/gamm4; bam with fast REML) using penalized cubic regression splines (e.g., bs="cs") with smoothing parameters estimated by (fast) REML. Significance of smooth terms and tensor interactions was assessed using mgcv's approximate (Wald-type) tests (reported as F statistics for Gaussian models). Nonlinearity at each vertex was assessed via partial F-tests comparing nested models (linear vs smooth GAM). For cross-sectional HCP–YA cognition, we used ordinary least squares (lm) and tested regression coefficients using two-sided t-tests. For BCP cognition with repeated measures, we used linear mixed-effects models (lme4::lmer) with Satterthwaite degrees of freedom (lmerTest) for fixed-effect t-tests. Transcriptomic PLS models were evaluated using permutation testing of the in-sample correlation. All tests were two-sided unless stated otherwise. |
| Correction | Multiple comparisons were controlled using the Benjamini–Hochberg false discovery rate (FDR) procedure (q=0.05) within each analysis family. Specifically: (i) vertex-wise nonlinearity p-values (linear vs smooth GAM) were FDR-corrected across cortical vertices (separately for each gradient axis); (ii) HCP–YA cognition regressions were FDR-corrected across 9 cognitive scores × 10 gradient metrics (90 tests); (iii) BCP Mullen mixed models were FDR-corrected across 6 outcomes × 10 gradient metrics (60 tests); and (iv) transcriptomic permutation p-values were FDR-corrected across ages (within each gradient axis). No cluster-based correction was used. |

## Models & analysis

| n/a | Involved in the study |
| ☐ | ☒ Functional and/or effective connectivity |
| ☐ | ☒ Graph analysis |
| ☒ | ☐ Multivariate modeling or predictive analysis |

| Functional and/or effective connectivity | We used Pearson's correlation coefficient to measure functional connectivity. |
| Graph analysis | We used weighted functional connectivity graphs (pairwise Pearson correlation coefficient) and computed their diffusion map embeddings (referred to in manuscript as functional connectivity gradients). We |

performed auxiliary analysis on the functional connectivity graphs that included computing functional
connectivity degree (row-wise sum of thresholded FC matrices).

