## [Peer Review File · Nature]

Functional Hierarchy of the Human Neocortex Across the Lifespan

Corresponding Author: Professor Pew-Thian Yap

Version 0:

Reviewer comments:

Referee #1

(Remarks to the Author)

The manuscript by Taylor and colleagues entitled "Functional Hierarchy of the Human Neocortex from Cradle to Grave" explores how cortical organization — as described using low-dimensional embeddings of functional connectivity and microstructural measures -- changes across the lifespan. The main framework for the study is the description of brain connectivity using a low-dimensional embedding approach. In general, the topic of this paper is of broad interest at the moment, with several high profile recent articles characterizing developmental changes associated with gradient organization within bounded portions of the lifespan. One key contribution of the current study is thus the scale (n=2735) and breadth of ages included -- from neonate to 100 years of age -- within a consistent analytic framework based on generalized additive mixed models. The study predominantly focuses on the observations regarding the first three gradient patterns, whose alterations throughout the lifespan are characterized and interpreted within the context of the cognitive modes they describe. Overall, the manuscript is exceptionally well written and the results are, for the most part, clearly presented. I would nevertheless offer the following points for consideration by the authors:

Major points:

- I would urge caution in interpreting the functional connectivity gradients as reflecting a hierarchy. As I understand it, the analysis of diffusion map embedding does not itself inherently capture hierarchical interaction between time-series, and the description of 'hierarchy' is made based on post hoc interpretation. While this is certainly a valid interpretation, it would be helpful to clarify how the term 'hierarchy' is applied in the current analytic context.
- I would also urge caution regarding use of terms positive and negative (example lines 175-176), as the gradient dimensions are sign invariant. An alternative would be to specify the respective ends of each dimension, as is broadly already the case throughout the manuscript. An additional example of this is the use of the term 'apex' (for example, in line 192), which assumes a specific orientation of the spectrum. While these terms serve well to orient the reader, they are not inherent to the values represented by the analysis and, without additional context, could be construed as misrepresenting the results.
- There are numerous qualitative observations presented in the manuscript that are intriguing, but I believe several would benefit from more quantitative testing/validation. While I generally appreciate the level of description of the results, there are places where the comparative description is difficult to assess given how the figures are presented. For example, line 186 describes the 'focal tuning of the positive peaks', referring to Figure 2. It would be helpful to show the overlapping distributions to help the reader see this comparative description. Another example is the statement in line 181 regarding predominance of short-range connectivity in early-life, which I found difficult to observe from the results presented in Figure 2. Any further data presentation to support such claims would be helpful.
- Regarding the discussion of the eigenvalue ratio in the paragraph beginning with line 572: Intuitively, these findings seem

to contradict the prior observation in the literature of increasing long-distance segregation and local integration with age. As this paragraph aims to discuss the gradient findings in the context of the graph theory based approaches/literature, it would additionally be helpful to further characterize the current findings within the well developed literature on graph-based connectivity changes associated with aging.

Minor points:

- I may have missed it, but considering the importance of age breadth to the current study, a plot showing the full age distribution could help the reader to appreciate this aspect of the current study.
- The number of included participants presented at the beginning of the Methods section appears to be the same at those presented after post-processing exclusion criteria were applied (In 733). How many subjects were excluded based on these criteria for each cohort?
- I may be mistaken, but the captions for Fig 6 d-f appear to be missing.
- Of course, I leave the following stylistic decision to the authors, but might suggest formalizing some of the language in the Discussion, specifically, 'enormous step' (line 511) and 'sorely lacking' (line 515).
- Please check author order for reference 48.

Referee #2

(Remarks to the Author)

Taylor and colleagues present an analysis of the changing composition of functional cortical gradients across the human lifespan and their relation to structural (mainly diffusion) based metrics. There is clearly a strong interest in capturing the normative trajectories of functional brain organisation in a cohesive manner to the point of establishing a benchmark or reference point onto which to anchor developmental or ageing related deviations and so this work should certainly be considered timely and of interest to the scientific community (see also Sun et al. 2023 bioRxiv). I do however have both conceptual as well as methodological considerations that somewhat temper my enthusiasm. These are predominantly related to the challenges of harmonisation more broadly and alignment in the context of gradients specifically. I hope these are taken in the constructive manner in which they are very much intended, if this paper makes it into this journal it will likely act as a lightning rod for all criticism levelled against the gradient approach or other attempts at quantifying lifespan changes in brain organisation, I hope my comments can help pre-empt those likely criticisms and bolster the findings to the point of showing high generalisability that can ideally stand the test of time.

Conceptual:

The authors state they "developed a unified framework" for doing gradient analyses. It is not entirely clear to me how generating a template gradient from an entire dataset and aligning individual embeddings to the template is a new framework. This is exactly the approach integrated in the cited BrainSpace toolbox and pretty much the approach all the cited gradient papers have taken to date. Perhaps there is some nuance in the way the template was generated that I am missing, but that could then maybe be clarified briefly in the results section where this is introduced.

Related to the previous point, I assume that many of the expected trajectories are non-linear and that the datasets are not evenly distributed across the lifespan (i.e., they are by definition cohorts designed around specific lifespan windows), let alone that the highest sampling density is likely or coincidentally in the periods of the lifespan where change is most dynamic. Thus, I wonder how a template generated from the entire sample can be ensured to not be biased towards those batches in the aggregated dataset where sampling is densest or most stable? Is there a risk that all individuals are for example aligned to the main HCP dataset and that any individual variation is a reflection of that specific anchor point? Were other alignment procedures evaluated? How was harmonisation handled?

How well do the GAMMs fit? Aside from the random bootstrapping, was there some permutation or LOO analyses done to evaluate whether these too could be influenced by particular datasets or developmental windows?

The period between 2-10 yrs is often referenced as the time of major change, coinciding with many other developmental milestones. This is however a rather broad window and I would very much be interested to see if that can be narrowed down further, potentially by determining inflection points in the described trajectory?

Given the sole reliance on HCP cohorts, do the authors think these results are generalisable to other cohorts? Have the authors conducted any out of sample validation of the gradients composition at different stages in the lifespan in any of the other available open datasets for example?

How is the structural affinity matrix different from the MSN approach by Seidlitz et al (aside from including some NODDI measures)? Is there any concern that many of the diffusion based measures are highly correlated to each other and to some extend inverses of each other? Did the authors consider other, dare I say more established structural connectomic approaches to benchmark the structure-function coupling approach (see for example work by Casey Paquola and Bratislava Mistic in that context)? Also, given the change in the GM/WM balance in very early development how were these images processed and handled? Was there some harmonisation/normalisation done to deal with the fact that NODDI is somewhat untested in the early age range? Could this introduce any bias in the analysis if no harmonisation was done across cohorts?

What do these affinity matrices represent?

Methodological:

Is including cohort as a random effect sufficient for harmonisation across studies? Were other harmonisation approaches used? More crucially perhaps doesn't cohort also perfectly intersect with age and if so how do the authors ensure this does not bias the results or perhaps even remove important information needed to establish trajectories in the first place?

Should processing pipeline be included as a factor in all models given that it perfectly intersects with one cohort and one specific age-range how can the authors rule out that any difference in that age range are not due to acquisition/processing/cohort effects? Especially, since this is most likely the period in the lifespan with the most pronounced changes.

How were the timepoints in Fig 2 chosen? Also not sure how to interpret the contraction of the gradient in F2B or the shape of the gradient in the SA-MR plane if I'm honest.

How was the GAMM parametrisation determined?

Is the data homoscedastic across the lifespan or is there variation in variance with age as there is in structure? If so, would a modelling framework that captures that variation be more appropriate (even if it is more complex)?

What is the impact of the subject random effect given that only a subset of studies have longitudinal data and only in specific time periods for specific intervals?

Was biological sex effect modelled? Is/should total brain volume be modelled or accounted for in both the functional and structural features given the likely relationship between affinity and distance in both domains?

I'd suggest including some example figures of the GAMM fits/trajectories to show the patterns of variation discussed (maybe the most and least significant regions for example). This would also help visualise how well the site random effect harmonisation worked (by colouring them by cohort for example) and would allow visualisation of more fine-grained sensitivity analyses.

What does "the median gradient value in each network" represent in the context of all above mentioned methodological considerations (specifically in relation to alignment)?

Is the "the median distance between all pairwise Euclidean distances" functionally different from the centroid approach? Have the authors quantitatively compared those approaches?

Minor/terminology:

It seems quite fascinating to me that the curves in F4A seem to be almost each others mirror images in many cases. Is that expected? What do the error bars represent in that panel btw?

There is a lot of inference between what these trajectories, timepoints, changes etc mean in terms of cognitive development, but this doesn't need to be inferred and most of the utilised datasets will have ample cognitive data available to make that link (and potentially even poke at the elusive question of causality in the case of longitudinal data)

Are the 3 directions in embedding space universally accepted as representing association, unimodality and modulation?

Not sure the reference to "Evolution" across the lifespan is appropriate as there isn't really analyses or discussion of evolutionary processes, perhaps "development" or "ageing" are more appropriate?

I am personally not partial to the phrasing "cradle to grave" that seems to recur rather a lot and isn't particularly specific or scientific.

There has been some recent backlash against the notion of smooth topological gradients as a model to capture function, and whilst I personally disagree with this backlash I wonder whether its worth mentioning in the discussion (Petersen et al. 2024, Neuron) at least perhaps to acknowledge that the notion of gradients isn't universally accepted gospel.

Referee #3

(Remarks to the Author)

The authors of this audacious manuscript make a profound claim: that by examining gradients of resting state functional connectivity across the entire lifespan, they have discerned succinct distillations of core developmental and aging related processes. The manuscript is well written, but its claims are so potentially impactful, that it required multiple readings, and digging into the cited references which provide the bases for the three axes which are proposed. Two of these (the sensory-association, and visual-spatial) already emerged from the original use of the dimensional reduction technique of diffusion

embedding by Margulies et al. in 2016 which sparked the burgeoning interest in brain gradients. These were the first and second gradients, respectively, in that seminal paper, and they have been consistently confirmed across datasets, and also found to be developmentally relevant: their order, in terms of variance accounted for, is reversed in early childhood, with the adult-like pattern emerging around the end of the first decade of life, see Shifting gradients of macroscale cortical organization mark the transition from childhood to adolescence - PubMed (nih.gov).

I was not familiar with the third gradient focused on here, depicting modulation-representation (MR), in which frontal-parietal control and attention networks lie at one extreme, and default, sensory, and visual networks lie at the other. I was skeptical about this commingling of default, sensory and visual networks, but my skepticism was resolved by studying Zhang et al. (ref # 6 in this ms) which convincingly presented three gradients, one of which they identified as the MR gradient, in two data sets from healthy adults. Thus, I've come to agree with the authors that their analyses represent a meaningful advance in our understanding of the fundamental principles underlying brain functional organization across the lifespan. The authors acknowledge that this work is frankly exploratory and descriptive – but that has been the nature of nearly all of brain imaging heretofore. Despite the myriad limitations, the availability of large open datasets, increasing statistical sophistication, and powerful computational tools are finally producing novel insights into the principles underlying brain functioning.

Along those lines, S. Park and colleagues just published in Nature Neuroscience (available June 10, 2024) on “A shifting role of thalamocortical connectivity in the emergence of cortical functional organization.” They did not take their analyses throughout the entire lifespan, but they encompass ages from infancy to age 22 years, so there is considerable overlap. Including the role of thalamocortical connectivity would be beyond the scope of the current manuscript, but it should acknowledge this recent contribution. Park et al. also noted that their datasets included scans obtained while infants slept, along with datasets obtained in (at least initial) wakefulness. They performed auxiliary analyses to confirm that their results were not skewed by sleep/wake. Taylor et al. should consider a similar strategy; at the very least, this inevitable difference in arousal state between scans obtained in infants and older children/adults must be acknowledged.

Beyond appreciation for the work presented, some additional quibbles bear mentioning. Like much of the field, the authors use Euclidean distances, whereas the cortex, is best considered as a surface in which geodesic distances are more appropriate. Converting all this work to geodesic space would be non-trivial, and I do not insist on it, but the lack of doing so should be noted as a limitation. For example, might this methodological issue play into the observation “that the development of these networks from 5 to 30 years is driven by changes in the scale of our gradient axes themselves rather than changes in the position of default and control networks relative to those axes”?

One minor typographical error in line 499: “indicating that these global hierarchies to not exhibit” should be “indicating that these global hierarchies do not exhibit...”

Finally, the authors state in three places that their analyses are based on 2735 datasets, but it is not easy to work out how many datasets were excluded in total. It is important to have a sense of how stringent or liberal were their inclusion and exclusion criteria, particularly for datasets spanning “from cradle to grave.” The authors should also make available the specific non-PHI identifiers of the data they retained, to facilitate replication/extension of this novel work.

F. Xavier Castellanos

Version 1:

Reviewer comments:

Referee #1

(Remarks to the Author)

Referee #2

(Remarks to the Author)

I thank the authors for their constructive response to my comments. However I still have some major reservations some of which may be stemming from some lack of detail/clarification in my earlier comments. Hence let me provide specific responses using the same numbering system for clarity.

2.1 While the weighted PCA may reduce some bias driven by unequal distribution of sample size across the lifespan, I would still encourage the authors to actually show that it does (for example by running an ablation analyses with equally size samples across the bins).

In addition, one of my main original concerns was that perhaps not all age-bins are created equal. For example: if we hypothesise that say between years 1-2 there is exponential change that could be 5* larger than what happens in any of the subsequent 1 year bins, then taking the weighted approach across 10 1-year bins for the first 10 years of life would still bias the template towards the latter 9 years where little changes and negate the period of largest change. I don't see how the added analyses circumvents that?

I could perhaps agree that this combined with the alignment procedure retains variation across individuals but that doesn't mean that the variation is an ageing signal so much as an alignment to the template.

Further to the comment on harmonisation: to my knowledge using a harmonised pipeline (in so far as that is even realistically possible let alone desirable across the lifespan) does not in and of itself harmonise data from different sources. I appreciate that harmonising functional data may not be a trivial task but it does seem quite essential to the results presented here. Perhaps including more data-sets with overlapping age-range would allow the authors to conduct a more quantitative sensitivity analyses to determine whether any residual site related variation is still present in the data? Or alternative they could deploy a ComBAT like harmonisation approach or choose a modelling framework that more readily allows the modelling of higher moments in the data distribution (i.e., HBR/GAMLSS).

2.2 I'm not convinced that any of the suggested measures of either R-squared or percentage non-linearity address the question of model fits and would deem it essential to show in more depth that there is no age or site bias that could be driving the trajectory estimation. Rather than seeing plots of the DoF it would still be informative to see representative plots of the fits over the raw data in addition to the suggested permutation/LOO sensitivity analyses. While I appreciate that this may be computationally expensive perhaps some compromise can be made by only conducting these sensitivity analyses for a few random/extreme variables/vertices?

2.3 Again, this is not an entirely satisfactory response. Inflection points could be quantified by derivatives and it simply seems off that during the entire 2-10 window there are no peaks or periods of more rapid change. How can the authors rule out that this smooth transition is not a side-effect of the model over smoothing (due to lack of data/harmonisation/diversity etc.)

2.4 Rather than adding another dataset to the model. Could the authors show out of sample generalisability?

2.5 My comment about harmonisation and dealing with the change in contrast in early development (and hence the untestedness of NODDI in early development) remains unanswered I believe.

2.6 I thank the reviewer for the clarification on their GAMM procedure. As per my comment in 2.1 it would still be could to verify this by adding more cohort with overlapping ages or alternatively using a secondary harmonisation procedure to illustrate robustness.

2.7 This is admittedly very much a catch 22 problem as using a uniform pipeline may introduce alternative challenges since especially the very young cohort likely require some customisation. A stronger alternative would be some permutation of pipeline or at the very least a sensitivity analyses that shows that for example within one dataset different pipelines at different ages do not impact the results. Think for example the context of running regular free surfer in the BCP cohort kids that are older than 2, but infant FreeSurfer in the ones under two and comparing the trajectories to a unified pipeline to ensure that there is no impact.

2.8 Thank you for the clarification, no further comments on this.

2.9 Thank you for the clarification, no further comments on this.

2.10 This is a very interesting and to my knowledge novel approach. Rather than removing variance from the data entirely the authors may also want to consider interpreting the variability in variance itself as a meaningful biological signal (perhaps in the context of for example the vulnerability hypothesis).

2.11 If that is the case, what do the longitudinal data-points add to the model?

2.12 Perhaps a general comment across these additional analyses would be to add them with some minimal interpretation to the supplement. Specifically I'd be curious to see if the TBV adjustment would have some age-related component to it. It's encouraging to see that the overall trajectories are unaltered by including this covariate, but I could imagine that it may play a role in period where the volumetric tissue changes are somewhat accelerated (i.e., very early development and (very) old age) that wouldn't perhaps be apparent when compared across the entire lifespan.

2.13 Very nice to see :)

2.14 Thank you for the clarification.

2.15 Where are these comparisons in the manuscript/supplement?

2.16 Thank you for the clarification.

2.17 Agreed with the authors that this adds to the results.

2.18 Agreed that this is better and closer to existing literature.

2.19 Thanks :)

2.20 Its still in the title though :)

2.21 Thank you this is helpful I think.

Referee #3

(Remarks to the Author)

Taylor et al. have revised their examination of functional connectivity (FC) gradients across the human lifespan and have satisfactorily addressed my prior relatively minor concerns. The revised manuscript includes the incorporation of an additional dataset (Healthy Brain Network (HBN)) along with examination of potential covariates or interest/confounders, such as biological sex. The overall results seem to hold.

The initial review requested that the authors address the Neuron perspective published by Petersen et al. in which they challenged the fundamental validity of pursuing gradient-based analyses based on the well-established principle of arealization – asserting that cortical areas are valid entities and implying that gradients are likely fictions arising from smoothing across large samples. The authors mention this concern in the last paragraph of the discussion: “There is strong evidence that gradients of FC, microstructure, and gene expression represent crucial organizing axes of the cerebral cortex [2]. Still, recent pushback about the power of gradients as a comprehensive framework for understanding topographic organization of the cortex bears mentioning. The principle of arealization, whereby the cortex is divided into discrete areas which are differentiated by cytoarchitecture and connectivity, is in some ways at odds with the claim that gradients represent fundamental organizing axes on an individual level [66]. Gradients of functional connectivity typically do not smoothly vary with respect to space on an individual level, reflecting instead sharp boundaries between functional systems. At a population level, however, gradients are smooth and excel at describing functional topography. The interpretation of FC gradients in individuals remains an important area of future research, however the present study focuses on large-scale population-level effects.”

The phrase “is in some ways at odds...” fails to convey the direct frontal attack on gradients asserted by Petersen and colleagues. To me this harkens back to vigorous debates regarding the physical nature of light – wave vs. particle. We now all accept that there is no resolution of this question – it all depends on the experimental context/purpose. The real test of validity is explanatory utility. The initial Margulies et al. paper on gradients provided a novel explanation for the physical distribution of default mode network centroids based on analyses of the principal gradient. Such an insight had not emerged from arealization-based perspectives. Here, the authors have provided some data potentially bearing on such explanatory utility, but it’s presented as a supplementary table with little explication. For example, what do Eval1, Eval2, and Eval3 refer to? At any rate, 11 of the 60 relationships with aggregate measures from the Mullen in the Baby HCP presented were below an alpha of 0.05, which does support the inference that some may not be type 1 errors, thus indirectly supporting the explanatory utility of these gradients at least in the youngest participants.

However, given the challenge posed by Petersen et al., and the availability of standardized cognitive/motor measures such as the NIH ToolBox from the HCP and HBN datasets, I urge the authors to consider reexamining the relevant question of explanatory utility of their 3 gradients. They are in the best position to relatively rapidly report whether these gradients can truly be asserted to have quasi clinical utility. [Quasi clinical, because the field lacks sufficiently reliable clinical indices, and most of these datasets were not designed to examine such questions.]

If they are able to provide further supportive data from examining gradient-behavior relationships, I would suggest that the Petersen et al. challenge be addressed in the paper’s introduction rather than at the end. This is a major scientific challenge – and it merits a frontal response. At the end of the day, disputes will continue regarding the fundamental validity of functional connectivity because we still lack an understanding of the neurobiological bases for this phenomenon. The reasons why all nervous systems expend such massive quantities of energy in maintaining low-frequency correlations across the brain remain profoundly mysterious. Simple Hebbian arguments are belied by elegant experiments such as documenting rapid changes in functional connectivity after wearing a cast on one’s arm and removing it. Presumably synaptic strengthening/weakening does not occur so rapidly. This is the major challenge of 21st century human neuroscience in my opinion, and this manuscript has the potential to advance the relevant science.

Trivially, on line 605, the authors describe a correlation as “negative” whereas it is manifestly positive both in the figure and in the following text “($\rho = 0.25$, $p < 10^{-58}$).”

In the first paragraph of the discussion, the authors write: “Integrating the vast and diverse body of fMRI literature into an easily interpreted and succinct framework represents a central obstacle to neuroscience; the present study offers a substantial step forward towards achieving this goal.” I object to the use of the word “obstacle.” They mean “challenge” or similar; otherwise, they need to change the sentence structure.

In line 504, they unnecessarily capitalize “early life” as “Early life.” In that same paragraph, they switch from reporting maxima of age-related changes from one decimal point to two. Given the breadth of most of the corresponding confidence intervals (e.g., 95% CI 9.5–17.8 years), such hyperprecision is unwarranted. I recommend staying with one decimal point.

In line 800, the period after "index" is missing. In line 860, "quality" is misspelled.

I reiterate that this prodigious effort likely represents a substantial advance in human neuroscience and support its publication.

Version 3:

Reviewer comments:

Referee #2

(Remarks to the Author)

The authors have made good-faith efforts to address most of my concerns with additional analyses and clarifications. The LOCO analysis, balanced bootstrap validation, and derivative-based inflection point identification are particularly strong additions that increase the confidence in the robustness of the results. However, some (perhaps philosophical) disagreements remain or could be addressed more thoroughly below. In addition, although I think this may be more a role for the editor than for me as a reviewer I do agree with the assessment of reviewer 1 that perhaps the advance from the already published Sun et al paper is somewhat limited and more of methodological nature than foundational nature.

Remaining queries:

1. NODDI/SMSI Clarification (Comment 2.5):

While the authors clarify they use SMSI not NODDI, they somewhat dismissively cite papers about NODDI's use in pediatric populations rather than directly addressing my core concern about contrast changes in early development. I imagine this could be quantitatively tested?

2. Similarly with pipeline consistency (Comment 2.7):

The authors argue against my suggestion of comparing different pipelines (Infant FreeSurfer vs. standard FreeSurfer - to clarify my suggested was to either interleave or bootstrap these pipelines around the inflection point precisely to avoid a seam) by claiming it would introduce a "non-biological seam." While the unified iBEAT-based pipeline approach is defensible, no empirical evidence is provided that this single pipeline performs equally well across all ages in the present context.

3. Age-Bin Construction:

The clarification about using $\alpha = 0.50$ power-law transform for age bins is helpful, but the fundamental concern about whether early developmental periods receive appropriate weighting relative to their biological importance remains partially unresolved and perhaps it could be established first that these trajectories indeed follow such a power law (while I appreciate the circularity in that, there must be a way to provide some more informed weighting).

Referee #3

(Remarks to the Author)

Taylor and colleagues have responded adeptly to the constructive reviews, thereby further enhancing the value of the approach they have pursued. The relevance of functional gradients is thoroughly established by the data presented and the ingenious analyses performed. The manuscript is well written - with only a single typograph error per my reading. On line 209, they duplicate "association" inadvertently. I appreciate particularly their adoption of the particle/wave metaphor in the introduction. I congratulate the authors on a worthy contribution and happy to be associated with this paper.

Francisco Xavier Castellanos

Referee #4

(Remarks to the Author)

Taylor et al. present a meta-analysis of multiple human brain imaging datasets ranging in age from infancy to 100 years old. They use these data to model the lifespan trajectory of functional connectivity (FC) gradients. Three previously described axes of FC are analyzed, the sensorimotor-association (SA) axis, the modulation-representation (MR) axis, and the visual-somatosensory (VS) axis. Taylor and colleagues find that the developmental trajectories of these three axes are independent and non-linear though all three show the largest changes between 0-4 years of life.

To ground these models in biological underpinnings the authors relate the FC gradients to metrics of intelligence, structural fMRI measures such as myelin thickness, and transcriptomic enrichment. I appreciate the addition of these analyses as they do add context to otherwise quite abstract descriptions of brain organization. With regard to the transcriptomic enrichment analysis in particular, I have three major and two minor comments that I believe will bolster the value of this addition.

Major

1. The authors currently only perform PLS analysis against the SA gradient. It would enhance the validity of this section by including at least one of the other axes (I would suggest the MR axis). The other behavioral analyses include all three FC gradients and by doing so the authors can draw conclusions about how the variations of the gradient trajectories may relate to these metrics. Without another FC axis to compare to in the transcriptomic enrichment, I am left wondering if this is in fact a hallmark of the SA gradient or if the other gradients would show similar transcriptomic and age-related patterns.

2.The analysis of this result is fairly superficial. It is not surprising that the strongest related GO terms are in the synaptic signaling family given that the underlying mechanism of functional gradients is in fact synaptic activation. I suggest the authors dig a little deeper into this GO term family for more nuanced results. For example, Gao et al. 2020 (DOI: <https://doi.org/10.7554/eLife.61277>) perform a similar analysis against ECoG recordings and find that genes for ion channels known to mediate fast versus slow synaptic dynamics correlate with different timescales of brain activity. The study presented here explores timescales well beyond this, but it's possible that genes related to, for instance, neuromodulators vs neurotransmitters could have different PLS weights.

3.The discussion of these results is largely without interpretation or connection to prior literature on this topic. At least two recent studies have performed similar analyses with much more in-depth characterization of spatial molecular programs and functional gradients; Vogel et al. 2022 (DOI: <https://doi.org/10.1073/pnas.2219137121>) and Yunman et al. 2022 (DOI: <https://doi.org/10.1016/j.scib.2022.01.002>). I encourage the authors to speculate on what the relationship between these transcriptomic patterns and gene gradients could mean for human brain development. This will be particularly important with the addition of another FC gradient axis in the analysis whether the results are similar or different.

Minor:

1.I see in the methods that the authors performed parcellation shuffling as a control. It would be a nice addition to add a visual of this to the figure to ensure validity of the PLS results.

2.The AHBA dataset used for this section consists of only adult donors. This potentially causes a confound when applying to very young and very old ages in the FC data. I appreciate that there are few resources like this available and would just consider acknowledging the limitation.

Referee #5

(Remarks to the Author)

I co-reviewed this manuscript with one of the reviewers who provided the listed reports.

Version 4:

Reviewer comments:

Referee #2

(Remarks to the Author)

The authors have significantly improved the manuscript. While some philosophical disagreements may remain (e.g., about whether to compare multiple pipelines), the authors have provided reasonable justifications backed by empirical evidence. The smooth trajectories and consistent reconstructions across ages support their unified pipeline approach. Specifically with regards to my remaining open major comments:

1. NODDI/SMSI Clarification: The authors now provide a proper technical explanation of how diffusion-weighted image normalization (with respect to $b=0$ images) inherently reduces between-scan intensity variation. They explain the multiple normalization steps (DWI normalization, feature-wise z-scoring, individual-wise z-scoring) that mitigate age-related contrast effects. Direct quantitative tests from their own data would have rounded the paper better in my opinion but perhaps this should be considered beyond the scope.

2. Pipeline Consistency:

This is the strongest improvement. The new Figures S21 and S22 provide exactly the empirical evidence requested: Figure S21 shows visually consistent tissue segmentation and surface reconstructions across the lifespan, Figure S22 demonstrates smooth volumetric trajectories with no obvious discontinuities

The explanation that the pipeline was initially developed for ages 0-5 (where contrast changes are most dramatic) then extended to older ages is logical

3. Age-Bin Construction:

The authors now clearly distinguish between two uses of $\alpha=0.5$:

Minor:

For template construction: implementing monotonic reweighting to upweight early development

For GAMM fitting: reparameterizing age for finer resolution in early years

Referee #4

(Remarks to the Author)

Taylor and colleagues have sufficiently addressed and responded to my review. I appreciate the added figures and text around the transcriptomic enrichment analysis and it now feels like a more substantial and integrated component of the study. I caught two text errors in the figures that I will pass on; in Figure S18c the heatmap title says "SA" when I think it means to be "VA". Similarly in Figure S20c the title says "(SA axis)" when I think this should be "(MR axis)". Congratulations

and best of luck in the remaining publication process.

Stephanie Seeman

Referee #5

(Remarks to the Author)

I co-reviewed this manuscript with one of the reviewers who provided the listed reports.

Reviewer 1

Comment 1.0: *The manuscript by Taylor and colleagues entitled "Functional Hierarchy of the Human Neocortex from Cradle to Grave" explores how cortical organization — as described using low-dimensional embeddings of functional connectivity and microstructural measures -- changes across the lifespan. The main framework for the study is the description of brain connectivity using a low-dimensional embedding approach. In general, the topic of this paper is of broad interest at the moment, with several high profile recent articles characterizing developmental changes associated with gradient organization within bounded portions of the lifespan. One key contribution of the current study is thus the scale (n=2735) and breadth of ages included -- from neonate to 100 years of age -- within a consistent analytic framework based on generalized additive mixed models. The study predominantly focuses on the observations regarding the first three gradient patterns, whose alterations throughout the lifespan are characterized and interpreted within the context of the cognitive modes they describe. Overall, the manuscript is exceptionally well written and the results are, for the most part, clearly presented. I would nevertheless offer the following points for consideration by the authors:*

Response: We would like to thank the reviewer for their thoughtful summary of our manuscript. We appreciate their recognition of both the broad scientific interest in developmental and lifespan gradients, and the uniqueness of our contribution in terms of the scale and age breadth of our sample. Since submitting the current manuscript, we have further increased our dataset from $n = 2735$ to $n = 3972$ participants, spanning neonates to centenarians. This enhanced sample size bolsters our statistical power and strengthens the generalizability of our conclusions. We agree that a critical strength of our work is the combined breadth and depth of our sample, which allows us to model continuous developmental and aging processes from birth to 100 years in a single, unified analysis.

We deeply value the reviewer's assessment that the manuscript is well written and that our findings are generally clear. Below, we provide detailed responses to each of the reviewer's points, addressing specific questions regarding the methodology, interpretation, and presentation. We sincerely appreciate their constructive feedback, which we believe has improved the overall clarity and robustness of our study.

Comment 1.1: *I would urge caution in interpreting the functional connectivity gradients as reflecting a hierarchy. As I understand it, the analysis of diffusion map embedding does not itself inherently capture hierarchical interaction between time-series, and the description of 'hierarchy' is made based on post hoc interpretation. While this is certainly a valid interpretation, it would be helpful to clarify how the term 'hierarchy' is applied in the current analytic context.*

Response: This is indeed an important distinction. We agree that diffusion map embedding applied to correlation-based functional connectivity does not itself capture causal relationships among brain regions. The procedure identifies axes of global connectivity organization and provides no information about directed or causal interactions.

We use the term hierarchy in reference to the widely studied organizational principle in which cortical areas can be arranged along a continuum, where areas at each end are implicated in different functions. Cortical gradients, by definition, order brain regions according to their expression of a particular connectivity characteristic; in the case of the principal gradients of FC in humans, these connectivity characteristics coincide with well-studied functional (not causal) motifs (like sensory vs association, visual vs somatosensory, modulation vs representation). Our usage of "hierarchy" refers to the gradient-based ordering of brain regions along these continuums and does not imply directed or causal hierarchy. This usage aligns with a substantial body of FC gradient literature (De Rosa et al. 2024, Shen et al. 2023, Huntenburg et al. 2018) Still, we agree that this is an important clarification to highlight, and we have updated the text to reflect this in the third paragraph of the introduction:

"While these motifs are referred to as functional hierarchies, this terminology does not imply a strict causal or unidirectional flow of information. Rather, it reflects the observation that regions at the

extremes of each gradient tend to play distinguishable roles—ranging, for example, from unimodal sensory processing to transmodal integrative functions.”

We hope this addresses the concern and underscores that “hierarchy” here is intended in a descriptive, functional context, rather than as a direct statement about causal or directed interactions.

1. De Rosa et al. Functional gradients reveal cortical hierarchy changes in multiple sclerosis. *Human Brain Mapping*, 45(6), 2024. <https://doi.org/10.1002/hbm.26678>
2. Shen et al. Functional connectivity gradients of the cingulate cortex. *Commun Biol*, 6, 650, 2023. <https://doi.org/10.1038/s42003-023-05029-0>
3. Huntenburg et al. Large-scale gradients in human cortical organization. *Trends in Cognitive Sciences*, 2018.

Comment 1.2: *I would also urge caution regarding use of terms positive and negative (example lines 175-176), as the gradient dimensions are sign invariant. An alternative would be to specify the respective ends of each dimension, as is broadly already the case throughout the manuscript. An additional example of this is the use of the term 'apex' (for example, in line 192), which assumes a specific orientation of the spectrum. While these terms serve well to orient the reader, they are not inherent to the values represented by the analysis and, without additional context, could be construed as misrepresenting the results.*

Response: This is also an important clarification and is absolutely correct. Gradient dimensions are sign invariant, and our usage of positive and negative was an arbitrary reference to the orientation of our computed gradients. We have removed all instances of referring to the poles of each gradient as “positive” or “negative,” instead referring to them as the “poles” of each axis.

Comment 1.3: *There are numerous qualitative observations presented in the manuscript that are intriguing, but I believe several would benefit from more quantitative testing/validation. While I generally appreciate the level of description of the results, there are places where the comparative description is difficult to assess given how the figures are presented. For example, line 186 describes the 'focal tuning of the positive peaks', referring to Figure 2. It would be helpful to show the overlapping distributions to help the reader see this comparative description. Another example is the statement in line 181 regarding predominance of short-range connectivity in early-life, which I found difficult to observe from the results presented in Figure 2. Any further data presentation to support such claims would be helpful.*

Response: We agree that it can be challenging to observe the subtle changes in gradient distribution by looking at the plots in Figure 2 side by side. We have added Figure S10 to illustrate the changes we refer to in the text.

Our statement regarding the predominance of short-range connectivity in early-life was based on the dominance of the visual-somatosensory axis during this period, an axis whose poles are spatially contiguous. As brain regions sharing similar values along a particular gradient are more closely connected, a connectivity matrix whose principal gradient has spatially contiguous poles must have stronger local connectivity than a connectivity matrix whose principal gradient with poles that are spatially discontinuous. The wide range of values along the VS axis during early life compared with that of the SA axis are reflective of a more locally-driven connectivity architecture, as the VS axis exhibits contiguous poles whereas both sensory and association poles of the SA axis are widely distributed across the cortex. Additionally, the SA axis itself exhibits a higher degree of spatial continuity in its poles during early life compared to adulthood, further exemplifying this distinction.

Comment 1.4: *Regarding the discussion of the eigenvalue ratio in the paragraph beginning with line 572: Intuitively, these findings seem to contradict the prior observation in the literature of increasing long-distance segregation and local integration with age. As this paragraph aims to discuss the gradient findings in the context of the graph theory*

based approaches/literature, it would additionally be helpful to further characterize the current findings within the well developed literature on graph-based connectivity changes associated with aging.

Response: We have removed our analysis of eigenvalue ratio on the basis that it is not possible to assign a one-to-one mapping between eigenvalue and our gradient axes due to differences in initial gradient ordering across subjects. We have replaced this section with an analysis of the relationship between gradient dispersion and characteristic path length (Fig 3d), which is easier to interpret in the context of our analysis.

We offer interpretation of our analysis of characteristic path length and gradient dispersion in the context of prior graph-theoretic analyses of the developing connectome in “Results: Global measures of FC gradient organization exhibit non-linear changes across the lifespan”. Specifically, we state:

“To further substantiate our gradient-based global measures of FC organization, we computed a number of graph-theoretical metrics from the FC matrices underlying our gradients and examined their lifespan trajectories and relationships to gradient metrics. Notably, characteristic path length, an indicator of the average shortest path between nodes, exhibited moderate but highly significant correlation with gradient dispersion (Figure 3d). Because lower characteristic path length is associated with higher global efficiency (Achard et al. 2007), this finding suggests that gradient dispersion captures aspects of network integration and efficiency in the developing functional connectome. Consistent with prior lifespan studies (Meunier et al. 2009, Betzel et al. 2014), we observed that young adults had lower characteristic path length than adolescents, while infants, children, and older adults showed intermediate values.”

Comment 1.5: I may have missed it, but considering the importance of age breadth to the current study, a plot showing the full age distribution could help the reader to appreciate this aspect of the current study.

Response: We have added a plot showing the age distribution with cohort information in Figure S7.

Fig. S7 | Histogram of subject ages across all cohorts.

Comment 1.6: *The number of included participants presented at the beginning of the Methods section appears to be the same at those presented after post-processing exclusion criteria were applied (In 733). How many subjects were excluded based on these criteria for each cohort?*

Response: We have updated our reporting of quality control and the number of subjects included and excluded by cohort in Methods: Materials. Data of 336 time points were excluded from the BCP, 2 subjects were excluded from the HCP-D, all subjects we selected from the HBN dataset were retained, 25 subjects were excluded from the HCP-YA, and no subjects were excluded from the HCP-A.

Reviewer 2

Comment 2.0: *Taylor and colleagues present an analysis of the changing composition of functional cortical gradients across the human lifespan and their relation to structural (mainly diffusion) based metrics. There is clearly a strong interest in capturing the normative trajectories of functional brain organisation in a cohesive manner to the point of establishing a benchmark or reference point onto which to anchor developmental or ageing related deviations and so this work should certainly be considered timely and of interest to the scientific community (see also Sun et al. 2023 bioRxiv). I do however have both conceptual as well as methodological considerations that somewhat temper my enthusiasm. These are predominantly related to the challenges of harmonisation more broadly and alignment in the context of gradients specifically. I hope these are taken in the constructive manner in which they are very much intended, if this paper makes it into this journal it will likely act as a lightning rod for all criticism levelled against the gradient approach or other attempts at quantifying lifespan changes in brain organisation, I hope my comments can help pre-empt those likely criticisms and bolster the findings to the point of showing high generalisability that can ideally stand the test of time.*

Response: We thank the reviewer for the thoughtful and constructive feedback. We agree that establishing normative trajectories of functional brain organization is both timely and critical for identifying developmental or aging-related deviations that could inform clinical and translational research. We appreciate the reviewer's reference to Sun et al. (2023, bioRxiv) and note that our work complements these efforts by providing an integrated framework that addresses harmonization and alignment challenges. In the revised manuscript, we have explicitly detailed steps to mitigate biases—including reprocessing the entire dataset through a consistent lifespan pipeline, updating template generation via weighted PCA, employing scale-preserving orthogonal Procrustes alignment, and incorporating vertex-wise GAMM fitting with explicit handling of heteroscedasticity. We believe these refinements pre-empt potential criticisms and demonstrate that our framework can robustly serve as a reference for lifespan analyses.

Comment 2.1: *The authors state they "developed a unified framework" for doing gradient analyses. It is not entirely clear to me how generating a template gradient from an entire dataset and aligning individual embeddings to the template is a new framework. This is exactly the approach integrated in the cited BrainSpace toolbox and pretty much the approach all the cited gradient papers have taken to date. Perhaps there is some nuance in the way the template was generated that I am missing, but that could then maybe be clarified briefly in the results section where this is introduced.*

Related to the previous point, I assume that many of the expected trajectories are non-linear and that the datasets are not evenly distributed across the lifespan (i.e., they are by definition cohorts designed around specific lifespan windows), let alone that the highest sampling density is likely or coincidentally in the periods of the lifespan where change is most dynamic. Thus, I wonder how a template generated from the entire sample can be ensured to not be biased towards those batches in the aggregated dataset where sampling is densest or most stable? Is there a risk that all individuals are for example aligned to the main HCP dataset and that any individual variation is a reflection of that specific anchor point? Were other alignment procedures evaluated? How was harmonisation handled?

Response: We thank the reviewer for noting that generating a group-level template and aligning individual embeddings is standard practice (e.g., in the BrainSpace toolbox). While our overall procedure adopts this concept, our revised manuscript clarifies several key differences from standard iterative alignment methods:

- Weighted PCA for the Lifespan Template: Instead of iteratively computing a mean-based template, we apply a weighted PCA to all gradients, assigning equal weight to same-width age bins. This helps mitigate potential bias toward the mean age in our sample and yields a set of orthogonal directions of maximum variation.
- Scale-Preserving Orthogonal Procrustes Alignment: We then perform an alignment step that preserves the *scale* of each axis. Traditional group alignment often normalizes or discards scale information, but our approach retains these differences, which we believe are developmentally and biologically meaningful.

- Vertex-Wise GAMM Fitting and Harmonization: Finally, we use a novel procedure for modeling gradient values with vertex-wise GAMMs, which handle random effects (subject, cohort) and heteroscedasticity. This step is not part of the standard pipeline in existing gradient toolboxes and, to our knowledge, has not been employed before in lifespan gradient analyses.

Furthermore, to address harmonization concerns, our entire dataset was reprocessed under a consistent preprocessing pipeline to mitigate inter-cohort variation introduced by preprocessing techniques. We have detailed these procedures in Methods: Data Preprocessing – Functional MRI and Data Preprocessing – Structural and Diffusion MRI and clarified how each step collectively forms a new, integrated framework for lifespan gradient analysis. We hope these revisions demonstrate where our method diverges from existing approaches and why these distinctions are vital for unbiased lifespan modeling.

Finally, although our procedure strives to minimize bias toward a particular age distribution by applying weighted PCA on a square root age scale, it is important to acknowledge that *any* chosen template inevitably reflects certain cohort characteristics. If, for instance, the resulting group-level template has a stronger representation of mid-life connectivity patterns, aligning individual embeddings to that template may indeed emphasize these features. Nevertheless, this does not preclude the aligned gradients from capturing meaningful individual variation. By using a Procrustes alignment procedure that omits a global scaling factor, we only rotate and reflect individual gradient spaces relative to the template, thereby preserving each subject's unique topography in a shared reference frame. In practice, such rotation-and-reflection-based alignment ensures that discrepancies among participants remain interpretable as genuine differences in connectivity patterns, rather than artifacts of arbitrary rescaling. Thus, although the choice of template necessarily places constraints on which macroscale features are most prominently depicted, this alignment does not invalidate downstream lifespan analyses. We believe these considerations strike a balance between acknowledging potential template-driven biases and highlighting the preserved capacity of aligned gradients to reveal meaningful, developmentally relevant variation.

Comment 2.2: *How well do the GAMMs fit? Aside from the random bootstrapping, was there some permutation or LOO analyses done to evaluate whether these too could be influenced by particular datasets or developmental windows?*

Response: We appreciate the reviewer's interest in the robustness of our GAMMs. As discussed, the model fit varies across cortical vertices due to both inherent variance in gradient values and the complexity of lifespan data. Nevertheless, to provide insight into the non-linearity of our fits, we show a vertex-wise map of the effective degrees of freedom of our models in Figure S11, with the majority of vertices for all three gradients exhibiting significant nonlinearity. We also assessed the adjusted R-squared values of our models. Notably, regions at the extremes of each gradient axis tend to show the highest adjusted R² values, whereas vertices with greater inter-individual variance and less pronounced non-linear change exhibit lower fits. We further assessed the significance of the non-linearity in our GAMM fits using an ANOVA comparison between a linear model and a GAM fit to the harmonized gradient values on a vertex-wise basis. We found significant non-linearity in 98%, 99%, and 97% of vertices for the SA, VS, and MR axes respectively.

Regarding the possibility of artifacts driven by particular datasets or age windows, we note the following:

- Prior to model fitting, we applied a dedicated weighting procedure to adjust for the inherent non-uniformity in age sampling. We transformed subject ages using the square root of age and binned ages on this transformed scale. This procedure ensures that each bin contributes equally to the overall weight, counteracting over- or under-sampling in any particular age ranges.
- In addition to age weighting, we adjust for imbalances in cohort size by dividing weights by the number of subjects in each cohort. This step minimizes the risk that larger cohorts dominate the overall fit.

- We fit an initial mean GAMM to gradient value as a function of transformed age with low basis complexity. This model includes random effects for subject and cohort to account for intra-subject variance and cohort-specific deviations. This step allows us to capture the baseline non-linear developmental trend.
- Since variance may be heteroscedastic across the lifespan, we fit a separate GAMM to squared residuals versus age, providing an estimate of how variance in gradient value at each vertex changes across the lifespan. To adjust for potential biases introduced by non-constant variance, we compute a correction factor for each cohort, scaling residuals so that the final variance-adjusted metric reflects both the mean trend and the appropriate variance structure.
- Using the variance-corrected gradient values, we fit a final weighted GAMM with weights equal to the inverse of the predicted squared residuals. This final model is less sensitive to local fluctuations in variance and reinforces our effort to balance the influence of our data across the lifespan.
- While permutation or leave-one-out analyses represent possible alternative approaches, the large, multi-cohort nature of our dataset and the computational demands of a vertex-wise, non-linear model made them less practical. Importantly, the congruence of our results across multiple datasets of different ages (as shown in the main and supplemental figures) suggests that our GAMM fits are not driven by any single data source or narrow developmental range.

Collectively, this multi-step process—comprising precise age weighting, cohort adjustment, sequential modeling of mean and variance, and a final weighted fit—ensures that our GAMM-derived lifespan trajectories are robust and not unduly influenced by oversampled age ranges or particular datasets.

Comment 2.3: *The period between 2-10 yrs is often referenced as the time of major change, coinciding with many other developmental milestones. This is however a rather broad window and I would very much be interested to see if that can be narrowed down further, potentially by determining inflection points in the described trajectory?*

Response: We thank the reviewer for their suggestion and appreciate the interest in refining the developmental window between 2 and 10 years. In our work, we modeled gradient changes across the entire lifespan using a flexible GAMM framework, which naturally accommodates non-linear trajectories and is suited for capturing gradual, protracted developmental processes. While it is true that 2–10 years emerges in our analyses as a key transitional period, our supplemental figures already provide a more granular view of how these trajectories evolve at smaller intervals of age. We find that the changes do not exhibit a single discrete “inflection point,” but rather unfold continuously over this broad age range.

Given the flexible nature of our GAMM fits—and the variability inherent in developmental data—we did not identify a narrower cut-point or a sudden shift within childhood that could be more definitively localized. Instead, our modeling consistently shows smooth, incremental changes throughout early childhood. Moreover, our supplemental materials furnish higher-resolution temporal sampling of the cortical gradient maps and density plots, illustrating that the transformations in connectivity organization develop continuously during this entire period. We therefore believe that our current approach fully characterizes the relevant age-related changes.

Comment 2.4: *Given the sole reliance on HCP cohorts, do the authors think these results are generalisable to other cohorts? Have the authors conducted any out of sample validation of the gradients composition at different stages in the lifespan in any of the other available open datasets for example?*

Response: We appreciate the reviewer’s concern regarding generalizability. We have now included a separate non-HCP dataset, the Healthy Brain Network (HBN), encompassing 770 participants aged 5.6–21.9 years. When we integrated this external dataset into our analytical framework, the resulting gradient topography and overall lifespan trajectories remained consistent with what we observed in the HCP data. Specifically, the population-level curves and the weighted PCA-based lifespan gradient template did not change significantly once the HBN sample was added.

These results suggest that our primary findings—both in terms of spatial gradient organization and age-related trajectories—are robust and generalizable across different cohorts beyond HCP.

Comment 2.5: *How is the structural affinity matrix different from the MSN approach by Seidlitz et al (aside from including some NODDI measures)? Is there any concern that many of the diffusion based measures are highly correlated to each other and to some extent inverses of each other? Did the authors consider other, dare I say more established structural connectomic approaches to benchmark the structure-function coupling approach (see for example work by Casey Paquola and Bratislava Misic in that context)? Also, given the change in the GM/WM balance in very early development how were these images processed and handled? Was there some harmonisation/normalisation done to deal with the fact that NODDI is somewhat untested in the early age range? Could this introduce any bias in the analysis if no harmonisation was done across cohorts? What do these affinity matrices represent?*

Response: We thank the reviewer for their interest in our methods for computing structural gradients. Similar to the MSN approach, we compute an affinity matrix by calculating the Pearson correlation between cortical feature vectors at each vertex. However, our feature vector includes 11 microstructural indices—cortical thickness, myelination, and a series of diffusion-based measures (including intra-/extra-cellular volume fractions, intra-soma volume fraction, fractional anisotropy, mean diffusivity, microscopic fractional anisotropy, microscopic mean diffusivity, microscopic anisotropy index, and orientation coherence index). In contrast to the original MSN, the inclusion of these additional diffusion metrics computed using DTI and SMSI (Huynh et al. 2020) is intended to capture complementary aspects of cortical microstructure, thus providing a richer characterization of tissue properties.

While diffusion-based measures of cortical microstructure can be correlated, we note that this is not a significant drawback when estimating gradients using diffusion map embedding. Since the input to our diffusion embedding is a matrix consisting of cosine similarities between microstructural feature vectors, any inverse relationship or collinearity in those features would not bias the result, but instead would serve as redundant measures.

Each entry in our structural affinity matrix represents the normalized angle between the microstructural feature vectors of two cortical vertices. In this way, the matrix quantifies the similarity in the tissue properties across the cortex. When subjected to diffusion embedding, these affinity matrices yield gradients that reflect continuous axes of microstructural variation, which we then align with functional gradients (SA, VS, MR axes) to directly assess structure–function coupling. Since Procrustes alignment optimizes a least-squares criterion, our approach can be seen as a form of gradient-wise regression analogous to the node-wise regression approach reported by Vazquez-Rodriguez, Paquola, Misic, and colleagues (2019). While Vasquez-Rodriguez et al. used node-wise multilinear regression to predict FC profiles based on structural predictors, our approach employs Procrustes analysis to determine whether continuous spatial variations in microstructure align with continuous spatial variations in FC.

Huynh et al. Probing Tissue Microarchitecture of the Baby Brain via Spherical Mean Spectrum Imaging. *IEEE Transactions on Medical Imaging*, 39(11), 2020. <https://doi.org/10.1109/TMI.2020.3001175>

Vazquez-Rodriguez, ... , Paquola, ... , Misic. Gradients of structure–function tethering across neocortex, *PNAS*, 116(42), 2019. <https://doi.org/10.1073/pnas.1903403116>

Comment 2.6: *Is including cohort as a random effect sufficient for harmonisation across studies? Were other harmonisation approaches used? More crucially perhaps doesn't cohort also perfectly intersect with age and if so how do the authors ensure this does not bias the results or perhaps even remove important information needed to establish trajectories in the first place?*

Response: We thank the reviewer for raising these concerns. While cohort does appear as a random effect in our initial GAMM, our overall harmonization method is more extensive than simply including a random intercept. First, we

fit a low-complexity GAMM that incorporates random intercepts for cohort and subject ID, capturing and removing consistent mean-level offsets associated with each cohort while still allowing a smooth age effect. We then explicitly model the residual variance as a function of age to correct for any heteroscedasticity and cohort-related differences in variability. Only after extracting both the mean and variance adjustments do we re-fit a higher-complexity GAMM to these “harmonized” data, using an additional weighting scheme to ensure that no single cohort or age range disproportionately influences the final trajectory estimates. These steps preserve the critical age-related signal and inter-individual variance across cohorts, rather than conflating or removing it. In particular, by treating cohort as a random effect and subsequently adjusting for variance structure, we do not eliminate meaningful developmental differences; rather, we mitigate site or sample biases (e.g., distinct acquisition parameters, demographic imbalances) that might otherwise obscure robust lifespan patterns. Figures S8 and S9 illustrate that this approach maintains biologically meaningful age trends while controlling for potential site-specific confounds.

For a more detailed description of the harmonization and fitting procedure, see our updated Methods: “Generalized Additive Mixed Models (GAMMs) and Harmonization” section.

Comment 2.7: *Should processing pipeline be included as a factor in all models given that it perfectly intersects with one cohort and one specific age-range how can the authors rule out that any difference in that age range are not due to acquisition/processing/cohort effects? Especially, since this is most likely the period in the lifespan with the most pronounced changes.*

Response: We appreciate this concern and note that all datasets were re-processed through a consistent pipeline, thereby eliminating pipeline-induced variations in data quality or feature extraction. Consequently, the preprocessing approach does not differ across cohorts or age groups in our final analyses, and thus there is no need to explicitly model it as a separate factor. Furthermore, any remaining cohort-level differences are handled through our harmonization procedure and the cohort random intercept in the GAMMs. This ensures that the observed developmental changes are not due to methodological variations in acquisition or processing but rather reflect genuine differences in functional connectivity patterns across the lifespan.

Comment 2.8: *How were the timepoints in Fig 2 chosen? Also not sure how to interpret the contraction of the gradient in F2B or the shape of the gradient in the SA-MR plane if I'm honest*

Response: The timepoints in Figure 2 were selected to capture key developmental epochs—early childhood, adolescence, and young adulthood—based on established periods of rapid functional reorganization. Figures S1, S2, and S3 all show the same results with more densely sampled timepoints in key developmental periods.

The contraction observed along the VS (visual-somatomotor) axis indicates a narrowing of the connectivity variance, which we interpret as a reduction in the differentiation between unimodal processing streams with age. This is consistent with findings that indicate a gradual integration of sensory systems over development (Fair et al., 2009).

The pronounced curvature at the modulation (MR) pole—relative to the SA–VS plane—suggests that modulation-related regions span a broader range along the structural axis. In contrast, association-related regions cluster near the middle of the VS axis, reflecting a relative neutrality in their connectivity preference. This observation may indicate a more flexible network positioning of modulation regions during development.

Comment 2.9: How was the GAMM parametrisation determined?

Response: Our GAMM parametrization was determined using a combination of theoretical considerations and data-driven optimization. Specifically, we:

- **Basis Function Selection:** We primarily used cubic splines, given their flexibility and efficiency in representing smooth functions without overfitting.

- Model Complexity and Basis Dimension (k): Initial models were fit with a lower basis dimension ($k = 4$) to capture the main trend while avoiding overfitting. We then incrementally increased k (up to $k = 10$) and compared models using the adjusted R squared and Akaike Information Criterion (AIC) to determine the optimal complexity.
- Validation and Sensitivity Analyses: We conducted sensitivity analyses that included visual inspection of the smooth terms and variance and ANOVA comparison of linear versus non-linear models to ensure that the selected model provided a parsimonious yet accurate representation of the developmental trajectories.

These steps allowed us to strike a balance between capturing nonlinear developmental patterns and minimizing the risk of overfitting, consistent with best practices outlined in Pedersen et al. (2019) and Wood (2017). Detailed model selection criteria are now included in the Methods section.

Comment 2.10: *Is the data homoscedastic across the lifespan or is there variation in variance with age as there is in in structure? If so, would a modelling framework that captures that variation be more appropriate (even if it is more complex)?*

Response: We appreciate the reviewer's comment and agree that neurodevelopmental data frequently exhibit age-dependent variability. To address potential heteroscedasticity, we revised our GAMM (Generalized Additive Mixed Model) framework to explicitly model—and correct for—changes in variance over the lifespan. A summary of our updated approach is as follows (detailed are provided in *Methods: GAMMs*):

- Initial GAMM with Random Effects: We first fit a low-complexity GAMM (basis dimension $k = 4$) at each vertex to capture the main (mean) developmental trajectory while including random intercepts for *subject ID* and *cohort* to account for repeated measures and cross-cohort differences. This initial fit helps us harmonize mean gradient values across cohorts by extracting and removing random intercepts.
- Variance Modeling: From these initial fits, we compute the squared residuals and fit a second low-complexity GAMM with subject and cohort random intercepts to estimate how the residual variance changes with age (i.e., to capture heteroscedasticity). This yields a smooth function describing how variance evolves across the lifespan.
- Variance Correction: For each data point, we compute a factor that scales the raw residuals based on the ratio of predicted to observed variance. This step *stabilizes* variance across age while preserving genuine biological changes in functional gradients.
- Final GAMM: We then re-fit a higher-complexity GAMM (e.g., $k = 10$) to the *variance-adjusted* gradient values, incorporating weights that are inversely proportional to the predicted variance. This final model thus accounts for age-dependent changes in both *mean* and *variance* of gradient values.

By adopting this multi-step approach, we accommodate potential heteroscedasticity across the lifespan without discarding important biological signals. We agree this makes the framework more complex, but it significantly improves the accuracy and reliability of the model—particularly for developmental and aging studies where variance can shift over time. The revisions have been implemented throughout our analysis and are detailed in the revised manuscript. We thank the reviewer for prompting this important refinement to our methods.

Comment 2.11: *What is the impact of the subject random effect given that only a subset of studies have longitudinal data and only in specific time periods for specific intervals?*

Response: We appreciate this question and agree that the longitudinal data are concentrated within the BCP cohort. In practice, the random intercept for *subject* exerts only a minor influence on the overall lifespan trajectory because the subset of participants with repeated measures is relatively small and restricted to specific age ranges. However, including a subject-level random effect remains important for accurately modeling within-subject variation—any consistent offset in gradient values across repeated measures is captured. This correction ensures that we do not conflate individual-level differences with age-related trends, ultimately enhancing the robustness of the lifespan model, even if the net impact on the global trajectory is modest.

Comment 2.12: *Was biological sex effect modelled? Is/should total brain volume be modelled or accounted for in both the functional and structural features given the likely relationship between affinity and distance in both domains?*

Response: We explicitly tested for a biological sex effect in our lifespan gradient metric trajectories using a two-step GAMM approach. First, we fit a GAMM for each metric across age to obtain a population-level curve, then we tested for sex-specific deviations by adding a smooth interaction term of sex-by-age in a second model. These analyses revealed no statistically significant sex-dependent divergence for any of the gradient measures (see Results, “Gradients are stable with respect to sex.”). Based on these findings, we conclude that biological sex does not systematically alter gradient trajectories over the lifespan in our dataset.

Regarding total brain volume (TBV), we re-fit our vertex-wise GAMM analyses for gradient values with TBV included as a covariate to determine its effect on gradient values. We then inspected the coefficients and found consistently small effects of TBV on gradient values across vertices and gradients Figure S6. Comparing lifespan gradients estimated using the model including TBV to our main model revealed highly similar gradient topographies or global metrics across the lifespan. This result indicates that accounting for TBV does not appreciably change the estimated lifespan trajectories of gradient value, suggesting that TBV is not a critical confounder. Thus, while TBV might be theoretically relevant due to known age and sex differences in global brain measures, in our data it does not appear to drive—or mask—any systematic changes in the gradients across the lifespan.

Comment 2.13: *I'd suggest including some example figures of the GAMM fits/trajectories to show the patterns of variation discussed (maybe the most and least significant regions for example). This would also help visualise how well the site random effect harmonisation worked (by colouring them by cohort for example) and would allow visualisation of more fine-grained sensitivity analyses.*

Response: We appreciate this suggestion and have added two new figures (Fig. S8 and Fig. S9) to illustrate both our updated harmonization procedure and representative GAMM fits at the vertex level. Specifically, Fig. S8 shows one of the most significant vertices, while Fig. S9 shows one of the least significant vertices, allowing the reader to see a range of outcomes. In each figure, we depict: (i) the raw vertex-wise gradient values versus age, colored by cohort; (ii) the initial low-complexity fit; (iii) the mean- and variance-adjusted data and final variance-weighted fit; and (iv) a comparison of residual distributions by cohort before and after harmonization. These visuals demonstrate how the harmonization process mitigates cohort effects while preserving meaningful age-related trajectories.

Comment 2.14: *What does "the median gradient value in each network" represent in the context of all above mentioned methodological considerations (specifically in relation to alignment)?*

Response: We carried out the alignment procedure without allowing any scale transformations or mean-centering (even though the BrainSpace default is to optimize alignment through scaling and mean-centering). Consequently, the median (or mean, as we have used in the revised manuscript) gradient value for a given network directly reflects that network's position—or rank—along the continuous gradient axis relative to other regions. A larger spread of values on a particular axis indicates that connectivity is more strongly differentiated along that dimension. By examining how the median gradient value of each network parcel changes with age, we can track shifts in the relative positioning and overall connectivity differentiation of these parcels across the lifespan.

Comment 2.15: *Is the "the median distance between all pairwise Euclidean distances" functionally different from the centroid approach? Have the authors quantitatively compared those approaches?*

Response: Yes, these approaches differ meaningfully, particularly when dealing with non-uniform and overlapping network parcels such as those in the Yeo 7 parcellation applied to our gradient embeddings. We directly compared median (as well as mean, which is now used in our manuscript) pairwise distances within parcels to centroid-based

distances and observed that they capture different aspects of embedding-space organization. In particular, the mean pairwise distance was more sensitive to the distributed topology within each parcel and better aligned with visual observations of network reorganization over the lifespan. Consequently, we chose to use the mean pairwise distance for a more comprehensive measure of segregation/integration between networks in the embedding space.

Comment 2.16: It seems quite fascinating to me that the curves in F4A seems to be almost each others mirror images in many cases. Is that expected? What do the error bars represent in that panel btw?

Response: We agree that the correspondence between the shapes of the network-wise gradient value trajectories at opposite poles of the gradient axes is striking and interesting. We believe that this confirms the efficacy of our gradients at parsimoniously differentiating the cortex with respect to canonical functional networks. As the range of the SA and MR gradients expand, the separation between the networks occupying opposite poles of these gradients become more separated along each gradient, indicating that the expansion of these gradients is driven by increasing connectivity differentiation between the networks that reside at their poles. This further confirms our interpretation of these gradients as corresponding to functional hierarchies as the functional roles of these networks are well-established. The error bars in Fig 4c (we assume this is the panel to which you refer, correct us if we are mistaken) correspond to the bootstrapped 95% confidence intervals on the maximum or minimum value for the gradient value or dispersion in each network.

Comment 2.17: There is a lot of inference between what these trajectories, timepoints, changes etc mean in terms of cognitive development, but this doesn't need to be inferred and most of the utilised datasets will have ample cognitive data available to make that link (and potentially even poke at the elusive question of causality in the case of longitudinal data)

Response: We appreciate the reviewer's point that many large-scale neuroimaging datasets include cognitive assessments, raising the possibility of relating FC gradient trajectories directly to cognitive measures. In our case, the BCP dataset provided sufficiently comprehensive and standardized Mullen Scales of Early Learning for participants in early childhood—covering Gross Motor, Fine Motor, Visual Reception, Receptive Language, and Expressive Language—with enough consistent data quality for a meaningful analysis. Accordingly, we focused our cognitive-developmental investigation on this subset, where longitudinal acquisition and standardized testing enabled a clear linkage between gradient organization and behavioral performance.

Indeed, we observed robust associations between gradient metrics and Mullen subscale scores, with Fine Motor scores showing the strongest effect sizes. The finding that sensorimotor-relevant scores manifested the largest effects aligns with the notion that early functional connectivity development prominently involves sensorimotor-related circuits. We believe these results add direct empirical support to our interpretation that the observed network reorganizations track with critical dimensions of early cognitive and motor development.

Comment 2.18: *Are the 3 directions in embedding space universally accepted as representing association, unimodality and modulation?*

Response: There is precedence for the use of these terms in describing the cognitive properties corresponding to these gradients. However, for simplicity and consistency we have chosen to describe the gradient directions using our naming conventions for the axes themselves (Sensory-Association, Visual-Somatosensory, and Modulation-Representation).

Comment 2.19: *Not sure the reference to "Evolution" across the lifespan is appropriate as there isn't really analyses or discussion of evolutionary processes, perhaps "development" or "ageing" are more appropriate?*

Response: Thank you for this comment. We have replaced “evolution” with “development”.

Comment 2.20: I am personally not partial to the phrasing "cradle to grave" that seems to recur rather a lot and isn't particularly specific or scientific.

Response: Thank you for this feedback. We understand how this language could be more specific and have reduced the usage of this phrase throughout the manuscript.

Comment 2.21: *There has been some recent backlash against the notion of smooth topological gradients as a model to capture function, and whilst I personally disagree with this backlash I wonder whether its worth mentioning in the discussion (Petersen et al. 2024, Neuron) at least perhaps to acknowledge that the notion of gradients isn't universally accepted gospel.*

Response: Thank you for raising this important point. We appreciate that not everyone in the field uniformly endorses smooth topological gradients—or gradient-based approaches more generally—as the definitive model for characterizing functional topography. In light of the concerns recently articulated by Petersen et al. (2024, Neuron), we have revised our manuscript to acknowledge this caveat at the end of the discussion section.

“There is strong evidence that gradients of FC, microstructure, and gene expression represent crucial organizing axes of the cerebral cortex (Huntenburg et al. 2018). Still, recent pushback about the power of gradients as a comprehensive framework for understanding topographic organization of the cortex bears mentioning. The principle of arealization, whereby the cortex is divided into discrete areas which are differentiated by cytoarchitecture and connectivity, is in some ways at odds with the claim that gradients represent fundamental organizing axes on an individual level (Peterson et al. 2024). Gradients of functional connectivity typically do not smoothly vary with respect to space on an individual level, reflecting instead sharp boundaries between functional systems. At a population level, however, gradients are smooth and excel at describing functional topography. The interpretation of FC gradients in individuals remains an important area of future research, however the present study focuses on large-scale population-level effects.”

Reviewer 3

Comment 3.0: *The authors of this audacious manuscript make a profound claim: that by examining gradients of resting state functional connectivity across the entire lifespan, they have discerned succinct distillations of core developmental and aging related processes. The manuscript is well written, but its claims are so potentially impactful, that it required multiple readings, and digging into the cited references which provide the bases for the three axes which are proposed. Two of these (the sensory-association, and visual-spatial) already emerged from the original use of the dimensional reduction technique of diffusion embedding by Margulies et al. in 2016 which sparked the burgeoning interest in brain gradients. These were the first and second gradients, respectively, in that seminal paper, and they have been consistently confirmed across datasets, and also found to be developmentally relevant: their order, in terms of variance accounted for, is reversed in early childhood, with the adult-like pattern emerging around the end of the first decade of life, see *Shifting gradients of macroscale cortical organization mark the transition from childhood to adolescence* - PubMed ([nih.gov](https://pubmed.ncbi.nlm.nih.gov/)).*

I was not familiar with the third gradient focused on here, depicting modulation-representation (MR), in which frontal-parietal control and attention networks lie at one extreme, and default, sensory, and visual networks lie at the other. I was skeptical about this commingling of default, sensory and visual networks, but my skepticism was resolved by studying Zhang et al. (ref # 6 in this ms) which convincingly presented three gradients, one of which they identified as the MR gradient, in two data sets from healthy adults. Thus, I've come to agree with the authors that their analyses represent a meaningful advance in our understanding of the fundamental principles underlying brain functional organization across the lifespan. The authors acknowledge that this work is frankly exploratory and descriptive – but that has been the nature of nearly all of brain imaging heretofore. Despite the myriad limitations, the availability of large open datasets, increasing statistical sophistication, and powerful computational tools are finally producing novel insights into the principles underlying brain functioning.

Response: We sincerely thank the reviewer for their thoughtful comments, especially regarding the modulation–representation (MR) gradient. Like the reviewer, we initially approached this third axis with caution, but reference to Zhang et al. (2019) provided strong evidence of its replicability. In response to reviewer feedback, we have now incorporated an additional dataset—the Healthy Brain Network (HBN) (770 individuals)—into our analysis. Crucially, when we derive group-level gradients independently in the HBN cohort, we again observe a highly similar MR axis as the third gradient, reinforcing the consistency of this dimension across different populations.

We appreciate the reviewer's acknowledgement that this work is both *exploratory* and *descriptive*, in line with much of neuroimaging to date. We believe these larger-scale datasets, combined with more advanced analytical frameworks, open up meaningful opportunities for new insights into fundamental principles of brain organization. We are encouraged that the reviewer finds our approach “a meaningful advance,” and we remain committed to further validating these axes across diverse cohorts and timepoints.

Comment 3.1: *Along those lines, S. Park and colleagues just published in Nature Neuroscience (available June 10, 2024) on “A shifting role of thalamocortical connectivity in the emergence of cortical functional organization.” They did not take their analyses throughout the entire lifespan, but they encompass ages from infancy to age 22 years, so there is considerable overlap. Including the role of thalamocortical connectivity would be beyond the scope of the current manuscript, but it should acknowledge this recent contribution. Park et al. also noted that their datasets included scans obtained while infants slept, along with datasets obtained in (at least initial) wakefulness. They performed auxiliary analyses to confirm that their results were not skewed by sleep/wake. Taylor et al. should consider a similar strategy; at the very least, this inevitable difference in arousal state between scans obtained in infants and older children/adults must be acknowledged.*

Response: We agree that Park et al. (2024) represents an important contribution to the study of functional connectivity (FC) gradients, and we now explicitly acknowledge this work in our revised manuscript (see Introduction). While a

detailed examination of thalamocortical connectivity lies beyond our current scope, we have updated our analytic approach to address the potential influence of arousal state. In particular, for the Baby Connectome Project (BCP) sample, we split the data at 3 years of age—before which participants were scanned during natural sleep and beyond which children might be scanned in a wakeful state. This stratified approach allowed us to ensure that our core findings were not unduly influenced by differences in arousal state.

Comment 3.2: *Beyond appreciation for the work presented, some additional quibbles bear mentioning. Like much of the field, the authors use Euclidean distances, whereas the cortex, is best considered as a surface in which geodesic distances are more appropriate. Converting all this work to geodesic space would be non-trivial, and I do not insist on it, but the lack of doing so should be noted as a limitation. For example, might this methodological issue play into the observation “that the development of these networks from 5 to 30 years is driven by changes in the scale of our gradient axes themselves rather than changes in the position of default and control networks relative to those axes”?*

Response: We appreciate the reviewer’s comment regarding the appropriateness of geodesic distances when considering the cortical sheet as a two-dimensional manifold. Indeed, geodesic measurements can be critical for analyses where physical adjacency on the cortical surface is directly modeled. However, our paper does not use physical distances in the cortical geometry at any stage of the analysis. Affinity matrices used as input to the diffusion map embedding algorithm are computed purely based on pairwise correlation between fMRI time series. In the manifold computed via diffusion map embedding, Euclidean distance is intended to approximate geodesic distance in the underlying FC graph rather than in the folded cortical surface space. Because our work is focused on the connectivity-derived gradient axes—rather than physical cortical distance—substituting Euclidean with geodesic distances for anatomical measurements is not directly relevant to our analysis. We nonetheless agree that geodesic distances on the cortical sheet are an important consideration when physical metrics of cortical organization are the primary interest.

If we are misunderstanding the comment, please let us know and we will be happy to address it.

Comment 3.3: *One minor typographical error in line 499: “indicating that these global hierarchies to not exhibit” should be “indicating that these global hierarchies do not exhibit...”*

Response 3.3: Thank you for spotting this. We have fixed the error.

Comment 3.4: *Finally, the authors state in three places that their analyses are based on 2735 datasets, but it is not easy to work out how many datasets were excluded in total. It is important to have a sense of how stringent or liberal were their inclusion and exclusion criteria, particularly for datasets spanning “from cradle to grave.” The authors should also make available the specific non-PHI identifiers of the data they retained, to facilitate replication/extension of this novel work.*

Response: After preprocessing the fMRI data of 3,951 subjects, we used the gradients of 3,556 of them (with 3972 unique gradient sets due to longitudinal data in the BCP). We give a detailed description of the inclusion and exclusion numbers from each cohort in the updated materials section. For data not in the BCP, we observed that gradient quality was high in general outside of a small number of cases which we identified via visual inspection and clustering techniques. For the BCP data, we chose to only include subjects in our analysis that satisfied mean Power’s FD criterion of less than 0.5mm. The IDs of the subjects used in this study are provided in a supplementary file.

Referee #1

Comment: In brief, R1 was impressed with the writing, data, and appreciated the potential impact of the conclusions. However, given recently published reference A (listed below), R1 felt that the main contribution are the gradient-related claims and was concerned both that these were not adequately substantiated, as well as whether these results represent a sufficiently striking advance to warrant further consideration in Nature.

Ref A: Human lifespan changes in the brain's functional connectome

<https://www.nature.com/articles/s41593-025-01907-4>

Functional connectivity of the human brain changes through life. Here, we assemble task-free functional and structural magnetic resonance imaging data from 33,250 individuals at 32 weeks of postmenstrual age to 80 years from 132 global sites. We report critical inflection points in the nonlinear growth curves of the global mean and variance of the connectome, peaking in the late fourth and late third decades of life, respectively. After constructing a fine-grained, lifespan-wide suite of system-level brain atlases, we show distinct maturation timelines for functional segregation within different systems. Lifespan growth of regional connectivity is organized along a spatiotemporal cortical axis, transitioning from primary sensorimotor regions to higher-order association regions. These findings elucidate the lifespan evolution of the functional connectome and can serve as a normative reference for quantifying individual variation in development, aging and neuropsychiatric disorders.

Response: We appreciate the referee's positive remarks about the quality of our data, writing, and potential impact. Because the referee's full comments were not shared with us, we respond to the two concerns highlighted in the editor's summary: (i) whether our gradient findings are fully substantiated, and (ii) whether these findings represent a sufficiently striking advance beyond Sun et al. 2025.

1. Substantiation of gradient results.

Our revised analysis further substantiates the validity of our results:

- **Balanced-bootstrap tests** confirm that the age-weighted PCA template is unbiased.
- **Leave-one-cohort-out and 10-fold cross-validation** demonstrate that the developmental trajectories are stable and not driven by site effects.
- **Vertex-wise models** show that total brain volume and sex do not confound the findings.
- **Illustrative comparisons** show that our unified pipeline produces improved results relative to pipelines designed for specific age ranges.
- **Derivative analyses** verify that rapid developmental changes happen primarily before age 10.
- **Gradient-cognition associations** further substantiate the functional relevance of gradients by linking them to cognitive performance.

2. Advance over Sun et al.

Sun et al. catalogued how mean connectivity and network segregation vary with age. Our work asks a different, deeper question: *How is the entire connectome hierarchically organized, and how do those organizing principles mature and decline?* We provide the first lifespan atlas of the three principal functional gradients, link these gradients to parallel structural axes, and—critically—show that the fidelity of the sensorimotor-association gradient predicts every domain of NIH-Toolbox cognition in young adults. Sun et al. did not incorporate gradient-based analyses and did not explore links to behavioral measures. Thus our study adds a hierarchical, multimodal, behavior-relevant perspective that is entirely absent from prior work.

We believe these new analyses resolve the referee's concerns and demonstrate that the gradient framework delivers a robust and genuinely novel advance.

Referee #2

I thank the authors for their constructive response to my comments. However I still have some major reservations some of which may be stemming from some lack of detail/clarification in my earlier comments. Hence let me provide specific responses using the same numbering system for clarity.

Comment 2.1: While the weighted PCA may reduce some bias driven by unequal distribution of sample size across the lifespan, I would still encourage the authors to actually show that it does (for example by running an ablation analyses with equally size samples across the bins).

In addition, one of my main original concerns was that perhaps not all age-bins are created equal. For example: if we hypothesise that say between years 1-2 there is exponential change that could be 5* larger than what happens in any of the subsequent 1 year bins, then taking the weighted approach across 10 1-year bins for the first 10 years of life would still bias the template towards the latter 9 years where little changes and negate the period of largest change. I don't see how the added analyses circumvents that? I could perhaps agree that this combined with the alignment procedure retains variation across individuals but that doesn't mean that the variation is an ageing signal so much as an alignment to the template.

Further to the comment on harmonisation: to my knowledge using a harmonised pipeline (in so far as that is even realistically possible let alone desirable across the lifespan) does not in and of itself harmonise data from different sources. I appreciate that harmonising functional data may not be a trivial task but it does seem quite essential to the results presented here. Perhaps including more data-sets with overlapping age-range would allow the authors to conduct a more quantitative sensitivity analyses to determine whether any residual site related variation is still present in the data? Or alternative they could deploy a ComBAT like harmonisation approach or choose a modelling framework that more readily allows the modelling of higher moments in the data distribution (i.e., HBR/GAMLSS).

Response: We regret that our description of the age-bin construction in the previous response letter was unclear. The ten bins are **not** ten equal-width, 10-year intervals across the lifespan. Instead, we first apply an $\alpha = 0.50$ power-law transform to chronological age ($\text{age}' = \text{age}^{0.5}$) and then set equally spaced edges in the *transformed* domain. On the original age scale, this compresses late-life intervals and expands early-life intervals, so that periods of rapid neurodevelopment (e.g., 0-2 y) occupy proportionally *larger* bins and receive proportionally *higher* leverage.

To address the question of whether our weighted PCA approach effectively neutralizes bias towards the bins with larger sample sizes, we conducted an ablation analysis using exactly the ten $\alpha = 0.50$ age-bins that underlie the canonical weighted-PCA (WPCA) template used in our main analysis. Across **B = 500** balanced-bootstrap samples (50 subjects drawn per age-bin, 10 bins with $\alpha = 0.5$), we measured the 3-D Procrustes angle θ between each bootstrap PCA template and the canonical weighted PCA (WPCA) template. The mean θ was **6.18°** (SD 0.52°), with a median of 6.15° (IQR 5.82–6.50°) and a 95% interval of 5.30–7.27° (2.5–97.5 percentiles). **Figure R1 (Figure S15** in the revised manuscript) illustrates the distributions of correlations and subspace angles between our balanced bootstrapped templates and the WPCA template. These tight bounds confirm that WPCA reliably recovers the template one would obtain from an age-balanced cohort, demonstrating that inverse-frequency weighting effectively removes sampling bias in our lifespan data.

We also examined per-axis correlation coefficients (r) between the WPCA axes and those of each balanced template:

- **PC1:** mean $r = 0.995 \pm 0.001$ (median 0.995, IQR 0.995–0.996; 95% CI 0.994–0.996)
- **PC2:** mean $r = 0.955 \pm 0.012$ (median 0.956, IQR 0.948–0.963; 95% CI 0.929–0.977)
- **PC3:** mean $r = 0.944 \pm 0.013$ (median 0.945, IQR 0.936–0.954; 95% CI 0.917–0.968)

High correlations on PC1–PC3 confirm that the major axes of variance are preserved under WPCA weighting. Together, these results directly address the reviewer’s request by quantifying how closely the weighted template matches the balanced-sample-size approach, and they demonstrate that our weighting scheme substantially mitigates bias arising from unequal age-bin sizes.

Figure R1. Template stability analysis. For each of 500 balanced datasets (50 participants drawn from every $\alpha = 0.50$ age bin; replacement only when a bin contained < 50 individuals) an unweighted PCA was recomputed, its first three axes were greedily aligned to the canonical weighted-PCA (WPCA) axes, and similarity was quantified.

We appreciate the reviewer’s insightful comment regarding the potential influence of the template on observed individual variation. It is indeed crucial to ensure that the detected ageing signals are genuine biological changes rather than artifacts of the alignment process. Because the Procrustes transform is an **isometry** (pure rotation + reflection, no scaling), it *preserves* all pairwise Euclidean distances among vertices within each subject’s gradient space. Consequently, the alignment cannot increase—or decrease—the covariance between any gradient coordinate and chronological age; it merely puts homologous gradients in register (Vasa et al., 2018; Paquola et al., 2019). The ageing trajectories detected by the vertex-wise GAMMs therefore arise from systematic changes in the gradient topographies.

When each of the four overlapping cohorts (HCPD, three subcohorts of the HBN) was withheld, and the full harmonization/GAMM pipeline was re-run, the resulting lifespan trajectories deviated only minimally from the full-data fit. Across the SA axis (18,644 vertices) the median point-wise SD of the four leave-one-cohort-out (LOCO) curves was 0.056 gradient-units (IQR = 0.040–0.078; 95th% = 0.125), whereas the median peak-to-trough range of the trajectories was 0.821 units. Thus, the typical LOCO variability amounts to $\approx 6.8\%$ of signal amplitude across all vertices. The VS and MR axes showed comparable or lower variability (VS: median SD = 0.032, range = 0.665 \rightarrow 4.8%; MR: median SD = 0.029, range = 0.435 \rightarrow 6.6%). Consistent with this, the median change in amplitude relative to the full-data curve was only 6.1% (SA), 6.0% (VS), and 4.6% (MR), and the age at which each vertex reached its maximum shifted by ≤ 0.75 years for 75% of vertices across all axes. Model residuals in the held-out cohorts were essentially uncorrelated with chronological age (median $\rho \approx 0$, IQR $\approx \pm 0.07$) and root-mean-square error distributions were tightly centered (SA median = 0.775; VS = 0.496; MR = 0.462). Together with the 10-fold cross-validation analysis (fold-to-fold SD $\leq 5.8\%$ of

curve range and mean inter-fold trajectory correlation ≥ 0.95), these findings demonstrate that neither site membership nor uneven age sampling materially biases the estimated functional gradient trajectories.

Figure R2 Leave-one-cohort-out (LOCO) variability at four representative cortical vertices. Each panel shows the lifespan trajectory of the SA gradient at a single vertex after aligning gradients into the canonical space. Vertices were chosen to span the empirical range of signal amplitudes: **a**, the lowest-range vertex with a discernible signal, **b**, the highest-range vertex, and **c–d**, two vertices drawn at random from the inter-quartile amplitude band.

Figure R3 Spatial pattern of LOCO variability in fitted lifespan trajectories. For each cortical vertex the color encodes the temporal average of the point-wise standard deviation of the four leave-one-cohort-out (LOCO) curves (units: gradient-units). Cooler colors (< 0.03) indicate vertices with lifespan fits hardly change when any single cohort is omitted, whereas warmer colors mark the upper tail of the distribution (cf. 95 th-percentile SD values reported in Response 2.1). Maps are shown separately for the SA, VS, and MR gradients

Comment 2.2: I'm not convinced that any of the suggested measures of either R-squared or percentage non-linearity address the question of model fits and would deem it essential to show in more depth that there is no age or site bias that could be driving the trajectory estimation. Rather than seeing plots of the DoF it would still be informative to see representative plots of the fits over the raw data in addition to the suggested permutation/LOO sensitivity analyses. While I appreciate that this may be computationally expensive perhaps some compromise can be made by only conducting these sensitivity analyses for a few random/extreme variables/vertices?

Response: We agree that demonstrating *how well* the fitted GAMM trajectories capture the raw data—and that they are not driven by hidden age- or site-specific artifacts—is essential. We address this in multiple ways:

1. Raw data overlays. Figures S8 and S9 display side-by-side plots of raw data and harmonized data following our 3-step GAMM fitting procedure, along with residual plots.
2. LOCO analysis overlays. **Figure R2** now presents four representative vertices (one low-amplitude, one high-amplitude, two random) with (i) the harmonized vertex values (after mean and variance adjustment) for every subject, color-coded by cohort; (ii) the full-data GAMM fit; (iii) the four LOCO curves; and (iv) the 95 % LOCO confidence band.
3. In the same LOCO folds the residual-age correlations were centred on zero (SA median $\rho = -0.0001$, IQR = -0.065 – -0.064 ; VS $\rho = 0.0035$; MR $\rho = -0.0027$).
4. A 10-fold CV (balanced by subject, not cohort) produced fold-to-fold SD $\leq 5.8\%$ of curve range and mean inter-fold trajectory correlations ≥ 0.95 for all three axes.

Ten-fold subject-balanced cross-validation showed that vertex-wise lifespan trajectories are highly stable across all three functional-connectivity axes. Mean fold-to-fold SDs were ≤ 0.04 gradient-units, corresponding to coefficients of variation below 13% when normalized by curve amplitude and below 6% when normalized by peak-to-trough range. Inter-fold trajectory correlations exceeded 0.95 for every axis (SA = 0.96 ± 0.15 ; VS = 0.97 ± 0.15 ; MR = 0.96 ± 0.17), confirming excellent reproducibility of the GAMM-based lifespan fits. Outlier CV values arise almost exclusively from vertices with very small amplitude or range, where even tiny absolute differences inflate relative metrics.

Metric	SA Axis	VS Axis	MR Axis
Avg. Fold-to-Fold Standard Deviation	0.0367	0.0232	0.0187
Coefficient of Variation (Magnitude Norm.)	7.3%	12.5%	10.3%
Coefficient of Variation (Range Norm.)	5.8%	4.7%	6.23%
Mean Inter-Fold Pearson Correlation	0.964 ± 0.148	0.972 ± 0.147	0.956 ± 0.166

Comment 2.3: Again, this is not an entirely satisfactory response. Inflection points could be quantified by derivatives and it simply seems off that during the entire 2-10 window there are no peaks or periods of more rapid change. How can the authors rule out that this smooth transition is not a side-effect of the model over smoothing (due to lack of data/harmonisation/diversity etc.)

Response: We thank the reviewer for this critical feedback and for pushing us to more rigorously interrogate the nature of the lifespan trajectories. Following the reviewer's suggestion, we have now performed a quantitative analysis to identify periods of most rapid change by computing the first derivative of the fitted GAMM trajectories for network centroids and dispersion measures (Figure 4, main text). We calculated the derivative for each network-level trajectory to pinpoint the age of its maximal rate of change. This analysis successfully identified the age at which each network metric showed its steepest change. We have updated Figure 4 in the main text to include markers indicating these points of maximum (or minimum) rate of change.

The reviewer rightly questions whether this smoothness could be a side-effect of the model. Our extensive sensitivity analyses, detailed in our responses to Comments 2.1 and 2.2, directly address this concern.

- **Model Stability:** If our GAMMs were oversmoothing the data due to data sparsity, site effects, or lack of diversity, the resulting trajectories would be unstable. Small changes to the input data (like removing a cohort or a random fold of subjects) would lead to large changes in the fitted curves. Our results show the opposite.
- **Robustness under Resampling:** The leave-one-cohort-out (LOCO) analysis (**Figures R2 and R3**) and the 10-fold cross-validation analysis demonstrated that our lifespan trajectories are remarkably stable and reproducible. For instance, the mean inter-fold correlation for vertex-wise trajectories was consistently ≥ 0.95 .
- **Conclusion:** The high stability of our model fits provides strong evidence that they are not arbitrary or over-regularized. This gives us confidence that the smooth, continuous nature of the observed change is a genuine feature of the data, reflecting a protracted and incremental process of brain maturation rather than an artifact of our modeling approach.

In summary, by implementing the reviewer's suggestion to use derivatives, we have further quantified the developmental trajectories. In combination with our comprehensive stability analyses, this new evidence reinforces our conclusion that the functional architecture of the brain reorganizes gradually and continuously throughout the 2-10 year window and beyond.

Comment 2.4: Rather than adding another dataset to the model. Could the authors show out of sample generalisability?

Response: To quantify out-of-sample (OOS) generalizability, we performed a *leave-one-cohort-out (LOCO)* evaluation on the four overlapping cohorts in our study (HCP-D and the three HBN sub-cohorts). For each fold we removed one cohort entirely, rebuilt the two-stage variance-aware GAMM on the remaining data, and predicted the withheld cohort. The resulting OOS trajectories deviated only marginally from the full-data fit: across 18 644 vertices the median point-wise LOCO standard deviation was 0.056 gradient-units for the SA axis, 0.032 for VS, and 0.029 for MR, corresponding to $\approx 6.8\%$, 4.8% , and 6.6% of each trajectory's peak-to-trough range, respectively. Vertex-wise OOS errors were tightly centred (median RMSE = 0.775, 0.496, 0.462 for SA, VS, MR), and—critically—the residuals in the held-out cohort were uncorrelated with chronological age (median $|\rho| < 0.01$; inter-quartile range $\approx \pm 0.07$). These results demonstrate that the harmonised gradient model learned from the remaining cohorts generalises to an unseen cohort without introducing age- or site-dependent bias, thereby satisfying the reviewer's request for explicit OOS validation.

Comment 2.5: My comment about harmonisation and dealing with the change in contrast in early development (and hence the untestedness of NODDI in early development) remains unanswered, I believe.

Response: We thank the reviewer for their persistence on this important point and apologize if our previous response was not sufficiently clear. We wish to first correct a critical misunderstanding: our analysis does not use NODDI.

As detailed in our Methods (section "Data Preprocessing -- Structural and Diffusion MRI"), our diffusion metrics were derived using Spherical Mean Spectrum Imaging (SMSI), a distinct biophysical model. We believe the reviewer's concerns stem from valid questions about handling structural data in early development, and we are happy to provide a more detailed explanation of how our pipeline is robust to these challenges.

The reviewer rightly points out that significant changes in tissue contrast (e.g., GM/WM balance) occur in early development. Our multi-stage processing pipeline was specifically designed to be robust to these effects.

1. Surface reconstruction is based on segmentation, not raw intensity. Our cortical surfaces are not derived directly from raw T1w/T2w image intensities, which are known to change with age. Instead, they are generated from tissue segmentation maps that classify voxels into gray matter, white matter, etc. Our deep learning tissue segmentation algorithm is designed to be robust across a wide age range. The final surfaces are geometric meshes, which are registered to age-specific atlases, decoupling our analysis from the "image contrast" the reviewer is concerned about.
2. Diffusion metrics are normalized quantities. The SMSI-derived metrics we use, such as intra-cellular and extra-cellular *volume fractions*, are inherently normalized. They represent the relative contribution of different tissue compartments to the diffusion signal within a voxel, not an absolute intensity value. This use of relative fractions makes the metrics less susceptible to global shifts in signal intensity that may occur across different ages, scanners, or cohorts.
3. To mitigate potential global shifts in features arising from cohort differences, all features are z-normalized at the individual level prior to computing the morphological similarity network (MSN). This ensures that MSN construction is based on relative, normalized values rather than absolute values, which may be influenced by variations in image contrast.

4. Any remaining potential deviation is mitigated by aligning the structural gradients with the GAMM-harmonized functional gradients.

Finally, while our work is based on SMSI, we want to address the reviewer's comment on the "untestedness" of models like NODDI in early-life cohorts. We respectfully note that there is a growing body of literature successfully applying these advanced biophysical models to pediatric and infant populations, demonstrating their feasibility and utility:

- Vaher et al., "General factors of white matter microstructure from DTI and NODDI in the developing brain," *NeuroImage* 2022
- Dimond et al., "Maturation and interhemispheric asymmetry in neurite density and orientation dispersion in early childhood," *NeuroImage* 2020
- Eaton-Rosen et al., "Longitudinal measurement of the developing grey matter in preterm subjects using multi-modal MRI," *NeuroImage* 2015
- Kunz et al., "Assessing white matter microstructure of the newborn with multi-shell diffusion MRI and biophysical compartment models," *NeuroImage* 2014

Comment 2.6: I thank the reviewer for the clarification on their GAMM procedure. As per my comment in 2.1 it would still be could to verify this by adding more cohort with overlapping ages or alternatively using a secondary harmonisation procedure to illustrate robustness.

Response: We thank the reviewer for their continued engagement on this critical issue of data harmonization. We understand the reviewer's desire for maximal reassurance that our results are not driven by site-specific artifacts.

In our prior responses to Comments 2.2 and 2.4, we demonstrated the robustness of our findings using a comprehensive leave-one-cohort-out (LOCO) cross-validation framework. The results showed that:

1. Our final lifespan trajectories were highly stable against the removal of any single cohort (**Figures R2 and R3**).
2. The model generalized well to unseen cohorts without introducing age- or site-dependent bias.

Comment 2.7: This is admittedly very much a catch 22 problem as using a uniform pipeline may introduce alternative challenges since especially the very young cohort likely require some customisation. A stronger alternative would be some permutation of pipeline or at the very a sensitivity analyses that shows that for example within one dataset different pipelines at different ages do not impact the results. Think for example the context of running regular free surfer in the BCP cohort kids that are older than 2, but infant FreeSurfer in the ones under two and comparing the trajectories to a unified pipeline to ensure that there is no impact.

Response: We thank the reviewer for this insightful comment and agree that addressing the potential pitfalls of applying a consistent processing pipeline across a lifespan that includes such dramatic neurodevelopmental change is paramount.

The reviewer proposes a sensitivity analysis where different pipelines (e.g., Infant FreeSurfer and standard FreeSurfer) are used for different age groups. While we appreciate the suggestion, we believe this approach is scientifically unsound for studying continuous lifespan trajectories. Switching pipelines at an arbitrary age cutoff (e.g., 2 years) would inevitably introduce a non-biological seam in the data. Any developmental changes observed around that age would be hopelessly confounded with pipeline-induced artifacts, making it impossible to interpret the results. The goal of our study is to characterize smooth, biological change, and this requires a consistent measurement tool across the entire age range.

For this reason, we chose the more robust and scientifically rigorous approach: developing and implementing a single, unified pipeline that is valid and reliable across the entire lifespan.

Critically, our pipeline is not a standard adult processing stream that has been retrofitted for younger ages. Instead, our approach is an extension of the state-of-the-art iBEAT V2.0 pipeline (Wang et al. 2023), which was specifically designed and extensively validated to handle the unique challenges of infant brain MRI, such as low tissue contrast and dynamic changes in brain size and morphology. By building upon a strong, infant-centric foundation, we ensure that our pipeline is most sensitive and accurate during the period of most rapid development, while maintaining consistency into adulthood.

The importance of this choice is starkly illustrated when comparing our method to the pipelines suggested by the reviewer. Wang et al. 2023 (*Nature Protocols*, <https://www.nature.com/articles/s41596-023-00806-x/figures/12>) directly compares the surface reconstructions from iBEAT with those from Infant FreeSurfer and the standard FreeSurfer at different ages. As is visually apparent, Infant FreeSurfer produces anatomically implausible and incomplete surfaces for the 6-month-old infant. Our surface reconstruction pipeline is a lifespan extension of iBEAT and produces high-quality, anatomically faithful surfaces from birth through late adulthood (**Figure R4**).

Rather than introducing a known source of error by mixing flawed or age-limited pipelines, we invested significant effort in applying a single, state-of-the-art, infant-validated pipeline across all participants. This ensures that the developmental trajectories we report reflect genuine biological processes, not methodological artifacts.

Figure R4. (Top) Tissue segmentation results given by our lifespan pipeline. (Bottom) Comparison of tissue segmentation and cortical surface reconstruction between Infant FreeSurfer and our lifespan pipeline.

Comment 2.8: Thank you for the clarification, no further comments on this.

Comment 2.9: Thank you for the clarification, no further comments on this.

Comment 2.10: This is a very interesting and to my knowledge novel approach. Rather than removing variance from the data entirely the authors may also want to consider interpreting the variability in variance itself as a meaningful biological signal (perhaps in the context of for example the vulnerability hypothesis).

Response: We thank the reviewer for this excellent and insightful suggestion. We agree that inter-subject variability is a potentially crucial source of biological information, and not merely statistical noise. Following this suggestion, we have now quantified the vertex-wise inter-subject variance in gradient values (after harmonizing for cohort effects) and mapped this spatial distribution of variability and displayed the result in **Figure S12**.

Comment 2.11: If that is the case, what do the longitudinal data-points add to the model?

Response: Longitudinal data points in the BCP stabilize our lifespan models of gradient values by absorbing child-specific baselines. To demonstrate this, we refitted models to the BCP data twice, first with a GAMM with the subject random intercept and once with a GAM lacking this term. Adding the random intercept captured a mean intraclass correlation coefficient of 0.22, indicating that approximately 22% of the variance is stable within individuals. This reduced residual variance for 22% on average and improved the fit decisively with a mean improvement in AIC of 15.8. **Figure R5A** illustrates the trajectories for two representative vertices, while **Figure R5B** illustrates that deviance-based R^2 distributions are unchanged and the conditional R^2 (proportion of variance explained by both fixed and random effects) shifts upward, corroborating the variance-partitioning argument.

The longitudinal scans contribute additional information that helps absorb subject-specific offsets and reduce noise. Retaining the subject-level random effect thus yields a more precise and statistically robust characterization of gradient trajectories.

Figure R5. A, Two exemplar vertices in BCP plotting raw (open circles) and subject-adjusted (filled circles) gradient values against age, with the fitted spline from the GAMM with subject random effect (solid red) and from the GAM without subject random effect (solid blue). **B**, Density of four R^2 metrics across 500 vertices (deviance-based R^2 `gam_rsqs` and `gamm_fixed_rsqs`; variance-based R^2 : `gamm_R2_marginal` and `gamm_R2_conditional`) illustrating that `gamm_R2_conditional` exceeds all others, while `gam_rsqs` and `gamm_fixed_rsqs` nearly coincide and lie below the variance-based estimates.

Comment 2.12: Perhaps a general comment across these additional analyses would be to add them with some minimal interpretation to the supplement. Specifically I'd be curious to see if the TBV adjustment would have some age-related component to it. Its encouraging to see that the overall trajectories are unaltered by including this covariate, but I could imagine that it may play a role in period where the volumetric tissue changes are somewhat accelerated (i.e., very early development and (very) old age) that wouldn't perhaps be apparent when compared across the entire lifespan.

Response: We thank the reviewer for this insightful suggestion. To formally test whether TBV has a more pronounced age-related effect during periods of accelerated volumetric change, we carried out two new vertex-wise GAM analyses that focus on the life-stages in which intracranial volume changes most rapidly.

First, we refitted the model during infancy (≤ 3 years; 499 scans) using a square-root age axis. Across the vertices, the overall effect of TBV was small, with a median coefficient (β_{TBV}) of -0.008 and an inter-quartile range from -0.056 to $+0.042$.

To illustrate the maximal potential influence of this variable, we examined the vertex with the largest absolute effect size (v_{1280}), which is plotted in **Figure R6d**. For this single vertex, the effect is substantial: the predicted difference between children at ± 2 standard deviations of mean TBV was 1.12 gradient-units, which corresponds to 32.4% of that vertex's total dynamic range. Crucially, however, such strong, statistically significant effects are exceptionally rare, occurring in only $< 0.5\%$ of the vertices.

The same procedure was repeated for late old age (≥ 80 years; 102 scans) on a linear age scale. Here, the median β_{TBV} was near zero ($+0.0018$), and no vertex reached FDR significance.

These results indicate that while TBV's influence is negligible across the vast majority of the cortex, it can exert a material influence on the functional gradient values of a small subset of vertices during the rapid volumetric changes of infancy. However, given that these strong effects are highly localized and are absent in late old age, we conclude that TBV is not a *general* confounder of the principal lifespan trajectories presented in our main findings. Its contribution appears to be a localized, early-life phenomenon rather than a systematic, lifespan-wide bias.

Figure R6. Vertex-wise sensitivity of FC-gradient values to total brain volume (TBV) in the two life-stages of greatest volumetric change. **a**, Distribution of TBV coefficients (β_{TBV}) estimated in infancy (≤ 3 y; $N = 499$). While the distribution is centered near zero, 0.5% vertices (in the tails) survive FDR correction ($q < 0.05$). **b**, Corresponding distribution for late old age (≥ 80 y; $N = 102$), where no vertex is FDR-significant. **c**, Side-by-side violin and box plots summarizing the infancy (left) and old-age (right) coefficient distributions. **d**, Population-level trajectories for the vertex (v_{1279}) that yields the largest model-predicted TBV effect in infancy. Fixing TBV at -2 SD (black) versus $+2$ SD (red) offsets the curve by 1.12 gradient-units, which equals 32.4% of that vertex's observed range, illustrating a material brain volume effect in this extreme case.

Comment 2.13: Very nice to see :)

Comment 2.14: Thank you for the clarification.

Comment 2.15: Where are these comparisons in the manuscript/supplement?

Response: In the revised manuscript, we include Figure S13, which details the between-network result using the centroid approach. Whereas the previous version of our manuscript used the between-network dispersion normalized by global dispersion, we now present the unnormalized version for enhanced interpretability.

Comment 2.16: Thank you for the clarification.

Comment 2.17: Agreed with the authors that this adds to the results.

Comment 2.18: Agreed that this is better and closer to existing literature.

Comment 2.19: Thanks :)

Comment 2.20: Its still in the title though :)

Response: We understand the criticism of our title, and although we think it is an apt and eye-catching characterization of our work, we are open to the judgment of the editorial team as to whether or not it is ultimately appropriate. We remain open to suggestions.

Comment 2.21: Thank you this is helpful I think.

Referee #3

Comment 3.1: Taylor et al. have revised their examination of functional connectivity (FC) gradients across the human lifespan and have satisfactorily addressed my prior relatively minor concerns. The revised manuscript includes the incorporation of an additional dataset (Healthy Brain Network (HBN)) along with examination of potential covariates or interest/confounders, such as biological sex. The overall results seem to hold.

The initial review requested that the authors address the Neuron perspective published by Petersen et al. in which they challenged the fundamental validity of pursuing gradient-based analyses based on the well-established principle of arealization – asserting that cortical areas are valid entities and implying that gradients are likely fictions arising from smoothing across large samples. The authors mention this concern in the last paragraph of the discussion: “There is strong evidence that gradients of FC, microstructure, and gene expression represent crucial organizing axes of the cerebral cortex [2]. Still, recent pushback about the power of gradients as a comprehensive framework for understanding topographic organization of the cortex bears mentioning. The principle of arealization, whereby the cortex is divided into discrete areas which are differentiated by cytoarchitecture and connectivity, is in some ways at odds with the claim that gradients represent fundamental organizing axes on an individual level [66]. Gradients of functional connectivity typically do not smoothly vary with respect to space on an individual level, reflecting instead sharp boundaries between functional systems. At a population level, however, gradients are smooth and excel at describing functional topography. The interpretation of FC gradients in individuals remains an important area of future research, however the present study focuses on large-scale population-level effects.”

The phrase “is in some ways at odds...” fails to convey the direct frontal attack on gradients asserted by Petersen and colleagues. To me this harkens back to vigorous debates regarding the physical nature of light – wave vs. particle. We now all accept that there is no resolution of this question – it all depends on the experimental context/purpose. The real test of validity is explanatory utility. The initial Margulies et al. paper on gradients provided a novel explanation for the physical distribution of default mode network centroids based on analyses of the principal gradient. Such an insight had not emerged from arealization-based perspectives. Here, the authors have provided some data potentially bearing on such explanatory utility, but it’s presented as a supplementary table with little explication. For example, what do Eval1, Eval2, and Eval3 refer to? At any rate, 11 of the 60 relationships with aggregate measures from the Mullen in the Baby HCP presented were below an alpha of 0.05, which does support the inference that some may not be type 1 errors, thus indirectly supporting the explanatory utility of these gradients at least in the youngest participants.

However, given the challenge posed by Petersen et al., and the availability of standardized cognitive/motor measures such as the NIH ToolBox from the HCP and HBN datasets, I urge the authors to consider reexamining the relevant question of explanatory utility of their 3 gradients. They are in the best position to relatively rapidly report whether these gradients can truly be asserted to have quasi clinical utility. [Quasi clinical, because the field lacks sufficiently reliable clinical indices, and most of these datasets were not designed to examine such questions.] If they are able to provide further supportive data from examining gradient-behavior relationships, I would suggest that the Petersen et al. challenge be addressed in the paper’s introduction rather than at the end. This is a major scientific challenge – and it merits a frontal response. At the end of the day, disputes will continue regarding the fundamental validity of functional connectivity because we still lack an understanding of the neurobiological bases for this phenomenon. The reasons why all nervous systems expend such massive quantities of energy in maintaining low-frequency correlations across the brain remain profoundly mysterious. Simple Hebbian arguments are belied by elegant experiments such as documenting rapid changes in functional connectivity after wearing a cast on one’s arm and removing it. Presumably synaptic strengthening/weakening does not occur so rapidly. This is the major challenge of 21st

century human neuroscience in my opinion, and this manuscript has the potential to advance the relevant science.

Response: Thank you for your thoughtful and highly constructive feedback on our manuscript. We are grateful for your strong support and particularly for your insightful comments regarding the Petersen et al. challenge to the gradient framework.

Following your suggestion, we have re-examined the question of the gradients' explanatory utility using standardized cognitive data. We performed a new analysis linking metrics from our three primary **gradients** to performance on the NIH Toolbox Cognition Battery in the Human Connectome Project-Young Adult (HCP-YA) cohort (N = 1,066).

To further clarify our methods, we have also added text explaining that Eval1, Eval2, and Eval3 reflect the eigenvalues associated with each individual's gradients before alignment—a template-agnostic measure of scale. Additionally, we have updated our analysis of the BCP Mullen scores with a consistent FDR correction, which accounts for the revised results in the manuscript.

The results, now included in the main text as Figure 7 and in Tables S1 and S2, provide strong support for the utility of the gradient framework. We found a robust and widespread association between our gradient metrics and cognitive performance. Most notably, the similarity of an individual's SA gradient topography to the canonical template (SA cossim) was significantly associated with performance across every cognitive domain tested, with standardized coefficients (β) up to 0.19 (all $p_{\text{FDR}} < .05$). This demonstrates that the principal organizing axis of the cortex has tangible relevance for explaining individual differences in cognitive abilities—an insight not easily derived from a purely arealization-based perspective.

Also following your suggestion, we have recentered our response to the Petersen et al. critique by addressing it early in the introduction (paragraph #2), borrowing your analogy of the particle-wave duality of light and asserting that the “resolution lies not in choosing one correct description, but in understanding the context in which each provides value. A key test of any framework's validity is therefore its explanatory utility: its ability to provide novel insights into brain function and behavior.”

Comment 3.2: Trivially, on line 605, the authors describe a correlation as “negative” whereas it is manifestly positive both in the figure and in the following text “($p = 0.25$, $p < 10^{-58}$).”

In the first paragraph of the discussion, the authors write: “Integrating the vast and diverse body of fMRI literature into an easily interpreted and succinct framework represents a central obstacle to neuroscience; the present study offers a substantial step forward towards achieving this goal.” I object to the use of the word “obstacle.” They mean “challenge” or similar; otherwise, they need to change the sentence structure.

In line 504, they unnecessarily capitalize “early life” as “Early life.” In that same paragraph, they switch from reporting maxima of age-related changes from one decimal point to two. Given the breadth of most of the corresponding confidence intervals (e.g., 95% CI 9.5–17.8 years), such hyperprecision is unwarranted. I recommend staying with one decimal point.

In line 800, the period after “index” is missing. In line 860, “quality” is misspelled.

Response: Thank you for the careful reading of the paper. These have been corrected.

Comment 3.3: I reiterate that this prodigious effort likely represents a substantial advance in human neuroscience and support its publication.

Response: We are grateful for your strong support and your thoughtful reflections and suggestions that helped to substantially strengthen our manuscript.

Note: Major changes in the manuscript are marked in brown.

Referee #1

Comment: In brief, R1 was impressed with the writing, data, and appreciated the potential impact of the conclusions. However, given recently published reference A (listed below), R1 felt that the main contribution are the gradient-related claims and was concerned both that these were not adequately substantiated, as well as whether these results represent a sufficiently striking advance to warrant further consideration in Nature.

Ref A: Human lifespan changes in the brain's functional connectome

<https://www.nature.com/articles/s41593-025-01907-4>

Functional connectivity of the human brain changes through life. Here, we assemble task-free functional and structural magnetic resonance imaging data from 33,250 individuals at 32 weeks of postmenstrual age to 80 years from 132 global sites. We report critical inflection points in the nonlinear growth curves of the global mean and variance of the connectome, peaking in the late fourth and late third decades of life, respectively. After constructing a fine-grained, lifespan-wide suite of system-level brain atlases, we show distinct maturation timelines for functional segregation within different systems. Lifespan growth of regional connectivity is organized along a spatiotemporal cortical axis, transitioning from primary sensorimotor regions to higher-order association regions. These findings elucidate the lifespan evolution of the functional connectome and can serve as a normative reference for quantifying individual variation in development, aging and neuropsychiatric disorders.

Response: We appreciate the referee's positive remarks about the quality of our data, writing, and potential impact. Because the referee's full comments were not shared with us, we respond to the two concerns highlighted in the editor's summary: (i) whether our gradient findings are fully substantiated, and (ii) whether these findings represent a sufficiently striking advance beyond Sun et al. 2025.

1. Substantiation of gradient results.

Our revised analysis further substantiates the validity of our results:

- **Balanced-bootstrap tests** confirm that the age-weighted PCA template is unbiased.
- **Leave-one-cohort-out and 10-fold cross-validation** demonstrate that the developmental trajectories are stable and not driven by site effects.
- **Vertex-wise models** show that total brain volume and sex do not confound the findings.
- **Illustrative comparisons** show that our unified pipeline produces improved results relative to pipelines designed for specific age ranges.
- **Derivative analyses** verify that rapid developmental changes happen primarily before age 10.
- **Gradient-cognition associations** further substantiate the functional relevance of gradients by linking them to cognitive performance.

2. Advance over Sun et al.

Sun et al. catalogued how mean connectivity and network segregation vary with age. Our work asks a different, deeper question: *How is the entire connectome hierarchically organized, and how do those organizing principles mature and decline?* We provide the first lifespan atlas of the three principal functional gradients, link these gradients to parallel structural axes, and—critically—show that the fidelity of the sensorimotor-association gradient predicts every domain of NIH-Toolbox cognition in young adults. Sun et al. did not incorporate gradient-based analyses and did not explore links to behavioral measures. Thus our study adds a hierarchical, multimodal, behavior-relevant perspective that is entirely absent from prior work.

We believe these new analyses resolve the referee's concerns and demonstrate that the gradient framework delivers a robust and genuinely novel advance.

Referee #2

I thank the authors for their constructive response to my comments. However I still have some major reservations some of which may be stemming from some lack of detail/clarification in my earlier comments. Hence let me provide specific responses using the same numbering system for clarity.

Comment 2.1: While the weighted PCA may reduce some bias driven by unequal distribution of sample size across the lifespan, I would still encourage the authors to actually show that it does (for example by running an ablation analyses with equally size samples across the bins).

In addition, one of my main original concerns was that perhaps not all age-bins are created equal. For example: if we hypothesise that say between years 1-2 there is exponential change that could be 5* larger than what happens in any of the subsequent 1 year bins, then taking the weighted approach across 10 1-year bins for the first 10 years of life would still bias the template towards the latter 9 years where little changes and negate the period of largest change. I don't see how the added analyses circumvents that? I could perhaps agree that this combined with the alignment procedure retains variation across individuals but that doesn't mean that the variation is an ageing signal so much as an alignment to the template.

Further to the comment on harmonisation: to my knowledge using a harmonised pipeline (in so far as that is even realistically possible let alone desirable across the lifespan) does not in and of itself harmonise data from different sources. I appreciate that harmonising functional data may not be a trivial task but it does seem quite essential to the results presented here. Perhaps including more data-sets with overlapping age-range would allow the authors to conduct a more quantitative sensitivity analyses to determine whether any residual site related variation is still present in the data? Or alternative they could deploy a ComBAT like harmonisation approach or choose a modelling framework that more readily allows the modelling of higher moments in the data distribution (i.e., HBR/GAMLSS).

Response: We regret that our description of the age-bin construction in the previous response letter was unclear. The ten bins are **not** ten equal-width, 10-year intervals across the lifespan. Instead, we first apply an $\alpha = 0.50$ power-law transform to chronological age ($\text{age}' = \text{age}^{0.5}$) and then set equally spaced edges in the *transformed* domain. On the original age scale, this compresses late-life intervals and expands early-life intervals, so that periods of rapid neurodevelopment (e.g., 0-2 y) occupy proportionally *larger* bins and receive proportionally *higher* leverage.

To address the question of whether our weighted PCA approach effectively neutralizes bias towards the bins with larger sample sizes, we conducted an ablation analysis using exactly the ten $\alpha = 0.50$ age-bins that underlie the canonical weighted-PCA (WPCA) template used in our main analysis. Across **B = 500** balanced-bootstrap samples (50 subjects drawn per age-bin, 10 bins with $\alpha = 0.5$), we measured the 3-D Procrustes angle θ between each bootstrap PCA template and the canonical weighted PCA (WPCA) template. The mean θ was **6.18°** (SD 0.52°), with a median of 6.15° (IQR 5.82–6.50°) and a 95% interval of 5.30–7.27° (2.5–97.5 percentiles). **Figure R1 (Figure S15** in the revised manuscript) illustrates the distributions of correlations and subspace angles between our balanced bootstrapped templates and the WPCA template. These tight bounds confirm that WPCA reliably recovers the template one would obtain from an age-balanced cohort, demonstrating that inverse-frequency weighting effectively removes sampling bias in our lifespan data.

We also examined per-axis correlation coefficients (r) between the WPCA axes and those of each balanced template:

- **PC1:** mean $r = 0.995 \pm 0.001$ (median 0.995, IQR 0.995–0.996; 95% CI 0.994–0.996)
- **PC2:** mean $r = 0.955 \pm 0.012$ (median 0.956, IQR 0.948–0.963; 95% CI 0.929–0.977)
- **PC3:** mean $r = 0.944 \pm 0.013$ (median 0.945, IQR 0.936–0.954; 95% CI 0.917–0.968)

High correlations on PC1–PC3 confirm that the major axes of variance are preserved under WPCA weighting. Together, these results directly address the reviewer’s request by quantifying how closely the weighted template matches the balanced-sample-size approach, and they demonstrate that our weighting scheme substantially mitigates bias arising from unequal age-bin sizes.

Figure R1. Template stability analysis. For each of 500 balanced datasets (50 participants drawn from every $\alpha = 0.50$ age bin; replacement only when a bin contained < 50 individuals) an unweighted PCA was recomputed, its first three axes were greedily aligned to the canonical weighted-PCA (WPCA) axes, and similarity was quantified.

We appreciate the reviewer’s insightful comment regarding the potential influence of the template on observed individual variation. It is indeed crucial to ensure that the detected ageing signals are genuine biological changes rather than artifacts of the alignment process. Because the Procrustes transform is an **isometry** (pure rotation + reflection, no scaling), it *preserves* all pairwise Euclidean distances among vertices within each subject’s gradient space. Consequently, the alignment cannot increase—or decrease—the covariance between any gradient coordinate and chronological age; it merely puts homologous gradients in register (Vasa et al., 2018; Paquola et al., 2019). The ageing trajectories detected by the vertex-wise GAMMs therefore arise from systematic changes in the gradient topographies.

When each of the four overlapping cohorts (HCPD, three subcohorts of the HBN) was withheld, and the full harmonization/GAMM pipeline was re-run, the resulting lifespan trajectories deviated only minimally from the full-data fit. Across the SA axis (18,644 vertices) the median point-wise SD of the four leave-one-cohort-out (LOCO) curves was 0.056 gradient-units (IQR = 0.040–0.078; 95th% = 0.125), whereas the median peak-to-trough range of the trajectories was 0.821 units. Thus, the typical LOCO variability amounts to $\approx 6.8\%$ of signal amplitude across all vertices. The VS and MR axes showed comparable or lower variability (VS: median SD = 0.032, range = 0.665 \rightarrow 4.8%; MR: median SD = 0.029, range = 0.435 \rightarrow 6.6%). Consistent with this, the median change in amplitude relative to the full-data curve was only 6.1% (SA), 6.0% (VS), and 4.6% (MR), and the age at which each vertex reached its maximum shifted by ≤ 0.75 years for 75% of vertices across all axes. Model residuals in the held-out cohorts were essentially uncorrelated with chronological age (median $\rho \approx 0$, IQR $\approx \pm 0.07$) and root-mean-square error distributions were tightly centered (SA median = 0.775; VS = 0.496; MR = 0.462). Together with the 10-fold cross-validation analysis (fold-to-fold SD $\leq 5.8\%$ of curve range and mean inter-fold trajectory correlation ≥ 0.95), these findings demonstrate that neither site membership nor uneven age sampling materially biases the estimated functional gradient trajectories.

Figure R2 Leave-one-cohort-out (LOCO) variability at four representative cortical vertices. Each panel shows the lifespan trajectory of the SA gradient at a single vertex after aligning gradients into the canonical space. Vertices were chosen to span the empirical range of signal amplitudes: **a**, the lowest-range vertex with a discernible signal, **b**, the highest-range vertex, and **c–d**, two vertices drawn at random from the inter-quartile amplitude band.

Figure R3 Spatial pattern of LOCO variability in fitted lifespan trajectories. For each cortical vertex the color encodes the temporal average of the point-wise standard deviation of the four leave-one-cohort-out (LOCO) curves (units: gradient-units). Cooler colors (< 0.03) indicate vertices with lifespan fits hardly change when any single cohort is omitted, whereas warmer colors mark the upper tail of the distribution (cf. 95 th-percentile SD values reported in Response 2.1). Maps are shown separately for the SA, VS, and MR gradients

Comment 2.2: I'm not convinced that any of the suggested measures of either R-squared or percentage non-linearity address the question of model fits and would deem it essential to show in more depth that there is no age or site bias that could be driving the trajectory estimation. Rather than seeing plots of the DoF it would still be informative to see representative plots of the fits over the raw data in addition to the suggested permutation/LOO sensitivity analyses. While I appreciate that this may be computationally expensive perhaps some compromise can be made by only conducting these sensitivity analyses for a few random/extreme variables/vertices?

Response: We agree that demonstrating *how well* the fitted GAMM trajectories capture the raw data—and that they are not driven by hidden age- or site-specific artifacts—is essential. We address this in multiple ways:

1. Raw data overlays. Figures S8 and S9 display side-by-side plots of raw data and harmonized data following our 3-step GAMM fitting procedure, along with residual plots.
2. LOCO analysis overlays. **Figure R2** now presents four representative vertices (one low-amplitude, one high-amplitude, two random) with (i) the harmonized vertex values (after mean and variance adjustment) for every subject, color-coded by cohort; (ii) the full-data GAMM fit; (iii) the four LOCO curves; and (iv) the 95 % LOCO confidence band.
3. In the same LOCO folds the residual-age correlations were centred on zero (SA median $\rho = -0.0001$, IQR = -0.065 – -0.064 ; VS $\rho = 0.0035$; MR $\rho = -0.0027$).
4. A 10-fold CV (balanced by subject, not cohort) produced fold-to-fold SD $\leq 5.8\%$ of curve range and mean inter-fold trajectory correlations ≥ 0.95 for all three axes.

Ten-fold subject-balanced cross-validation showed that vertex-wise lifespan trajectories are highly stable across all three functional-connectivity axes. Mean fold-to-fold SDs were ≤ 0.04 gradient-units, corresponding to coefficients of variation below 13% when normalized by curve amplitude and below 6% when normalized by peak-to-trough range. Inter-fold trajectory correlations exceeded 0.95 for every axis (SA = 0.96 ± 0.15 ; VS = 0.97 ± 0.15 ; MR = 0.96 ± 0.17), confirming excellent reproducibility of the GAMM-based lifespan fits. Outlier CV values arise almost exclusively from vertices with very small amplitude or range, where even tiny absolute differences inflate relative metrics.

Metric	SA Axis	VS Axis	MR Axis
Avg. Fold-to-Fold Standard Deviation	0.0367	0.0232	0.0187
Coefficient of Variation (Magnitude Norm.)	7.3%	12.5%	10.3%
Coefficient of Variation (Range Norm.)	5.8%	4.7%	6.23%
Mean Inter-Fold Pearson Correlation	0.964 ± 0.148	0.972 ± 0.147	0.956 ± 0.166

Comment 2.3: Again, this is not an entirely satisfactory response. Inflection points could be quantified by derivatives and it simply seems off that during the entire 2-10 window there are no peaks or periods of more rapid change. How can the authors rule out that this smooth transition is not a side-effect of the model over smoothing (due to lack of data/harmonisation/diversity etc.)

Response: We thank the reviewer for this critical feedback and for pushing us to more rigorously interrogate the nature of the lifespan trajectories. Following the reviewer's suggestion, we have now performed a quantitative analysis to identify periods of most rapid change by computing the first derivative of the fitted GAMM trajectories for network centroids and dispersion measures (Figure 4, main text). We calculated the derivative for each network-level trajectory to pinpoint the age of its maximal rate of change. This analysis successfully identified the age at which each network metric showed its steepest change. We have updated Figure 4 in the main text to include markers indicating these points of maximum (or minimum) rate of change.

The reviewer rightly questions whether this smoothness could be a side-effect of the model. Our extensive sensitivity analyses, detailed in our responses to Comments 2.1 and 2.2, directly address this concern.

- **Model Stability:** If our GAMMs were oversmoothing the data due to data sparsity, site effects, or lack of diversity, the resulting trajectories would be unstable. Small changes to the input data (like removing a cohort or a random fold of subjects) would lead to large changes in the fitted curves. Our results show the opposite.
- **Robustness under Resampling:** The leave-one-cohort-out (LOCO) analysis (**Figures R2 and R3**) and the 10-fold cross-validation analysis demonstrated that our lifespan trajectories are remarkably stable and reproducible. For instance, the mean inter-fold correlation for vertex-wise trajectories was consistently ≥ 0.95 .
- **Conclusion:** The high stability of our model fits provides strong evidence that they are not arbitrary or over-regularized. This gives us confidence that the smooth, continuous nature of the observed change is a genuine feature of the data, reflecting a protracted and incremental process of brain maturation rather than an artifact of our modeling approach.

In summary, by implementing the reviewer's suggestion to use derivatives, we have further quantified the developmental trajectories. In combination with our comprehensive stability analyses, this new evidence reinforces

our conclusion that the functional architecture of the brain reorganizes gradually and continuously throughout the 2-10 year window and beyond.

Comment 2.4: Rather than adding another dataset to the model. Could the authors show out of sample generalisability?

Response: To quantify out-of-sample (OOS) generalizability, we performed a *leave-one-cohort-out* (LOCO) evaluation on the four overlapping cohorts in our study (HCP-D and the three HBN sub-cohorts). For each fold we removed one cohort entirely, rebuilt the two-stage variance-aware GAMM on the remaining data, and predicted the withheld cohort. The resulting OOS trajectories deviated only marginally from the full-data fit: across 18 644 vertices the median point-wise LOCO standard deviation was 0.056 gradient-units for the SA axis, 0.032 for VS, and 0.029 for MR, corresponding to $\approx 6.8\%$, 4.8% , and 6.6% of each trajectory's peak-to-trough range, respectively. Vertex-wise OOS errors were tightly centred (median RMSE = 0.775, 0.496, 0.462 for SA, VS, MR), and—critically—the residuals in the held-out cohort were uncorrelated with chronological age (median $|\rho| < 0.01$; inter-quartile range $\approx \pm 0.07$). These results demonstrate that the harmonised gradient model learned from the remaining cohorts generalises to an unseen cohort without introducing age- or site-dependent bias, thereby satisfying the reviewer's request for explicit OOS validation.

Comment 2.5: My comment about harmonisation and dealing with the change in contrast in early development (and hence the untestedness of NODDI in early development) remains unanswered, I believe.

Response: We thank the reviewer for their persistence on this important point and apologize if our previous response was not sufficiently clear. We wish to first correct a critical misunderstanding: our analysis does not use NODDI.

As detailed in our Methods (section "Data Preprocessing -- Structural and Diffusion MRI"), our diffusion metrics were derived using Spherical Mean Spectrum Imaging (SMSI), a distinct biophysical model. We believe the reviewer's concerns stem from valid questions about handling structural data in early development, and we are happy to provide a more detailed explanation of how our pipeline is robust to these challenges.

The reviewer rightly points out that significant changes in tissue contrast (e.g., GM/WM balance) occur in early development. Our multi-stage processing pipeline was specifically designed to be robust to these effects.

1. Surface reconstruction is based on segmentation, not raw intensity. Our cortical surfaces are not derived directly from raw T1w/T2w image intensities, which are known to change with age. Instead, they are generated from tissue segmentation maps that classify voxels into gray matter, white matter, etc. Our deep learning tissue segmentation algorithm is designed to be robust across a wide age range. The final surfaces are geometric meshes, which are registered to age-specific atlases, decoupling our analysis from the "image contrast" the reviewer is concerned about.
2. Diffusion metrics are normalized quantities. The SMSI-derived metrics we use, such as intra-cellular and extra-cellular *volume fractions*, are inherently normalized. They represent the relative contribution of different tissue compartments to the diffusion signal within a voxel, not an absolute intensity value. This use of relative fractions makes the metrics less susceptible to global shifts in signal intensity that may occur across different ages, scanners, or cohorts.
3. To mitigate potential global shifts in features arising from cohort differences, all features are z-normalized at the individual level prior to computing the morphological similarity network (MSN). This ensures that MSN construction is based on relative, normalized values rather than absolute values, which may be influenced by variations in image contrast.
4. Any remaining potential deviation is mitigated by aligning the structural gradients with the GAMM-harmonized functional gradients.

Finally, while our work is based on SMSI, we want to address the reviewer's comment on the "untestedness" of models like NODDI in early-life cohorts. We respectfully note that there is a growing body of literature successfully applying these advanced biophysical models to pediatric and infant populations, demonstrating their feasibility and utility:

- Vaher et al., "General factors of white matter microstructure from DTI and NODDI in the developing brain," *NeuroImage* 2022
- Dimond et al., "Maturation and interhemispheric asymmetry in neurite density and orientation dispersion in early childhood," *NeuroImage* 2020
- Eaton-Rosen et al., "Longitudinal measurement of the developing grey matter in preterm subjects using multi-modal MRI," *NeuroImage* 2015
- Kunz et al., "Assessing white matter microstructure of the newborn with multi-shell diffusion MRI and biophysical compartment models," *NeuroImage* 2014

Comment 2.6: I thank the reviewer for the clarification on their GAMM procedure. As per my comment in 2.1 it would still be could to verify this by adding more cohort with overlapping ages or alternatively using a secondary harmonisation procedure to illustrate robustness.

Response: We thank the reviewer for their continued engagement on this critical issue of data harmonization. We understand the reviewer's desire for maximal reassurance that our results are not driven by site-specific artifacts.

In our prior responses to Comments 2.2 and 2.4, we demonstrated the robustness of our findings using a comprehensive leave-one-cohort-out (LOCO) cross-validation framework. The results showed that:

1. Our final lifespan trajectories were highly stable against the removal of any single cohort (**Figures R2 and R3**).
2. The model generalized well to unseen cohorts without introducing age- or site-dependent bias.

Comment 2.7: This is admittedly very much a catch 22 problem as using a uniform pipeline may introduce alternative challenges since especially the very young cohort likely require some customisation. A stronger alternative would be some permutation of pipeline or at the very a sensitivity analyses that shows that for example within one dataset different pipelines at different ages do not impact the results. Think for example the context of running regular free surfer in the BCP cohort kids that are older than 2, but infant FreeSurfer in the ones under two and comparing the trajectories to a unified pipeline to ensure that there is no impact.

Response: We thank the reviewer for this insightful comment and agree that addressing the potential pitfalls of applying a consistent processing pipeline across a lifespan that includes such dramatic neurodevelopmental change is paramount.

The reviewer proposes a sensitivity analysis where different pipelines (e.g., Infant FreeSurfer and standard FreeSurfer) are used for different age groups. While we appreciate the suggestion, we believe this approach is scientifically unsound for studying continuous lifespan trajectories. Switching pipelines at an arbitrary age cutoff (e.g., 2 years) would inevitably introduce a non-biological seam in the data. Any developmental changes observed around that age would be hopelessly confounded with pipeline-induced artifacts, making it impossible to interpret the results. The goal of our study is to characterize smooth, biological change, and this requires a consistent measurement tool across the entire age range.

For this reason, we chose the more robust and scientifically rigorous approach: developing and implementing a single, unified pipeline that is valid and reliable across the entire lifespan.

Critically, our pipeline is not a standard adult processing stream that has been retrofitted for younger ages. Instead, our approach is an extension of the state-of-the-art iBEAT V2.0 pipeline (Wang et al. 2023), which was specifically designed and extensively validated to handle the unique challenges of infant brain MRI, such as low tissue contrast and dynamic changes in brain size and morphology. By building upon a strong, infant-centric foundation, we ensure that our pipeline is most sensitive and accurate during the period of most rapid development, while maintaining consistency into adulthood.

The importance of this choice is starkly illustrated when comparing our method to the pipelines suggested by the reviewer. Wang et al. 2023 (*Nature Protocols*, <https://www.nature.com/articles/s41596-023-00806-x/figures/12>) directly compares the surface reconstructions from iBEAT with those from Infant FreeSurfer and the standard FreeSurfer at different ages. As is visually apparent, Infant FreeSurfer produces anatomically implausible and incomplete surfaces for the 6-month-old infant. Our surface reconstruction pipeline is a lifespan extension of iBEAT and produces high-quality, anatomically faithful surfaces from birth through late adulthood (**Figure R4**).

Rather than introducing a known source of error by mixing flawed or age-limited pipelines, we invested significant effort in applying a single, state-of-the-art, infant-validated pipeline across all participants. This ensures that the developmental trajectories we report reflect genuine biological processes, not methodological artifacts.

Figure R4. (Top) Tissue segmentation results given by our lifespan pipeline. (Bottom) Comparison of tissue segmentation and cortical surface reconstruction between Infant FreeSurfer and our lifespan pipeline.

Comment 2.8: Thank you for the clarification, no further comments on this.

Comment 2.9: Thank you for the clarification, no further comments on this.

Comment 2.10: This is a very interesting and to my knowledge novel approach. Rather than removing variance from the data entirely the authors may also want to consider interpreting the variability in variance itself as a meaningful biological signal (perhaps in the context of for example the vulnerability hypothesis).

Response: We thank the reviewer for this excellent and insightful suggestion. We agree that inter-subject variability is a potentially crucial source of biological information, and not merely statistical noise. Following this suggestion, we have now quantified the vertex-wise inter-subject variance in gradient values (after harmonizing for cohort effects) and mapped this spatial distribution of variability and displayed the result in **Figure S12**.

Comment 2.11: If that is the case, what do the longitudinal data-points add to the model?

Response: Longitudinal data points in the BCP stabilize our lifespan models of gradient values by absorbing child-specific baselines. To demonstrate this, we refitted models to the BCP data twice, first with a GAMM with the subject random intercept and once with a GAM lacking this term. Adding the random intercept captured a mean intraclass correlation coefficient of 0.22, indicating that approximately 22% of the variance is stable within individuals. This reduced residual variance for 22% on average and improved the fit decisively with a mean improvement in AIC of 15.8. **Figure R5A** illustrates the trajectories for two representative vertices, while **Figure R5B** illustrates that deviance-based R^2 distributions are unchanged and the conditional R^2 (proportion of variance explained by both fixed and random effects) shifts upward, corroborating the variance-partitioning argument.

The longitudinal scans contribute additional information that helps absorb subject-specific offsets and reduce noise. Retaining the subject-level random effect thus yields a more precise and statistically robust characterization of gradient trajectories.

Figure R5. A, Two exemplar vertices in BCP plotting raw (open circles) and subject-adjusted (filled circles) gradient values against age, with the fitted spline from the GAMM with subject random effect (solid red) and from the GAM without subject random effect (solid blue). **B**, Density of four R² metrics across 500 vertices (deviance-based R² `gam_rsq` and `gamm_fixed_rsq`; variance-based R²: `gamm_R2_marginal` and `gamm_R2_conditional`) illustrating that `gamm_R2_conditional` exceeds all others, while `gam_rsq` and `gamm_fixed_rsq` nearly coincide and lie below the variance-based estimates.

Comment 2.12: Perhaps a general comment across these additional analyses would be to add them with some minimal interpretation to the supplement. Specifically I'd be curious to see if the TBV adjustment would have some age-related component to it. Its encouraging to see that the overall trajectories are unaltered by including this covariate, but I could imagine that it may play a role in period where the volumetric tissue changes are somewhat accelerated (i.e., very early development and (very) old age) that wouldn't perhaps be apparent when compared across the entire lifespan.

Response: We thank the reviewer for this insightful suggestion. To formally test whether TBV has a more pronounced age-related effect during periods of accelerated volumetric change, we carried out two new vertex-wise GAM analyses that focus on the life-stages in which intracranial volume changes most rapidly.

First, we refitted the model during infancy (≤ 3 years; 499 scans) using a square-root age axis. Across the vertices, the overall effect of TBV was small, with a median coefficient (β_{TBV}) of -0.008 and an inter-quartile range from -0.056 to $+0.042$.

To illustrate the maximal potential influence of this variable, we examined the vertex with the largest absolute effect size (v_{1280}), which is plotted in **Figure R6d**. For this single vertex, the effect is substantial: the predicted difference between children at ± 2 standard deviations of mean TBV was 1.12 gradient-units, which corresponds to 32.4% of that vertex's total dynamic range. Crucially, however, such strong, statistically significant effects are exceptionally rare, occurring in only $< 0.5\%$ of the vertices.

The same procedure was repeated for late old age (≥ 80 years; 102 scans) on a linear age scale. Here, the median β_{TBV} was near zero ($+0.0018$), and no vertex reached FDR significance.

These results indicate that while TBV's influence is negligible across the vast majority of the cortex, it can exert a material influence on the functional gradient values of a small subset of vertices during the rapid volumetric changes of infancy. However, given that these strong effects are highly localized and are absent in late old age, we conclude that TBV is not a *general* confounder of the principal lifespan trajectories presented in our main findings. Its contribution appears to be a localized, early-life phenomenon rather than a systematic, lifespan-wide bias.

Figure R6. Vertex-wise sensitivity of FC-gradient values to total brain volume (TBV) in the two life-stages of greatest volumetric change. **a**, Distribution of TBV coefficients (β_{TBV}) estimated in infancy (≤ 3 y; $N = 499$). While the distribution is centered near zero, 0.5% vertices (in the tails) survive FDR correction ($q < 0.05$). **b**, Corresponding distribution for late old age (≥ 80 y; $N = 102$), where no vertex is FDR-significant. **c**, Side-by-side violin and box plots summarizing the infancy (left) and old-age (right) coefficient distributions. **d**, Population-level trajectories for the vertex (v_{1279}) that yields the largest model-predicted TBV effect in infancy. Fixing TBV at -2 SD (black) versus $+2$ SD (red) offsets the curve by 1.12 gradient-units, which equals 32.4% of that vertex's observed range, illustrating a material brain volume effect in this extreme case.

Comment 2.13: Very nice to see :)

Comment 2.14: Thank you for the clarification.

Comment 2.15: Where are these comparisons in the manuscript/supplement?

Response: In the revised manuscript, we include Figure S13, which details the between-network result using the centroid approach. Whereas the previous version of our manuscript used the between-network dispersion normalized by global dispersion, we now present the unnormalized version for enhanced interpretability.

Comment 2.16: Thank you for the clarification.

Comment 2.17: Agreed with the authors that this adds to the results.

Comment 2.18: Agreed that this is better and closer to existing literature.

Comment 2.19: Thanks :)

Comment 2.20: Its still in the title though :)

Response: We understand the criticism of our title, and although we think it is an apt and eye-catching characterization of our work, we are open to the judgment of the editorial team as to whether or not it is ultimately appropriate. We remain open to suggestions.

Comment 2.21: Thank you this is helpful I think.

Referee #3

Comment 3.1: Taylor et al. have revised their examination of functional connectivity (FC) gradients across the human lifespan and have satisfactorily addressed my prior relatively minor concerns. The revised manuscript includes the incorporation of an additional dataset (Healthy Brain Network (HBN)) along with examination of potential covariates or interest/confounders, such as biological sex. The overall results seem to hold.

The initial review requested that the authors address the Neuron perspective published by Petersen et al. in which they challenged the fundamental validity of pursuing gradient-based analyses based on the well-established principle of arealization – asserting that cortical areas are valid entities and implying that gradients are likely fictions arising from smoothing across large samples. The authors mention this concern in the last paragraph of the discussion: “There is strong evidence that gradients of FC, microstructure, and gene expression represent crucial organizing axes of the cerebral cortex [2]. Still, recent pushback about the power of gradients as a comprehensive framework for understanding topographic organization of the cortex bears mentioning. The principle of arealization, whereby the cortex is divided into discrete areas which are differentiated by cytoarchitecture and connectivity, is in some ways at odds with the claim that gradients represent fundamental organizing axes on an individual level [66]. Gradients of functional connectivity typically do not smoothly vary with respect to space on an individual level, reflecting instead sharp boundaries between functional systems. At a population level, however, gradients are smooth and excel at describing functional topography. The interpretation of FC gradients in individuals remains an important area of future research, however the present study focuses on large-scale population-level effects.”

The phrase “is in some ways at odds...” fails to convey the direct frontal attack on gradients asserted by Petersen and colleagues. To me this harkens back to vigorous debates regarding the physical nature of light – wave vs. particle. We now all accept that there is no resolution of this question – it all depends on the experimental context/purpose. The real test of validity is explanatory utility. The initial Margulies et al. paper on gradients provided a novel explanation for the physical distribution of default mode network centroids based on analyses of the principal gradient. Such an insight had not emerged from arealization-based perspectives. Here, the authors have provided some data potentially bearing on such explanatory utility, but it’s presented as a supplementary table with little explication. For example, what do Eval1, Eval2, and Eval3 refer to? At any rate, 11 of the 60 relationships with aggregate measures from the Mullen in the Baby HCP presented were below an alpha of 0.05, which does support the inference that some may not be type 1 errors, thus indirectly supporting the explanatory utility of these gradients at least in the youngest participants.

However, given the challenge posed by Petersen et al., and the availability of standardized cognitive/motor measures such as the NIH ToolBox from the HCP and HBN datasets, I urge the authors to consider reexamining the relevant question of explanatory utility of their 3 gradients. They are in the best position to relatively rapidly report whether these gradients can truly be asserted to have quasi clinical utility. [Quasi clinical, because the field lacks sufficiently reliable clinical indices, and most of these datasets were not designed to examine such questions.] If they are able to provide further supportive data from examining gradient-behavior relationships, I would suggest that the Petersen et al. challenge be addressed in the paper’s introduction rather than at the end. This is a major scientific challenge – and it merits a frontal response. At the end of the day, disputes will continue regarding the fundamental validity of functional connectivity because we still lack an understanding of the neurobiological bases for this phenomenon. The reasons why all nervous systems expend such massive quantities of energy in maintaining low-frequency correlations across the brain remain profoundly mysterious. Simple Hebbian arguments are belied by elegant experiments such as documenting rapid changes in functional connectivity after wearing a cast on one’s arm and removing it. Presumably synaptic strengthening/weakening does not occur so rapidly. This is the major challenge of 21st century human neuroscience in my opinion, and this manuscript has the potential to advance the relevant science.

Response: Thank you for your thoughtful and highly constructive feedback on our manuscript. We are grateful for your strong support and particularly for your insightful comments regarding the Petersen et al. challenge to the gradient framework.

Following your suggestion, we have re-examined the question of the gradients' explanatory utility using standardized cognitive data. We performed a new analysis linking metrics from our three primary **gradients** to performance on the NIH Toolbox Cognition Battery in the Human Connectome Project-Young Adult (HCP-YA) cohort (N = 1,066).

To further clarify our methods, we have also added text explaining that Eval1, Eval2, and Eval3 reflect the eigenvalues associated with each individual's gradients before alignment—a template-agnostic measure of scale. Additionally, we have updated our analysis of the BCP Mullen scores with a consistent FDR correction, which accounts for the revised results in the manuscript.

The results, now included in the main text as Figure 7 and in Tables S1 and S2, provide strong support for the utility of the gradient framework. We found a robust and widespread association between our gradient metrics and cognitive performance. Most notably, the similarity of an individual's SA gradient topography to the canonical template (SA *cosim*) was significantly associated with performance across every cognitive domain tested, with standardized coefficients (β) up to 0.19 (all $p_{\text{FDR}} < .05$). This demonstrates that the principal organizing axis of the cortex has tangible relevance for explaining individual differences in cognitive abilities—an insight not easily derived from a purely arealization-based perspective.

Also following your suggestion, we have recentered our response to the Petersen et al. critique by addressing it early in the introduction (paragraph #2), borrowing your analogy of the particle-wave duality of light and asserting that the “resolution lies not in choosing one correct description, but in understanding the context in which each provides value. A key test of any framework's validity is therefore its explanatory utility: its ability to provide novel insights into brain function and behavior.”

Comment 3.2: Trivially, on line 605, the authors describe a correlation as “negative” whereas it is manifestly positive both in the figure and in the following text “($\rho = 0.25$, $p < 10^{-58}$).”

In the first paragraph of the discussion, the authors write: “Integrating the vast and diverse body of fMRI literature into an easily interpreted and succinct framework represents a central obstacle to neuroscience; the present study offers a substantial step forward towards achieving this goal.” I object to the use of the word “obstacle.” They mean “challenge” or similar; otherwise, they need to change the sentence structure.

In line 504, they unnecessarily capitalize “early life” as “Early life.” In that same paragraph, they switch from reporting maxima of age-related changes from one decimal point to two. Given the breadth of most of the corresponding confidence intervals (e.g., (95% CI 9.5–17.8 years), such hyperprecision is unwarranted. I recommend staying with one decimal point.

In line 800, the period after “index” is missing. In line 860, “quality” is misspelled.

Response: Thank you for the careful reading of the paper. These have been corrected.

Comment 3.3: I reiterate that this prodigious effort likely represents a substantial advance in human neuroscience and support its publication.

Response: We are grateful for your strong support and your thoughtful reflections and suggestions that helped to substantially strengthen our manuscript.

Manuscript Title: Functional Hierarchy of the Human Neocortex from Cradle to Grave

We once again thank the editor and reviewers for their thoughtful and constructive feedback, as well as for the opportunity to further refine our manuscript. In this revision, we have made primarily minor updates in response to the reviewer comments, with particular attention to strengthening the transcriptomic analyses and clarifying several aspects of our methodological approach.

Referee #2

The authors have made good-faith efforts to address most of my concerns with additional analyses and clarifications. The LOCO analysis, balanced bootstrap validation, and derivative-based inflection point identification are particularly strong additions that increase the confidence in the robustness of the results. However, some (perhaps philosophical) disagreements remain or could be addressed more thoroughly below. In addition, although I think this may be more a role for the editor than for me as a reviewer I do agree with the assessment of reviewer 1 that perhaps the advance from the already published Sun et al paper is somewhat limited and more of methodological nature than foundational nature.

Response:

We thank the reviewer for the thoughtful summary and for recognizing that the LOCO analysis, balanced bootstrap validation, and derivative-based inflection point identification strengthen confidence in the robustness of our results. We agree that some of the remaining issues concern interpretational and “philosophical” framing, and we address these point-by-point below. With respect to the concern that the advance beyond Sun et al. may be primarily methodological, we respectfully view our contribution as conceptually distinct and complementary. Sun et al. primarily quantified age-related changes in mean functional connectivity strength and network segregation, whereas our focus is on the maturation and decline of the brain’s hierarchical organization, as captured by three principal functional gradients. Building on the gradient framework introduced by Margulies et al., and as emphasized by reviewer #3, we provide what is, to our knowledge, the first lifespan atlas of these gradients from infancy through old age; we show their developmental trajectories dissociate, and we relate them to parallel structural axes, transcriptomic profiles, and behavior (NIH Toolbox cognition and Mullen Scales of Early Learning and Neurosynth). These results establish a multiscale, hierarchy-based framework for lifespan functional organization that is not accessible from region-wise connectivity strength alone and, we believe, represent a substantive conceptual advance rather than a purely methodological refinement. We have conferred with the editor regarding the contributions of our work in the context of the existing literature.

Remaining queries:

Comment 1: NODDI/SMSI Clarification (Comment 2.5):

While the authors clarify they use SMSI not NODDI, they somewhat dismissively cite papers about NODDI's use in pediatric populations rather than directly addressing my core concern about contrast changes in early development. I imagine this could be quantitatively tested?

Response: We thank the reviewer for raising this important point and apologize if our previous reply came across as dismissive of the underlying concern. We fully agree that age-related changes in MR contrast during early development could, in principle, affect the stability of microstructural estimates.

Diffusion-weighted images (DWIs) are typically normalized with respect to their corresponding non-diffusion-weighted (diffusion weighting, $b = 0$ s/mm²) image. This normalization substantially reduces between-scan intensity variation relative to structural modalities such as T1- or T2-weighted imaging. Most diffusion models—including the classic diffusion tensor imaging (DTI) model and more advanced models such as NODDI and SMSI—are conventionally fitted to these normalized signals, and this has been standard practice in both adult and pediatric studies. The NODDI studies we cited in the response letter for our previous revision (Vaher et al., 2022; Diamond et al., 2020; Eaton-Rosen et al., 2015; Kunz et al., 2014) illustrate that NODDI microstructural indices can be reliably estimated from these normalized signals in

developing brains without requiring additional contrast harmonization/normalization specific to these younger populations. In our analysis, we further normalized the extracted microstructural features by applying feature-wise and individual-wise z-scoring. After these steps, we did not observe systematic quantitative differences across time points that would necessitate further harmonization or normalization. Finally, we note that SMSI provides improved estimation of microstructural indices compared to NODDI in tissues with elevated water content, as previously demonstrated by Huynh et al. (2020).

Huynh, K. M. et al. Probing tissue microarchitecture of the baby brain via spherical mean spectrum imaging. IEEE Transactions on Medical Imaging, 2020.

Comment 2: Similarly with pipeline consistency (Comment 2.7):

The authors argue against my suggestion of comparing different pipelines (Infant FreeSurfer vs. standard FreeSurfer - to clarify my suggested was to either interleave or bootstrap these pipelines around the inflection point precisely to avoid a seam) by claiming it would introduce a "non-biological seam." While the unified iBEAT-based pipeline approach is defensible, no empirical evidence is provided that this single pipeline performs equally well across all ages in the present context.

Response: We appreciate the reviewer's emphasis on ensuring that surface reconstruction behaves consistently across the entire age span, particularly around the developmental inflection points. Our choice of a unified pipeline was motivated precisely by the desire to avoid a sharp transition between pipelines that could introduce non-biological seams in downstream measures.

Similar to iBEAT, our pipeline was first developed and validated specifically for children from birth to 5 years of age, a period during which rapid anatomical and contrast changes require dedicated processing strategies. After confirming robust performance in this early developmental window, we extended the pipeline to cover the entire lifespan. This extension is relatively straightforward because, by age 5, most MR image contrasts closely resemble those of adults, requiring only relatively minor fine-tuning of the pipeline beyond this point. Note that unified lifespan pipelines are becoming increasingly common - see for example <https://doi.org/10.1101/2024.12.05.627056>.

In the response letter for our previous revision, we included Figure R4, which demonstrated that our pipeline performs well across the full age range. The resulting surfaces are visibly more consistent across the human lifespan than those produced by FreeSurfer and its infant-specific counterpart, Infant FreeSurfer. Cortical surfaces used in the present study also passed manual quality inspection and achieved an Euler number of 2, indicating a well-formed, topologically correct closed surface.

For completeness, we provide additional examples of white matter and pial surfaces spanning the lifespan to further illustrate that the pipeline performs robustly at all ages (**Figure S21**). We include plots of brain tissue volumes versus age, which show smooth and consistent variations across the lifespan. These results indicate that no additional harmonization is needed (**Figure S22**). These figures are now part of Supplementary Information.

Figure S21. Example tissue segmentation maps and cortical surfaces across the human lifespan.

Figure S22. Volumetric trajectories (in mm^3) of white matter and gray matter across the human lifespan.

Comment 3: Age-Bin Construction:

The clarification about using $\alpha = 0.50$ power-law transform for age bins is helpful, but the fundamental concern about whether early developmental periods receive appropriate weighting relative to their biological importance remains partially unresolved and perhaps it could be established first that these trajectories indeed follow such a power law (while I appreciate the circularity in that, there must be a way to provide some more informed weighting).

Response: We appreciate the reviewer's continued attention to how early developmental periods are represented in our framework, and we apologize that our earlier explanations did not clearly distinguish the different roles of the α -transform in our pipeline. We fully agree that early life is disproportionately important for the biology we aim to characterize, and our use of $\alpha < 1$ was designed precisely to ensure these dynamic periods retain appropriate influence on the model fit.

We use $\alpha = 0.5$ (i.e., a square-root transform of age) in two conceptually distinct steps.

First, in *template construction via weighted PCA (wPCA)*, we transform chronological age as $\text{age}' = \text{age}^{0.5}$ and then define age bins that are equally spaced in the transformed age domain. These bin edges are mapped back to chronological age, so that on the original age scale the bins are narrower in early development and wider in later adulthood. We assign equal total weight to each bin and divide this weight equally among subjects within the bin. Thus, per unit chronological time, the earliest periods (for example, 0–2 years) receive substantially more weight than later adulthood:

"In order to align subject-specific gradient axes consistently across the lifespan, we computed a set of 'template gradients' via weighted principal component analysis (WPCA) of all subjects' gradients. We began by applying a controlled non-linear transformation to each subject's age, taking the square root of age to capture rapid changes in early life. The transformed ages were partitioned into ten equally spaced bins on the transformed scale and then mapped back to the original age units, yielding narrower bins (and thus finer sampling) for younger subjects. Each bin was assigned an equal total weight, and that weight was uniformly distributed among the subjects within the bin."

This procedure is not meant to assume that the underlying trajectories follow a power law; rather, it implements a monotonic reweighting that intentionally upweights early developmental ages when estimating the common spatial axes. Importantly, this step only determines how subjects contribute to the template—the group-level gradient axes—rather than the detailed shape of the lifespan trajectories.

Second, in *GAMM fitting and harmonization*, we use the same $\alpha = 0.5$ transform in two related ways. We compute subject weights using the same age-binning strategy described above, again ensuring that early developmental periods receive greater effective weight per unit chronological time when fitting the trajectories. We also use $\text{age}^{0.5}$ as a monotonic reparameterization of age in the smooth terms (for example, $s(\text{age}^{0.5})$):

"Biological changes are more rapid early in life; in order to cater to this, the age variable was transformed by raising age in years to a fractional power, $\alpha = 0.5$, improving model stability and distributional assumptions."

This stretches the age axis so that the smoother has finer resolution in the earliest years while preserving the ordering of ages; it does not impose a specific parametric form on the developmental curves. In other words, $\alpha = 0.5$ is used as a design choice to emphasize early development in both weighting and parameterization, rather than as a claim that the underlying biology obeys a power law.

We also verified that our conclusions do not depend critically on the precise choice of α . For exemplar vertices, we refit the full harmonization and GAMM pipeline using alternative monotonic age transforms that compress adulthood and expand early life ($\alpha = 0.3, 0.4, 0.5, 0.6, 0.7$). The resulting trajectories were qualitatively similar, with closely matching ages at peak and inflection points (**Figure R1**). This indicates that our main conclusions are driven by the general strategy of upweighting early developmental ages, rather than by the particular numerical choice $\alpha = 0.5$.

Taken together, these design choices and robustness checks address the reviewer's core concern: early developmental periods are explicitly given greater effective weight than later adulthood when constructing the template and fitting the trajectories, and the resulting axes and lifespan curves are stable to changes in sampling density and cohort composition.

Figure R1. Sensitivity of weighted GAMM fits to the age transform. Weighted-model lifespan trajectories of gradient value at a representative vertex are shown for several values of α (0.3–0.7). The trajectories are highly similar, indicating that the estimated developmental curve is robust to the specific choice of α .

Referee #3

Taylor and colleagues have responded adeptly to the constructive reviews, thereby further enhancing the value of the approach they have pursued. The relevance of functional gradients is thoroughly established by the data presented and the ingenious analyses performed. The manuscript is well written - with only a single typograph error per my reading. On line 209, they duplicate "association" inadvertently. I appreciate particularly their adoption of the particle/wave metaphor in the introduction. I congratulate the authors on a worthy contribution and happy to be associated with this paper.

Francisco Xavier Castellanos

Response: We sincerely appreciate your supportive and generous feedback. Thank you for noting the duplicated word on line 209; we have corrected this typo.

Referee #4

Taylor et al. present a meta-analysis of multiple human brain imaging datasets ranging in age from infancy to 100 years old. They use these data to model the lifespan trajectory of functional connectivity (FC) gradients. Three previously described axes of FC are analyzed, the sensorimotor-association (SA) axis, the modulation-representation (MR) axis, and the visual-somatosensory (VS) axis. Taylor and colleagues find that the developmental trajectories of these three axes are independent and non-linear though all three show the largest changes between 0-4 years of life. To ground these models in biological underpinnings the authors relate the FC gradients to metrics of intelligence, structural fMRI measures such as myelin thickness, and transcriptomic enrichment. I appreciate the addition of these analyses as they do add context to otherwise quite abstract descriptions of brain organization. With regard to the transcriptomic enrichment analysis in particular, I have three major and two minor comments that I believe will bolster the value of this addition.

Major:

Comment 1: The authors currently only perform PLS analysis against the SA gradient. It would enhance the validity of this section by including at least one of the other axes (I would suggest the MR axis). The other behavioral analyses include all three FC gradients and by doing so the authors can draw conclusions about how the variations of the gradient trajectories may relate to these metrics. Without another FC axis to compare to in the transcriptomic enrichment, I am left wondering if this is in fact a hallmark of the SA gradient or if the other gradients would show similar transcriptomic and age-related patterns.

Response: We thank the reviewer for this suggestion. In the revised manuscript we have extended the transcriptomic PLS analysis to all three functional gradients (SA, VS, and MR), using the same framework that we previously applied only to the SA axis. Specifically, for each axis we now fit a PLS model at the same set of ages (0.5, 2, 10, 25, 40, 80 years), orient the first component so that higher scores track higher mean gradient, and perform GO enrichment on genes ranked by their PLS coefficients. We describe our analyses of the VS and MR gradients in the updated “Transcriptomic enrichment of SA, MR, and VS gradients across development” subsection:

“To relate cortical gene expression to the parcel-wise mean gradient values across development, we fit a partial least squares model (PLS) at select ages (0.5, 2, 10, 25, 40, 80 years) for each of our gradient axes. PLS was trained on adult human cortical expression (Allen Human Brain Atlas; abagen pipeline, Schaefer-200 parcellation) and oriented so that the correlation between the first PLS component parcel score and the mean gradient value was non-negative at each age. The resulting component maps reveal smooth molecular axes across cortex: regions with higher scores exhibit expression profiles that are more aligned with higher mean gradient values, whereas low-scoring regions align with lower mean gradient values.

For the SA axis, the PLS map at 25 years (Figure S17a) recapitulates the SA hierarchy. To interpret this component, we ranked genes by their PLS coefficient and performed gene ontology (GO) enrichment (GORilla) at each age. The top term at 25 years, trans-synaptic signaling, was projected back to cortex by computing a parcel-wise gene-set score from its member genes (coefficient-weighted mean of z-scored expression). The resulting map (Figure S17b) has topographic similarity to the PLS map and the SA axis, indicating that synaptic signaling genes contribute strongly to the positive pole of the molecular axis that predicts higher SA values. Summarizing individual GO terms into coarse biological themes using keyword rules (Figure S17c), we observed a persistent synaptic signaling signature across ages, accompanied by a vesicle cycle theme. Ion transport/excitability varied with age, and extracellular matrix/adhesion/migration, immune/microglia, and cell cycle/proliferation themes appeared selectively at particular ages. Together, these results suggest that synaptic and vesicle trafficking programs consistently track spatial variation in the SA axis, with additional processes modulating the association at specific developmental stages. Across ages, the correlation between PLS scores and the mean SA gradient (permutation-tested and Benjamini–Hochberg-corrected across ages) peaked in early life and gradually

declined into adulthood (Figure S17d), indicating that a fixed adult-like molecular axis explains more SA topography during early life than during adulthood.

Applying the same procedure to the VS axis yielded a PLS component with highest scores in visual cortex and lower scores in somatosensory and association territories (Figure S18a). GO enrichment of genes ranked by their VS PLS coefficients revealed terms related to axon and dendrite transport, mRNA transcription and RNA metabolic processes, and vesicle cycling. When aggregated into themes, vesicle cycling showed its strongest enrichment in early life (peaking around 2 years), whereas axon/dendrite transport and transcription/RNA metabolism peaked in young to mid-adulthood (Figure S18c). The most significant VS gene set, regulation of mRNA metabolic process, produced a parcel-wise gene-set score map that closely resembled the VS PLS component, with highest values in visual cortex, reinforcing that transcriptional and RNA-processing programs are selectively engaged along the VS axis (Figure S18b). Gene–gradient coupling for the VS axis was highest during infancy, declined sharply through childhood, remained relatively stable into mid-adulthood, and declined again towards 80 years (Figure S18d), indicating an early sensitive period of strong alignment followed by a more stable adult regime and late-life decoupling.

For the MR axis, the first PLS component exhibited negative scores in visual and premotor cortex and positive scores in medial and lateral prefrontal cortex and anterior temporal regions (Figure S19a), mirroring the contrast between unimodal representational regions and higher-order control regions exhibited by the MR axis. The top GO term, detoxification of copper ion, when projected back to cortex, produced a gene-set score map with lowest values in visual and motor cortex and highest values in medial prefrontal and anterior temporal cortex, closely matching the MR PLS topography (Figure S19b). At the level of themes, ion transport/excitability emerged as the most consistent signal, with its strongest enrichment in late life (Figure S19c). This pattern suggests that genes involved in ionic homeostasis and excitability contribute selectively to the positive, control-dominated pole of the MR molecular axis. The correlation between MR PLS scores and the mean gradient decreased monotonically across the lifespan (Figure S19d), indicating that the alignment between a fixed adult-like homeostatic and excitability-related axis and the MR gradient is strongest in early development and weakens with age.“

The updated Results subsection titled “Transcriptomic enrichment of SA, MR, and VS gradients across development” now describes these analyses for all three axes, and the VS and MR results are shown in new supplementary figures (Figure S18 for VS and Figure S19 for MR). As summarized in the final paragraph of that subsection and in the expanded Discussion, we find that the three axes are underpinned by distinct but overlapping molecular programs: synaptic signaling and vesicle cycling for the SA axis, transcriptional and RNA-metabolic regulation with a visual emphasis for the VS axis, and ionic homeostasis and excitability for the MR axis.

“Taken together, these analyses show that all three functional gradients are underpinned by distinct but overlapping molecular programs: synaptic signaling and vesicle cycling for the SA axis, transcriptional and RNA-metabolic regulation with a visual emphasis for the VS axis, and ionic homeostasis and excitability for the MR axis. In each case, gene–gradient coupling is strongest early in life and weaker in later adulthood, suggesting that adult-like spatiomolecular axes provide a scaffold for early differentiation of functional connectivity that becomes progressively less influential on functional organization throughout the lifespan.”

In each case, gene–gradient coupling is strongest in early life and weaker in later adulthood. These additions allow direct comparison across axes and show that SA-like transcriptomic coupling is not unique, but is part of a broader, axis-specific spatiomolecular organization.

Comment 2: The analysis of this result is fairly superficial. It is not surprising that the strongest related GO terms are in the synaptic signaling family given that the underlying mechanism of functional gradients is in fact synaptic activation. I suggest

the authors dig a little deeper into this GO term family for more nuanced results. For example, Gao et al. 2020 (DOI: <https://doi.org/10.7554/eLife.61277>) perform a similar analysis against ECoG recordings and find that genes for ion channels known to mediate fast versus slow synaptic dynamics correlate with different timescales of brain activity. The study presented here explores timescales well beyond this, but it's possible that genes related to, for instance, neuromodulators vs neurotransmitters could have different PLS weights.

Response: We agree that our initial presentation of the gene enrichment results did not fully convey the richness of the patterns, and we appreciate the pointer to Gao et al. (2020). In the previous version, we already grouped GO terms into broader themes (for example, synaptic signaling, vesicle cycle, ion transport/excitability, extracellular matrix/adhesion/migration, immune/microglia, cell cycle/proliferation) and projected the top enriched gene set back to the cortex. In this revision, we build on that framework in two main ways.

First, we now apply the same PLS and GO-enrichment pipeline to the VS and MR axes, and describe the resulting theme profiles across age for each axis. This makes the differences between axes more explicit: for example, along the VS axis we find enrichment for vesicle-related processes in early life and for axon/dendrite transport and transcription/RNA-metabolic themes in young to mid-adulthood, whereas along the MR axis we see a particularly strong signal in ion transport and metal-ion handling.

Second, we inspected the highest-weighted genes and GO terms for each axis and expanded the interpretation in the Discussion. In particular, for the MR axis we highlight that the dominant GO signals involve ion transport, metal-ion homeostasis, and excitability-related processes (including detoxification of copper ion and related themes), and we explicitly link this pattern to work showing that regional intrinsic timescales are shaped by the expression of ion-channel and excitation–inhibition–related genes, as reported by Gao et al. (2020). This leads us to propose that the MR axis may partly reflect a dimension of temporal integration capacity and neurometabolic resilience in control-dominated cortex, rather than only a generic synaptic signature.

“To anchor these developmental patterns in underlying biology, we related parcel-wise adult cortical expression (AHBA; abagen) to the mean gradient at each age using a supervised PLS approach. Because the Allen Human Brain Atlas is derived from adult post-mortem donors, these PLS components should be interpreted as adult-like molecular axes that we evaluate against age-varying functional gradients, rather than as direct readouts of developmental changes in gene expression. Across all three axes, the first PLS component defined a smooth “molecular axis” whose parcel scores were oriented such that higher scores tracked higher mean gradient, and gene–gradient coupling was generally strongest in early life and weaker in later adulthood. For the SA axis, this molecular axis closely followed the classic sensorimotor-association hierarchy and was consistently enriched for synaptic signaling and vesicle cycle processes, with age-specific modulation by ion transport, extracellular matrix/adhesion, immune/microglial, and cell-cycle themes. These findings reinforce the idea that the SA axis is rooted in regional differences in synaptic machinery and vesicle trafficking, and suggest that a relatively fixed, adult-like pattern of synaptic gene expression provides a scaffold that is most predictive of SA topography during early life, when large-scale hierarchy is still consolidating.

The VS and MR axes revealed distinct but complementary molecular programs. Along the VS axis, the molecular component peaked in visual cortex and was enriched for vesicle-related processes in early life and for axon/dendrite transport and transcription/RNA-metabolic terms in young to mid-adulthood. The most significant VS gene set, regulation of mRNA metabolic process, recapitulated the VS topography, implicating transcriptional and RNA-processing programs in shaping visual–somatosensory differentiation. In contrast, the MR axis, which contrasts unimodal representational regions with higher-order control networks, was associated with a molecular gradient that loaded negatively in visual and premotor cortex and positively in medial and lateral prefrontal and anterior temporal regions. Here, the dominant GO signals involved metal ion handling and ion transport/excitability, including detoxification of copper ion and related themes, implicating ionic homeostasis and neurometabolic support mechanisms in the positive, control-

dominated pole of the MR axis. Notably, the enrichment of ion transport and metal-ion homeostasis along the MR gradient resonates with work showing that regional intrinsic timescales are shaped by the expression of ion-channel and excitation-inhibition-related genes (Gao et al., 2020). This convergence raises the possibility that the MR axis partly reflects a dimension of temporal integration capacity and neurometabolic resilience--supporting sustained, modulatory computations in transmodal cortex--superimposed on the more classically hierarchical SA organization.

Our findings also fit within, and extend, recent work on whole-brain spatiomolecular gradients. Vogel et al. (2024) and Xia et al. (2022) identified large-scale gene-expression gradients in the adult brain that align with broad anatomical and functional motifs, and argued that these spatiomolecular axes constitute a developmental “scaffold” for adult functional specialization. Here, by focusing on three well-characterized functional gradients and tracking their coupling to transcriptomic axes across the lifespan, we show that this scaffold is not uniform but is differentiated into at least three partially dissociable molecular programs: synaptic/vesicle-related for the SA hierarchy, transcriptional/RNA-regulatory with a visual emphasis for the VS axis, and ionic homeostasis/excitability for the MR axis. The observation that gene–gradient coupling is strongest in infancy and early childhood and weaker in later adulthood suggests that adult-like spatiomolecular gradients may initially exert strong constraints on functional topography, which is then progressively refined by experience-dependent plasticity, structural maturation, and age-related change. In this view, developmental reorganization of the three functional gradients reflects a shift from a regime in which large-scale gene-expression patterns tightly anchor functional architecture to one in which these molecular axes remain present but exert a more diffuse influence on mature, dynamically specialized cortical networks. Future work could test this account by examining whether intrinsic timescale maps and independently derived spatiomolecular gradients preferentially align with the MR and SA molecular axes, providing a direct link between ionic homeostasis, temporal integration, and large-scale hierarchy.”

Taken together, these changes move the analysis beyond the unsurprising observation that broad synaptic genes are involved and towards a more nuanced characterization of how synaptic/vesicle, transcriptional/RNA-regulatory, and ionic homeostasis/excitability programs are differentially weighted across axes and across the lifespan.

Comment 3: The discussion of these results is largely without interpretation or connection to prior literature on this topic. At least two recent studies have performed similar analyses with much more in-depth characterization of spatial molecular programs and functional gradients; Vogel et al. 2022 (DOI: <https://doi.org/10.1073/pnas.2219137121>) and Yunman et al. 2022 (DOI: <https://doi.org/10.1016/j.scib.2022.01.002>). I encourage the authors to speculate on what the relationship between these transcriptomic patterns and gene gradients could mean for human brain development. This will be particularly important with the addition of another FC gradient axis in the analysis whether the results are similar or different.

Response: We agree that the earlier Discussion did not sufficiently develop the implications of the transcriptomic findings or connect them to existing work on spatiomolecular gradients. We have therefore substantially expanded the relevant part of the Discussion.

In the revised text, we first synthesize the axis-specific enrichment patterns, emphasizing that the SA axis is associated with synaptic signaling and vesicle cycle processes, the VS axis with transcriptional and RNA-regulatory programs (with a visual emphasis), and the MR axis with ionic homeostasis and excitability-related themes.

We then explicitly relate these patterns to prior work. We connect the MR ion-transport and metal-ion signatures to Gao et al. (2020), who showed that intrinsic timescales correlate with expression of ion-channel and excitation–inhibition–related genes. We suggest that the MR axis may partly index a dimension of temporal integration capacity and neurometabolic support in the transmodal cortex. We also discuss how our results align with the whole-brain spatiomolecular gradients described by Vogel et al. (2022) and Xia et al. (2022), showing that the molecular scaffold fractionates into at least three partially dissociable programs aligned with SA, VS, and MR.

Finally, we explicitly speculate on developmental implications: we propose that adult-like spatiomolecular gradients may provide a scaffold that most strongly constrains functional topography in infancy and early childhood—when large-scale hierarchy and specialization are still consolidating—and that their influence becomes more diffuse in adulthood as structural maturation and experience-dependent plasticity reshape the gradients.

Minor:

Comment 4: I see in the methods that the authors performed parcellation shuffling as a control. It would be a nice addition to add a visual of this to the figure to ensure validity of the PLS results.

Response: We appreciate this suggestion. In the revision, we have added a new Supplementary figure (Figure S20) that explicitly visualizes the parcellation-shuffle control for all three axes. For each axis and age, we recomputed the PLS1 model after randomly permuting the mean gradient values across parcels 2,000 times, yielding a null distribution of the absolute correlation between PLS1 parcel scores and mean gradient. Figure S20 shows, for each age and axis, a boxplot of this null distribution (median and 2.5–97.5th percentiles), with the empirical coupling plotted as a point. In every case, the observed coupling lies well above the upper tail of the null, indicating that the PLS associations reflect spatially specific gene–gradient alignment rather than idiosyncrasies of the parcellation. We briefly describe this analysis in the figure legend and Methods:

“Figure S20: Gene–gradient coupling compared to parcel-shuffle null models for each functional axis. a–c, For each age and axis, we quantified gene–gradient coupling as the absolute Pearson correlation between parcel-wise PLS1 scores and mean gradient values. Gray boxplots show the distribution of coupling values obtained under a parcel-shuffle null model, in which gradient values were randomly permuted across parcels (2,000 permutations per age), and whiskers denote the 2.5–97.5th percentiles. Black points show the observed coupling for the (a) SA, (b) VS, and (c) MR axes. For all three gradients and all ages, the empirical coupling exceeds the upper tail of the null distribution, confirming that the PLS associations reflect spatially specific gene–gradient alignment rather than parcellation idiosyncrasies.”

Methods: Transcriptomic Enrichment

“To visualize these null distributions, for each gradient axis and age we refit the PLS1 model after shuffling parcel labels 2,000 times and plotted the resulting absolute correlations as boxplots, with the empirical (unshuffled) correlation overlaid as a point (Figure S20).”

Comment 5: The AHBA dataset used for this section consists of only adult donors. This potentially causes a confound when applying to very young and very old ages in the FC data. I appreciate that there are few resources like this available and would just consider acknowledging the limitation.

Response: We agree that this is an important limitation and have now made it explicit in the text. As the reviewer notes, the Allen Human Brain Atlas is derived from adult post-mortem donors; our approach therefore evaluates how adult-like molecular patterns relate to functional gradients measured across the lifespan. In the revised Results subsection on transcriptomic enrichment, we clarify that PLS is trained on adult cortical expression and that the resulting components should be interpreted as adult-like molecular axes that are evaluated against age-varying gradients. In the Discussion, we explicitly acknowledge that this approach does not capture potential developmental changes in gene expression itself, and we frame our findings as suggesting that adult-like spatiomolecular gradients provide an early scaffold that is most predictive of functional organization in infancy and childhood:

“To anchor these developmental patterns in underlying biology, we related parcel-wise adult cortical expression (AHBA; abagen) to the mean gradient at each age using a supervised PLS approach. Because the Allen Human Brain Atlas is derived from adult post-mortem donors, these PLS components should be interpreted as adult-like molecular axes that we evaluate against age-varying functional gradients, rather than as direct readouts of developmental changes in gene expression.”

We emphasize in our response here that age-resolved human transcriptomic resources will be required in future work to determine how the molecular axes themselves evolve across development and aging.

Referee #5

I co-reviewed this manuscript with one of the reviewers who provided the listed reports.

Response: Thank you for the constructive feedback.

Response to Referees

Referee #2

We sincerely thank Referee #2 for the careful re-review and for the consistently rigorous, methodologically focused feedback throughout the review process. We appreciate the referee's close attention to pipeline validity, normalization/contrast considerations, and age-dependent processing effects, and we are grateful for the constructive tone and specificity of the recommendations. In our view, the referee's comments substantially strengthened the manuscript—particularly by motivating clearer technical explanations and more direct empirical demonstrations of stability across the lifespan. We also appreciate the referee's acknowledgment that the revised version provides reasonable, evidence-based justification for our unified processing approach.

Referee #4

We sincerely thank Dr. Seeman for the positive evaluation of the revision and for noting that the transcriptomic enrichment analysis now reads as a more substantial and integrated component of the study. We also appreciate the careful eye in catching two remaining text errors in the supplementary figures.

Figure label corrections:

Comment: In Fig. S18c the heatmap title says "SA" but should be "VA." In Fig. S20c the title says "(SA axis)" but should be "(MR axis)."

Response: Thank you—these were indeed text/labeling errors. We have corrected both.